# Neural Emulator Superiority: When Machine Learning for PDEs Surpasses its Training Data

**Felix Koehler** & **Nils Thuerey**
Technical University of Munich
Munich Center for Machine Learning (MCML)
{f.koehler,nils.thuerey}@tum.de

## Abstract

Neural operators or emulators for PDEs trained on data from numerical solvers are conventionally assumed to be limited by their training data's fidelity. We challenge this assumption by identifying "emulator superiority," where neural networks trained purely on low-fidelity solver data can achieve higher accuracy than those solvers when evaluated against a higher-fidelity reference. Our theoretical analysis reveals how the interplay between emulator inductive biases, training objectives, and numerical error characteristics enables superior performance during multi-step rollouts. We empirically validate this finding across different PDEs using standard neural architectures, demonstrating that emulators can implicitly learn dynamics that are more regularized or exhibit more favorable error accumulation properties than their training data, potentially surpassing training data limitations and mitigating numerical artifacts. This work prompts a re-evaluation of emulator benchmarking, suggesting neural emulators might achieve greater physical fidelity than their training source within specific operational regimes.

Project Page: tum-pbs.github.io/emulator-superiority

## 1 Introduction

The numerical simulation of physical systems governed by partial differential equations (PDEs) involves an inherent tradeoff between computational cost and accuracy. Traditional numerical methods such as finite difference, finite element, and spectral approaches have well-understood error characteristics stemming from modeling, discretization, linearization, and iterative solver truncation. These errors can be systematically reduced by refining the physical model, increasing resolution, employing higher-order schemes, or tightening solver tolerances—but always at the cost of increased computational burden.

Machine learning approaches to PDE solving have emerged as promising alternatives, with neural networks trained to emulate the behavior of numerical solvers while offering computational advantages (Li et al., 2021a; Kochkov et al., 2024). The conventional wisdom suggests that neural emulators inherit the limitations of their training data: a neural network trained on data from a low-fidelity solver should, at best, reproduce the accuracy of that solver.

In this paper, we demonstrate that this conventional wisdom does not always hold. We identify and analyze a phenomenon we call "emulator superiority," in which neural networks trained on data from low-fidelity numerical solvers can outperform those same solvers when tested against high-fidelity references. For this counterintuitive result, we provide both theoretical proof for linear PDEs (advection, diffusion, Poisson) and empirical evidence across linear advection and the nonlinear Burgers equation using diverse neural network architectures (convolutional networks, dilated ResNets, FNOs, UNets, Transformers). Crucially, this observation is different from previous work that learned improved coarse solvers from high-fidelity data (Bar-Sinai et al., 2019), mixed low-fidelity data with

39th Conference on Neural Information Processing Systems (NeurIPS 2025).

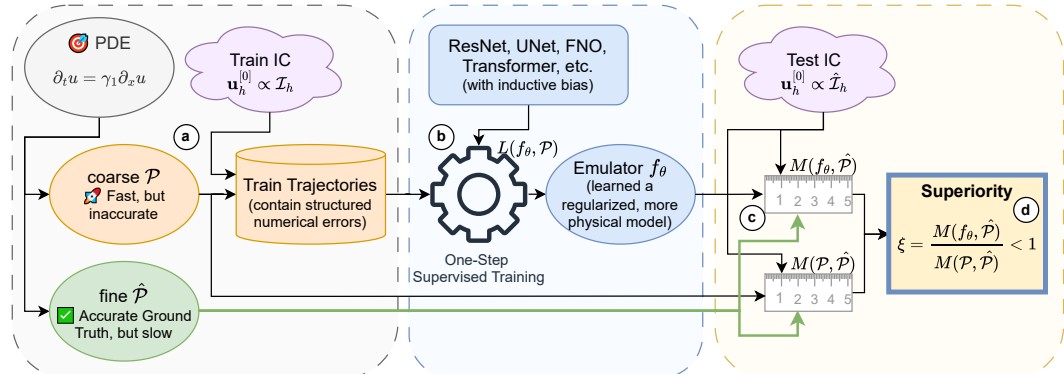

Figure 1: A PDE can be solved using different numerical algorithms. A coarse solver $\mathcal{P}$ produces data trajectories with inherent numerical errors (a). Neural emulators have inductive biases that enable them to learn a regularized, more physical model $f_\theta$ (b). When we measure the accuracy of both emulator $f_\theta$ and coarse simulator $\mathcal{P}$ against a higher fidelity ground truth $\hat{\mathcal{P}}$ (c), the networks can exceed the training reference, giving rise to "emulator superiority" (d).

high-fidelity data (Lu et al., 2022) or enhanced coarse data using physics-informed residuum soft penalizers (Li et al., 2021b). Instead, we *purely* train the neural emulator on low-fidelity data and still observe this *superiority*.

We do not suggest that emulator superiority is a universal guarantee; its occurrence and intensity are dependent on the specific PDE, the characteristics of the low-fidelity solver's errors, the chosen emulator architecture, the training objective, the evaluation metric, and the time horizon. However, demonstrating that such superiority exists, even within specific regimes, fundamentally challenges the notion that neural emulators are irrevocably bound by the accuracy limitations of their training data.

This finding suggests that neural emulators might implicitly learn regularized or corrected dynamics, potentially exceeding the fidelity of their low-fidelity training data. Furthermore, it prompts a re-evaluation of benchmarking practices, as emulators could potentially mitigate numerical artifacts even in high-fidelity reference solvers. Since imperfections, e.g., due to insufficient resolution, numerical diffusion, or non-converged iterative schemes are often practically unavoidable or specifically introduced to reduce compute cost (Turek, 1999; Müller et al., 2007), imperfections permeate into data trajectories used to train machine learning models. If the same data source is used for benchmarking, this creates the paradox that a better emulator might be unfairly penalized.

Our contributions are:

- We formalize the differences between training objective and inference task of autoregressive neural emulators to identify the concept of superiority.

- We provide theoretical proof for the existence of *state-space superiority* for the three major linear protypical PDEs using Fourier analysis. For this, we identify both forward and backward superiority based on how the superiority occurs spectrally.

- We confirm *state-space* and *autoregressive superiority* in emulator learning for the advection and Burgers equation using a wide range of nonlinear neural architectures.

## 2    Classical and Neural Solutions to PDEs

Numerical methods approximate solutions to continuous PDEs by transforming them into discrete problems solvable by computers. A general PDE describing the evolution of a field $u(t, \mathbf{x})$ over time $t$ and space $\boldsymbol{x} \in \Omega \subseteq \mathbb{R}^D$ can be written as

$$\partial_t u = \mathcal{L}(u), \tag{1}$$

where $\mathcal{L}$ is a spatial differential operator, which may be nonlinear and potentially inhomogeneous.

## 2.1 Numerical PDE Solvers and Error Sources

Classical numerical solvers are based on first-principled symbolic manipulations of the continuous description. Discretization in space (using methods like finite differences, finite elements, or spectral methods) and time (e.g., Euler steps, Runge-Kutta methods) leads to numerical schemes. Many such schemes can be expressed as discrete-time, first-order Markovian processes:

$$\mathbf{u}_h^{[t+1]} = \mathcal{P}_h(\mathbf{u}_h^{[t]}), \tag{2}$$

where $\mathbf{u}_h^{[t]}$ is the discrete state vector at time step $[t]$ on a grid characterized by $h$, and $\mathcal{P}_h$ is the discrete evolution operator encapsulating the numerical method and respecting boundary conditions.

The accuracy of $\mathcal{P}_h$ is limited by several error sources: (1) *spatial discretization errors*, (2) *time discretization errors*, (3) *linearization errors* when approximating nonlinear terms, and (4) *iterative solver errors* from terminating iterative methods prematurely. These errors can accumulate or interact over time, potentially leading to instability (often characterized by exponential error growth).

Different numerical schemes exhibit varying trade-offs regarding stability, accuracy, and computational cost. For example, implicit methods often allow larger time steps (better stability) at the cost of increased numerical diffusion (a form of spatial error), while high-order methods reduce discretization errors but can be computationally expensive or are prone to oscillations (LeVeque, 2007). Spectral methods can offer high spatial accuracy for smooth solutions but are typically restricted to simpler geometries (Boyd, 2001). The choice of domain, boundary conditions, and specific physical parameters further influences solver performance and error behavior.

## 2.2 Machine Learning Approaches for PDE Solving

The potential for computational speedups and tackling unsolved problems has motivated the use of neural networks (NNs) for helping solve PDEs. We are interested in approaches that learn operators that map function spaces, analogously to the numerical evolution operator $\mathcal{P}_h$ in equation 2. However, we relax the strict requirement of discretization invariance common in the neural operator literature (Kovachki et al., 2023; Gao et al., 2025) and instead focus on networks trained and applied at a *fixed* discretization $h$. We refer to such a neural surrogate $f_\theta \approx \mathcal{P}_h$ as a *neural emulator* (Koehler et al., 2024). Specifically, we consider *autoregressive* neural emulators that operate similarly to equation 2, predicting the next state based only on the current state: $\mathbf{u}_h^{[t+1]} \approx f_\theta(\mathbf{u}_h^{[t]})$. While one can motivate non-Markovian approaches for states with truncated spectra (Ruiz et al., 2025), the Markovian assumption allows for a direct comparison with traditional numerical time steppers.

**Supervised Training for Autoregressive Emulators**   A common paradigm for training an autoregressive emulator $f_\theta$ involves supervised learning on data generated by a chosen numerical solver $\mathcal{P}_h$. First, trajectories are generated by rolling out the solver from various initial conditions $\mathbf{u}_h^{[0]} \sim \mathcal{I}_h$:

$$\mathcal{T}_{\mathcal{P}_h}(\mathbf{u}_h^{[0]}) = \left\{\mathbf{u}_h^{[0]}, \mathbf{u}_h^{[1]}, \ldots, \mathbf{u}_h^{[S]}\right\} = \left\{\mathbf{u}_h^{[0]}, \mathcal{P}_h(\mathbf{u}_h^{[0]}), \ldots, \mathcal{P}_h^S(\mathbf{u}_h^{[0]})\right\}.$$

Pairs of consecutive states $\{\mathbf{u}_h^{[t]}, \mathbf{u}_h^{[t+1]}\}$ are sampled from these trajectories to form a dataset. The emulator $f_\theta$ is then trained to minimize a *one-step prediction error* objective:

$$L(\theta) = \mathbb{E}_{\{\mathbf{u}_h^{[t]}, \mathbf{u}_h^{[t+1]}\} \sim \mathcal{T}_{\mathcal{P}_h}(\mathbf{u}_h^{[0]}), \, \mathbf{u}_h^{[0]} \sim \mathcal{I}_h} \left[\zeta\left(f_\theta(\mathbf{u}_h^{[t]}), \mathbf{u}_h^{[t+1]}\right)\right], \tag{3}$$

where $\zeta$ is a metric, e.g., a mean squared error (MSE). Variations exist, such as unrolled training, which minimizes errors over multiple steps (Brandstetter et al., 2022; List et al., 2025), or incorporating physical constraints (Li et al., 2021b). However, the fundamental principle remains learning to mimic the behavior of $\mathcal{P}_h$.

**Generalization Challenges.**   Autoregressive emulators are evaluated by recursively rolling out the model for multiple time steps and comparing to a reference trajectory,

$$e^{[t]} = \mathbb{E}_{\mathbf{u}_h^{[0]} \sim \tilde{\mathcal{I}}_h} \left[\tilde{\zeta}\left(f_\theta^t(\mathbf{u}_h^{[0]}), \tilde{\mathcal{P}}_h^t(\mathbf{u}_h^{[0]})\right)\right]. \tag{4}$$

Here, $f_\theta^t$ denotes applying the emulator $t$ times autoregressively. The test setup (denoted by tildes) might differ from the training setup in several ways, probing different aspects of generalization.

**Temporal/Autoregressive generalization** involves evaluating long-term accuracy and stability by testing for $t > 1$, i.e., for more autoregressive applications than during training. Exponential error growth $e^{[t]}$ usually signals instability. **State-space generalization** examines performance on different initial conditions $\tilde{\mathcal{I}}_h \neq \mathcal{I}_h$ or over extended time horizons beyond those used for training data generation, where the physics might enter a regime that is not well-represented by the training data. **Metric generalization** refers to evaluating the emulator using a different metric $\tilde{\zeta}$ than the one used for training $\zeta$. This allows for probing specific aspects of performance, such as accuracy in higher frequencies/modes. **Solver generalization** evaluates against a different numerical simulator $\tilde{\mathcal{P}}_h \neq \mathcal{P}_h$, where both solvers are consistent with the same underlying PDE equation 1 but employ different numerical implementations, varying in aspects such as scheme order or solver tolerance.

Emulators can be tested for a single type of generalization or a combination of many. This paper focuses on combining solver generalization with temporal or state-space generalization. We are particularly interested in the scenario where the emulator $f_\theta$ is trained on data from a *low-fidelity* solver $\mathcal{P}_h$ but evaluated against a *high-fidelity* reference solver $\tilde{\mathcal{P}}_h$. This sets the stage for investigating whether $f_\theta$ can outperform its own training data source $\mathcal{P}_h$ when compared to the more accurate reference $\tilde{\mathcal{P}}_h$ in different regimes of physics or for multiple autoregressive evaluation steps.

## 2.3 Comparing Emulators and Emulator Superiority

Given two neural emulators $f$ and $\breve{f}$ in order to determine which one is better we could evaluate their rollout error $e^{[t]}$ and $\breve{e}^{[t]}$, respectively. Then, if $\xi^{[t]} = e^{[t]}/\breve{e}^{[t]} < 1$ we know that $f$ is better than $\breve{f}$ at rollout step $[t]$ (under given $\tilde{\zeta}$, $\tilde{\mathcal{I}}_h$ and $\tilde{\mathcal{P}}_h$). Now, assume $\breve{f}$ was not another baseline neural emulator, but the training simulator $\mathcal{P}_h$ used to inform $f_\theta$. Then, we can define the superiority ratio as

$$
\xi^{[t]} = \frac{\mathbb{E}_{\mathbf{u}_h^{[0]} \sim \tilde{\mathcal{I}}_h} \left[ \tilde{\zeta} \left( f_\theta^t(\mathbf{u}_h^{[0]}), \tilde{\mathcal{P}}_h^t(\mathbf{u}_h^{[0]}) \right) \right]}{\mathbb{E}_{\mathbf{u}_h^{[0]} \sim \tilde{\mathcal{I}}_h} \left[ \tilde{\zeta} \left( \mathcal{P}_h^t(\mathbf{u}_h^{[0]}), \tilde{\mathcal{P}}_h^t(\mathbf{u}_h^{[0]}) \right) \right]}. \tag{5}
$$

Intuitively, one might expect $\xi^{[t]} \geq 1$; the emulator, trained to mimic $\mathcal{P}_h$, should not be more accurate than $\mathcal{P}_h$ itself when compared to a better reference $\tilde{\mathcal{P}}_h$. However, this intuition overlooks crucial differences between the training process and the evaluation scenario. The emulator $f_\theta$ optimizes the specific loss function of equation 3 (often a one-step MSE) over a training distribution $\mathcal{I}_h$, while superiority is measured via multi-step rollouts using potentially different initial conditions $\tilde{\mathcal{I}}_h$. Furthermore, the function class of the emulator (e.g., a specific neural network architecture) imposes its own inductive biases. These differences create an opportunity for the emulator to find a parameterization $\theta$ that, while faithfully reproducing $\mathcal{P}_h$ in terms of the training objective, exhibits more favorable error accumulation properties during autoregressive rollouts or might generalize better to different regimes, leading to $\xi^{[t]} < 1$. We term this phenomenon *emulator superiority*.

This superiority is unlikely to hold universally ($\forall [t], \forall \tilde{\mathcal{I}}_h, \forall \tilde{\zeta}$), but its existence, even within specific regimes (e.g., for some time horizon, certain initial conditions or error metrics), challenges the notion that emulators are strictly bound by their training data's fidelity. Based on whether superiority is due to better state-space generalization or autoregressive generalization, we define **state-space superiority** or **autoregressive superiority**, respectively. Note that while state-space superiority is already observable in the first step, i.e., $\xi^{[1]} < 1$, autoregressive superiority requires at least two steps, i.e., $\xi^{[1]} \geq 1$ and $\xi^{[2]} < 1$.

## 3 A Case Study for Superiority under Linear Conditions

In the following subsections, we demonstrate that state-space superiority can be achieved in closed-form for three prototypical linear PDEs which are the basis for modelling many advanced phenomena, e.g., in fluid mechanics (Ferziger et al., 2020). Crucially, these derivations show how a simple linear emulator ansatz, by virtue of its own distinct error characteristics (stemming from its simple functional form, i.e., inductive bias) relative to higher and lower modes in Fourier space, can outperform the more complex (but numerically imperfect) low-fidelity solver it was trained on; already after the first step, i.e., $\xi^{[1]} < 1$.

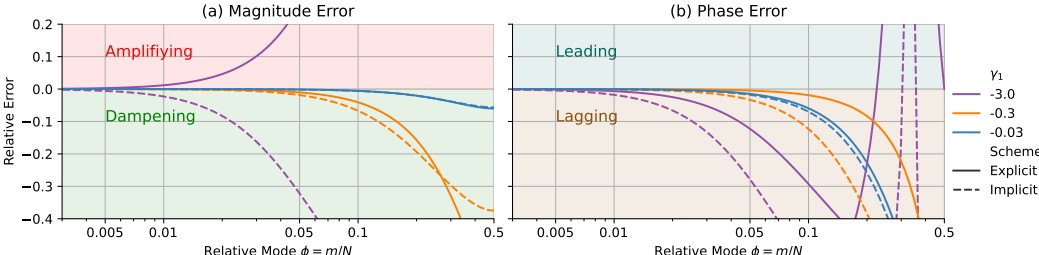

Figure 2: Both explicit $\hat{\mathbf{e}}_h$ and implicit $\hat{\imath}_h$ schemes introduce an error in the modes' magnitude either in dampening (reducing) or unstably amplifying (increasing) it. The former is visible as numerical diffusion in state space (a). This error is more pronounced for higher modes $\phi$ and larger $|\gamma_1|$. For low $\phi < 0.25$, the explicit scheme has less numerical diffusion. For higher $\phi$, the implicit scheme is better. The explicit scheme incurs a stability restriction $|\gamma_1| < 1$ to be stable for all modes. Similarly, the explicit scheme is better in terms of phase error within stability limits (b).

## 3.1 Advection Equation

The advection equation $\partial_t u + c\partial_x u = 0$ describes a simple linear and hyperbolic transport process, e.g., the movement of a concentration of a species $u(t, x)$ via the homogeneous velocity $c$. For simplicity, let us consider the one-dimensional spatial domain $\Omega = (0, 1)$ with periodic boundary conditions. We discretize the domain into $N$ intervals of equal length $\Delta x = 1/N$ and denote by $\mathbf{u}_h \in \mathbb{R}^N$ the vector of nodal values of $u$ at the left ends of each interval.

**Explicit Finite Difference Schemes in State Space**  The class of finite difference schemes on equidistant grids can be represented by cross-correlations[1] with a kernel $\mathbf{k}_h$, i.e., they advance the state in time via $\mathbf{u}_h^{[t+1]} = \mathbf{k}_h \star_\infty \mathbf{u}_h^{[t]}$, where $\star_\infty$ denotes the cross-correlation with periodic boundary conditions (i.e., circular "same" padding). For the advection equation, the simplest possible scheme is a first-order upwind method, which is given by the kernel

$$\mathbf{k}_h = [0 \quad 1 - \gamma_1 \quad \gamma_1] \quad \gamma_1 > 0, \qquad \mathbf{k}_h = [-\gamma_1 \quad 1 + \gamma_1 \quad 0] \quad \gamma_1 < 0, \tag{6}$$

where we used $\gamma_1 = -cN\Delta t \in \mathbb{R}$ to absorb all parameters into one variable. Based on the sign of $\gamma_1$, the scheme is either forward or backward differencing. The absolute value of this variable, i.e. $|\gamma_1|$, is also referred to as the Courant-Friedrichs-Lewy (CFL) number (LeVeque, 2007).

**Explicit Schemes via Fourier Multipliers**  Working under periodic boundary conditions has the merit that we can equally represent our state space as a vector of complex-valued Fourier coefficients $\hat{\mathbf{u}}_h = \mathcal{F}_h(\mathbf{u}_h) \in \mathbb{C}^{N/2+1}$, where we used the real-valued Fourier transform $\mathcal{F}_h$. In this case, the time-stepping can equivalently be done in Fourier space via $\hat{\mathbf{u}}_h^{[t+1]} = \mathcal{F}_h(\mathbf{k}_h) \odot \hat{\mathbf{u}}_h^{[t]}$ where $\odot$ denotes the element-wise product. Without loss of generality, we restrict ourselves to $\gamma_1 < 0$. Then, the Fourier coefficients of the explicit first-order upwind method are given by $\mathcal{F}_h(\mathbf{k}_h) = \hat{\mathbf{e}}_h = 1 + \gamma_1 - \gamma_1 e^{\circ \mathbf{w}_h}$ where $e^{\circ \mathbf{w}_h}$ denotes the element-wise exponential applied to the vector of scaled roots of unity $\mathbf{w}_h = -i\frac{2\pi}{N}[0, 1, \ldots, N/2]^T \in \mathbb{C}^{N/2+1}$, which are based on the imaginary unit $i^2 = -1$.

**Implicit and Analytical Schemes in Fourier Space**  In the Fourier domain, we can easily define two more numerical schemes: An implicit one that would require the solution to a linear system of equations in state space, and an analytical scheme. The implicit scheme is given by the Fourier multiplier $\hat{\imath}_h = 1 \oslash (1 - \gamma_1 + \gamma_1 e^{\circ \mathbf{w}_h})$ where $\oslash$ denotes the element-wise division. The analytical scheme is given by the Fourier multiplier $\hat{\alpha}_h = e^{\circ -\gamma_1 \mathbf{w}_h}$. More details on their derivation are presented in the appendix section A.1.1.

**Numerical Errors of the Schemes**  Since all numerical schemes, the explicit $\hat{\mathbf{e}}_h$, the implicit $\hat{\imath}_h$, and the analytical $\hat{\alpha}_h$ *diagonalize* in Fourier space, we can analyze their mode-wise errors individually. The analytical time stepper serves as the reference since it does not introduce any

---

[1]In a deep learning context, this is often loosely referred to as a convolution (LeCun et al., 1989).

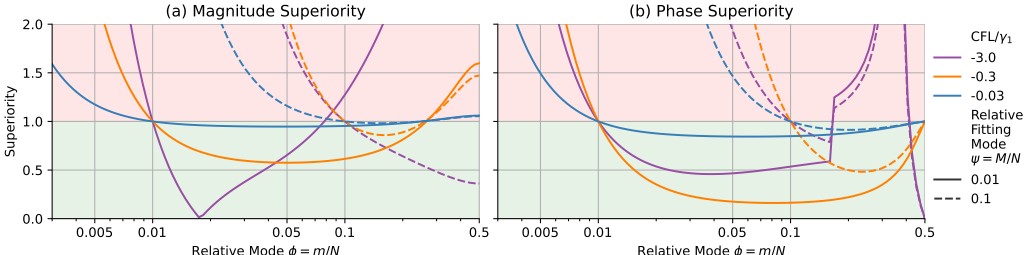

Figure 3: The superiority ratio for a simple two-parameter ansatz when trained on data generated by an implicit first-order upwind fitted at relative mode $\psi = M/N \in [0, 1/2]$ and tested across relative mode $\phi = m/N \in [0, 1/2]$. $M$ is the one mode at which we fit the emulator, $m$ is the one mode we test it at, and $N$ is the spatial resolution. A value below 1 indicates that the learned method is superior to its training simulator. Superiority occurs in both magnitude and phase error "forwardly" in that we are superior for a region with $\phi > \psi$.

numerical errors. To express the errors more conveniently, let us introduce the relative mode $\phi = m/N \in [0, 1/2]$. Then, we can analyze the magnitude error $|\hat{e}_h| - |\hat{\alpha}_h|$ and $|\hat{\imath}_h| - |\hat{\alpha}_h|$ and the phase error $(|\arg(\hat{e}_h)| - |\arg(\hat{\alpha}_h)|)/|\arg(\hat{\alpha}_h)|$ and $(|\arg(\hat{\imath}_h)| - |\arg(\hat{\alpha}_h)|)/|\arg(\hat{\alpha}_h)|$. These can be expressed in closed-form and are functions of $\phi$ and $\gamma_1$. We summarize an evaluation at representative values of $\phi$ and $\gamma_1$ in figure 2. The key insight is that these errors are not random perturbations but rather systematic deviations that follow well-defined patterns in Fourier space.

**Training a Linear Emulator**  Let us return to the setting of training an autoregressive emulator $f_\theta$ for the mapping $\mathbf{u}_h^{[t+1]} = f_\theta(\mathbf{u}_h^{[t]})$. Many architectures from image-to-image tasks in computer vision (Gupta and Brandstetter, 2023) or the more targeted field of neural operators (Kovachki et al., 2023) are a natural fit for this task. For simplicity, we first consider a simple linear cross-correlation/convolution, which equivalently diagonalizes in Fourier space

$$\mathbf{u}_h^{[t+1]} = [\theta_1 \quad \theta_0 \quad 0] \star_\infty \mathbf{u}_h^{[t]} \qquad \Longleftrightarrow \qquad \hat{\mathbf{u}}_h^{[t+1]} = \underbrace{(\theta_0 + \theta_1 e^{\circ \mathbf{w}_h})}_{=:\hat{\mathbf{q}}_h} \odot \hat{\mathbf{u}}_h^{[t]}. \tag{7}$$

If the train metric function is the mean squared error $\zeta(\mathbf{a}, \mathbf{b}) = \frac{1}{N}\|\mathbf{a} - \mathbf{b}\|_2^2$ the training objective of equation 3 diagonalizes. Let us further consider the scenario in which the initial condition distribution only produces states with solely mode $M$, i.e., $\boldsymbol{u}_h = c\sin_\circ(2\pi M \boldsymbol{x}_h - d)$ with some meaningful distributions over $c$ and $d$. In this case, the ansatz is only informed at mode $M$, i.e., we get

$$\arg\min_\theta (\hat{q}_M - \hat{r}_M) \iff \theta_0 + \theta_1 e^{-i2\pi \frac{M}{N}} \overset{!}{=} \hat{r}_M \tag{8}$$

with any of the aforementioned references $\hat{\boldsymbol{r}}_h \in \{\hat{\boldsymbol{e}}_h, \hat{\imath}_h, \hat{\alpha}_h\}$. For a proof, see section A.4.1. This is a complex-valued equation for the two real parameters $\theta_0, \theta_1 \in \mathbb{R}$ that can be solved in closed-form.

**Superiority of a Trained Linear Emulator**  Let us use the abbreviation $\psi = M/N \in [0, 1/2]$ for the relative mode at which we informed the ansatz. Then $\hat{q}_{\hat{\boldsymbol{r}}_\psi, h}$ shall denote the linear ansatz with the parameters found at training mode $\psi$ using any reference $\hat{\boldsymbol{r}}_h \in \{\hat{\boldsymbol{e}}_h, \hat{\imath}_h, \hat{\alpha}_h\}$. Again using the diagonalization properties and further assuming that the test distribution $\tilde{\mathcal{I}}_h$ also only produces states with one mode $m$ present (potentially different from the training mode $M$), we can express the superiority in magnitude simply as $\xi^{[t]} = \left| (|\hat{q}_{\hat{\boldsymbol{r}}_\psi, m}^t| - |\tilde{\tilde{r}}_m^t|)/(|\hat{r}_m^t| - |\tilde{\tilde{r}}_m^t|) \right|$. As such, the superiority ratio is a function of relative train mode $\psi$, relative test mode $\phi$, system properties $\gamma_1$ and the choice of training reference $\hat{\boldsymbol{r}}_h$ (that here also serves as the baseline) and testing reference $\tilde{\tilde{\boldsymbol{r}}}_h$. Using the implicit scheme for training and the analytic scheme for testing, we find the one-step superiority $\xi^{[1]}$ depicted in figure 3. When training on an implicit first-order upwind method, we can become better than this method for a different part of the Fourier space because we have an ansatz with a favorable generalization property.

Our theoretical analysis reveals several crucial conditions for achieving superiority. First, we must train a low-capacity linear ansatz on a high-capacity (non-analytical) reference method. Second,

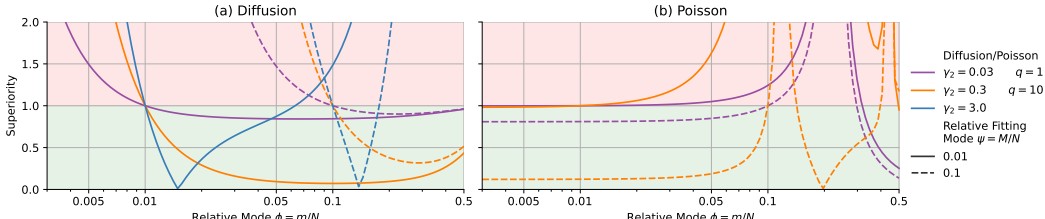

Figure 4: Superiority of an emulator fitted on relative mode $\psi = M/N \in [0, 1/2]$ for the diffusion (across different difficulties/intensities $\gamma_2$) and Poisson equation (across different numbers of iterations $q$ of the Richardson iteration). Similar to the advection equation (figure 3), diffusion displays superiority forwardly, i.e., $\phi > \psi$ whereas Poisson has backward superiority, i.e., $\phi < \psi$.

superiority emerges when testing at higher modes than those used during training, demonstrating a "forward" generalization property. Third, the difficulty parameter $\gamma_1$ must be sufficiently large in absolute value, where numerical errors are most pronounced. In the appendix, we extend the discussion to how the superiority develops over multiple test rollout steps (see A.1.6) and we discuss that the results equivalently hold for physics-informed training (see section A.4.2).

## 3.2 Diffusion Equation

Similar reasoning can be applied to the linear diffusion equation, $\partial_t u = \nu \nabla^2 u$, again under periodic boundaries on the unit interval where a Fourier analysis diagonalizes standard finite difference schemes. Similar to the advection equation, we learn an explicit functional form based on data from an implicit scheme. In figure 4a we present the results of the closed-form superiority expression. It is again a function of the fitting mode $\psi = M/N \in [0, 1/2]$, the testing mode $\phi = m/N \in [0, 1/2]$, and a new combined parameter $\gamma_2 = 2\nu \Delta t N^2 > 0$.

The diffusion equation demonstrates a similar superiority pattern to that of advection. Specifically, superiority emerges in regions where $\phi > \psi$, indicating that emulators generalize effectively to higher modes than those used during training. The magnitude of this superiority effect increases with higher values of the difficulty parameter $\gamma_2$. Fundamentally, this reinforces our core finding: when an emulator $f_\theta$ with favorable inductive biases (in this case, a structure based on simple cross-correlation/convolution mimicking a forward-in-time central-in-space scheme) is trained on data from a numerically imperfect solver $\mathcal{P}_h$ (here, a backward-in-time central-in-space scheme), it can develop representations that outperform its training data when evaluated against a high-fidelity analytical solution $\tilde{\mathcal{P}}_h$, i.e., it is *state-space superior*, here already after the first time step.

## 3.3 Poisson Equation

The superiority principle can also be observed when solving stationary elliptic problems like the Poisson equation, $-\nabla^2 u = f$, again under periodic boundary conditions. For this, we consider the low-fidelity scheme an unconverged iterative solver. For simplicity, we here use a Richardson iteration. Such solvers act as smoothers: they efficiently damp high-frequency errors (modes $\phi = m/N$ near $1/2$) but converge slowly for low-frequency errors ($\phi$ near 0). Let the resulting Fourier multiplier after $q$ iterative steps (mapping source $\hat{f}_m$ to approximate/truncated solution $\hat{u}_m$) be $\hat{\iota}_m(q)$.

If we train an ansatz with a second-order accurate discretization of the second derivative in Fourier space, but with a free parameter $\theta$, this trained ansatz can perform superior to the low-fidelity solver when evaluated against a fully analytic solution to the Poisson equation. This superiority can again be derived in closed-form and is also a function of the fitting mode $\psi = M/N \in [0, 1/2]$ and the testing mode $\phi = m/N \in [0, 1/2]$. Here, it also depends on the number of iterations $q$ of the solver. We display configurations in figure 4b. Crucially, in this setting we see *backward superiority*: the emulator outperforms the iterative baseline largely for $\phi < \psi$. This contrasts with the forward superiority patterns seen earlier and lets us conclude that a *state-space superiority* is possible in the direction where the reference produces more errors and the ansatz provides a favorable inductive bias.

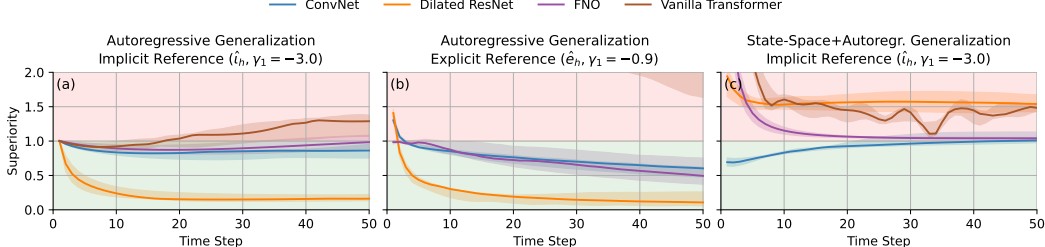

Figure 5: Almost all considered emulator architectures achieve autoregressive superiority when the IC distribution does not change from training to test (a) & (b). Hence, superiority can also be achieved against the better explicit scheme within its stability limit (b). State-space superiority (under varied IC distribution) is only possible with the favorable local inductive bias of a ConvNet (c).

## 4    Extension to Nonlinear Settings

While our analysis in section 3 rigorously established the principle of state-space superiority for linear emulators under specific conditions, practical scientific machine learning often involves complex, nonlinear neural network emulators. We now shift our focus to these settings, investigating whether the core insights—namely, that an emulator's inductive biases and training process leads to dynamics that are implicitly more regularized or exhibit more stable error propagation—extend to nonlinear architectures and complex PDE systems. This transition also allows to explore the emergence of autoregressive superiority, a phenomenon less readily observed with the linear ansatz of before.

### 4.1    Nonlinear Emulators for the Linear Advection Equation

Building on our linear analysis of state-space superiority, we now investigate if nonlinear emulators—specifically ConvNets (Fukushima, 1980; LeCun et al., 1989), Dilated ResNets (He et al., 2016; Yu and Koltun, 2016; Stachenfeld et al., 2021), Fourier Neural Operators (FNOs) (Li et al., 2021a), and Transformers (Vaswani et al., 2017)—can also achieve superiority for the linear advection equation, potentially through different mechanisms, with key findings illustrated in figure 5.

Our experiments focus on two primary settings designed to probe different aspects of superiority. First, to assess **autoregressive superiority**, we train and test emulators using identical initial condition (IC) distributions. This is examined for emulators trained on data from an implicit scheme ($\hat{\imath}_h$ at $\gamma_1 = -3.0$, figure 5a) and an explicit first-order upwind scheme operating within its stability limit ($\gamma_1 = -0.9$, figure 5b). This setup tests if emulators can achieve more stable or accurate multi-step rollouts than their training solver. We observe clear autoregressive superiority. Nearly all tested architectures achieve a superiority ratio $\xi^{[t]} < 1$ after some initial time steps. As expected, there cannot be superiority on the first time step ($\xi^{[1]} \geq 1$) because the emulators are trained to mimic the one-step behavior of the training solver on these specific ICs. Remarkably, this superiority manifests even when training on data from the relatively accurate explicit scheme (figure 5b) for which we cannot prove superiority in the linear case (see A.1.5). This indicates that the nonlinear emulators learn dynamics that accumulate errors more favorably during rollouts compared to the solver they were trained on, a phenomenon not readily achievable with simple linear emulators.

Second, to evaluate **state-space superiority**, we train emulators on data from the implicit scheme ($\hat{\imath}_h$ at $\gamma_1 = -3.0$, figure 5c) and then test their generalization to ICs containing a broader range of modes, thereby probing generalization to unseen physical states. We find that only the local feed-forward ConvNet is able to generalize to the same extent as the linear case. In contrast, architectures designed with global receptive fields, such as FNOs and Transformers, show limited ability to generalize in this particular task of extrapolating from single-mode training to multi-mode testing.

### 4.2    Nonlinear Emulators for the Burgers equation

We now extend our investigation to a scenario where both the PDE and the reference solvers are nonlinear, using the Burgers equation as a test case. The data is generated by an implicit first-order upwind scheme. The key difference between our low-fidelity training solver and the high-fidelity

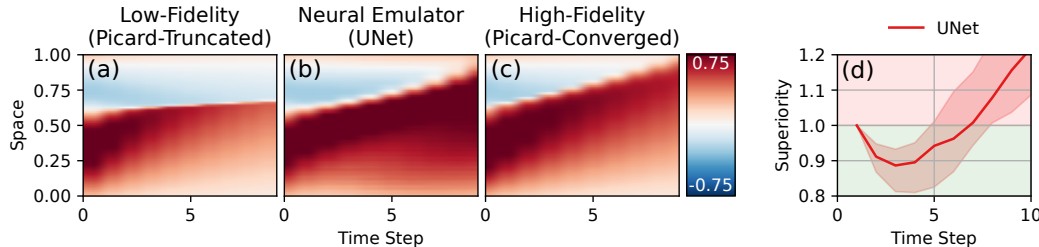

Figure 6: When an autoregressive neural emulator is trained on the first two snapshots of a simulator that truncates the nonlinear solve of an implicit Burgers integrator (a), it learns a better representation of the shock propagation phase than the simulator (b), thereby more closely matching a simulator which is nonlinearly converged (c). As such, it becomes superior over its training simulator for a certain number of time steps (d).

reference lies in the treatment of the nonlinearity: the low-fidelity solver employs a Picard iteration that is truncated (i.e., not fully converged), while the high-fidelity reference uses a converged nonlinear solve (the schemes are derived in appendix section B.2.1).

Not fully resolving the nonlinearity results in an incorrectly propagated shock (i.e., a wrong shock angle) as depicted in figure 6a and figure 6c. The Burgers equation, courtesy of being a nonlinear PDE, has the property that it develops a rich spectrum over the trajectory. The richer the spectrum, i.e., once we are in the shock formation and shock propagation phase, the more errors a nonlinearly truncated scheme makes (see figure 13 for more details). Hence, for this experiment, we train a vanilla UNet (Ronneberger et al., 2015) on only the initial "smooth" frames from this low-fidelity solver. Despite training only on these smooth initial states, the UNet, when rolled out, learns to propagate shocks more accurately than its training data, developing a shock angle that closely matches the high-fidelity reference (figure 6b,d) and thus demonstrating superiority. This superiority is hypothesized to stem from the UNet implicitly correcting the truncated nonlinear solve and its convolutional architecture generalizing effectively in Fourier space: training on smooth, low-frequency data enables it to better handle the emergent high-frequency shocks, mirroring mechanisms from section 3.

## 5 Limitations & Outlook

The phenomenon of emulator superiority, while intriguing, is primarily observed within specific operating regimes rather than as a universal guarantee. This characteristic, however, aligns with the needs of many scientific and engineering applications, such as Model Predictive Control (MPC) (Rawlings et al., 2017), where systems often operate within well-defined boundaries, over receding horizons, or near particular trajectories. In such scenarios, an emulator's accuracy-cost proposition, even if confined to a specific regime, offers significant practical value. Future work should focus on precisely identifying these operational boundaries for superiority and understanding their dependence on the interplay between the PDE, solver characteristics, and emulator architecture.

Furthermore, the intensity of the superiority effect appears linked to the low-fidelity training solver being *sufficiently inaccurate* relative to the high-fidelity reference; a highly precise training solver naturally reduces the margin for emulator improvement. While our examples often use solvers with clear error differentials to illustrate the principle, real-world scenarios frequently involve suboptimal or computationally constrained solvers (Turek, 1999; Müller et al., 2007). Thus, an important research avenue is to investigate the sensitivity of superiority to varying degrees of training data fidelity and to explore whether emulators can still offer advantages when trained on moderately accurate data.

There is an interesting relationship of emulator superiority and adversarial examples in image classification (Goodfellow et al., 2015). One could use equation 5 and an optimization over it to find inputs $\mathbf{u}_h^{[0]}$ (or corresponding distributions $\tilde{\mathcal{I}}_h$), metrics $\tilde{\zeta}$, or time horizons $[t]$ that give rise to superiority. We believe this could be a valuable tool to further understand this intriguing phenomenon.

Beyond these points, a broader agenda includes deeper investigation into how different training strategies (e.g., unrolled training, physics-informed losses) and diverse architectural choices influence superiority. Extending this analysis to more complex, multi-physics problems will also be

crucial. Ultimately, understanding and strategically harnessing emulator superiority could lead to the development of more efficient and physically realistic emulation tools.

# 6 Related Work

The potential for computational speedups and tackling unsolved problems has motivated the use of neural networks (NNs) for helping solve PDEs. Physics-Informed Neural Networks (PINNs) approximate the continuous solution function $u(t, \mathbf{x})$ directly (Lagaris et al., 1998; Raissi et al., 2019). Machine learning models can also be integrated into traditional numerical methods to replace heuristic components, such as turbulence models (Duraisamy et al., 2019) or flux limiters (Discacciati et al., 2020). On a different abstraction level, NNs can be used to correct coarse solver outputs (Um et al., 2020; Kochkov et al., 2021). A machine learning model can also be used to fully replace the numerical solver, for time-dependent problems typically in an autoregressive manner (Brandstetter et al., 2022; Lippe et al., 2023). The Neural Operator subfield poses stricter constraints on the architecture (Kovachki et al., 2023). A recent trend is to design emulators capable of solving multiple PDEs zero-shot or as foundation models (McCabe et al., 2024; Herde et al., 2024; Holzschuh et al., 2025).

Oftentimes, the neural architectures contain inductive biases showing similarities to numerical solvers (Alt et al., 2023; McCabe et al., 2023), albeit them not being strictly bound to their numerical constraints (Koehler et al., 2024). This indicates why we can use them as compute-efficient surrogates. While McGreivy and Hakim (2024) emphasize that some speedups might not hold to the strictest mathematical scrutiny, from a more practical perspective, there have been extensive positive results for neural approaches, e.g., in weather prediction (Kochkov et al., 2024).

Training these networks is oftentimes based on synthetic data produced by existing numerical schemes. Mohan et al. (2024) find that training on numerical simulation data containing structural errors can lead neural models to learn these artifacts, potentially limiting their generalization despite appearing to learn. In the domain of particle physics, generative models have demonstrated "amplification," where high-fidelity distributions are recovered from limited or noisy training data (Butter et al., 2021). To the best of our knowledge, our work is the first to identify how aspects of the emulator training pipeline can lead to networks outperforming data with significant (deterministic) structural errors.

# 7 Conclusion & Impact

We introduced "emulator superiority," where neural emulators trained on low-fidelity solver data outperform those solvers against a high-fidelity reference in specific regimes. Our analysis reveals this stems from the interplay of emulator inductive biases, training objectives, solver error characteristics, and multi-step rollout dynamics, allowing emulators to implicitly correct structured errors. This work invites a reconsideration of the relationship between neural emulators and their training data, suggesting that the emulators' architectural inductive biases and learning dynamics can, under specific conditions, enable them to transcend the accuracy limitations of their training simulators.

Our findings critically highlight the importance of high-quality testing references in emulator evaluation, as flawed references with numerical artifacts can obscure true performance, potentially penalizing emulators that achieve greater physical fidelity. This paradox challenges conventional benchmarking, urging careful interpretation of metrics considering the reference's own limitations. Consequently, developing robust evaluation frameworks that account for superiority effects is essential for accurately assessing neural emulators in scientific computing. Embracing these insights can therefore foster the development of more powerful and reliable emulation tools, opening new avenues for scientific discovery and engineering innovation.

# 8 Acknowledgements

The authors thank Bjoern List, Patrick Schnell, Rene Winchenbach, Michael McCabe, and our anonymous reviewers for fruitful discussions. Felix Koehler acknowledges funding from the Munich Center for Machine Learning (MCML).

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

# A More Details on the theoretical considerations

The derivations presented herein are supported by symbolic computations performed with SymPy (Meurer et al., 2017). To facilitate reproducibility, Jupyter notebooks are available in our supplemental material: github.com/tum-pbs/emulator-superiority. This section aims to provide a more comprehensive exposition of the mathematical foundations underlying the numerical schemes and the concept of emulator superiority discussed in the main text.

## A.1 Advection Equation

We begin by considering the one-dimensional linear advection equation on a periodic domain of unit length ($L = 1$):

$$\partial_t u + c\partial_x u = 0, \quad u(0, t) = u(1, t), \tag{9}$$

where $c \in \mathbb{R}$ is the constant advection speed. To simplify the analysis by reducing the number of free parameters, we adopt the following reparametrized form

$$\partial_t u = \frac{\gamma_1}{N\Delta t}\partial_x u, \tag{10}$$

where

$$\gamma_1 = -cN\Delta t \in \mathbb{R} \tag{11}$$

is a combined variable to absorb all experimental parameters. Its absolute, i.e., $|\gamma_1|$, is also referred to as the Courant-Friedrichs-Lewy (CFL) number. The number of discretization points $N$ and the time step size $\Delta t$ will cancel out in the following analysis. As discussed in the main text in section 3.1, we choose an equidistant discretization of the spatial domain $\Omega$ into $N$ intervals of equal length $\Delta x = 1/N$. Then, we consider the left endpoint of each interval a nodal degree of freedom. Collectively, we can write the vector of grid coordinates as $\mathbf{x}_h = [0, \Delta x, 2\Delta x, \dots, (N-1)\Delta x]^T \in \mathbb{R}^N$. The left boundary point is part of our discretization, but the right boundary point is not. Let $\mathbf{u}_h \in \mathbb{R}^N$ be the vector of nodal values of $u$ at the coordinates $\mathbf{x}_h$. An upper index in square brackets is used to denote time index, i.e., $\mathbf{u}_h^{[0]}$ would be the initial state, $\mathbf{u}_h^{[1]}$ is the state one time step later, etc.

### A.1.1 Extended Derivations for the Advection Equation Schemes

The simplest possible discretization in space is based on a first-order approximation to the first spatial derivative. To ensure stability of the scheme, we have to choose between forward and backward differencing based on the direction of the advection speed $c$, i.e., based on the sign of $\gamma_1$. The stable configuration would be a forward difference if $c$ is negative and a backward difference if $c$ is positive. Hence, since $\gamma_1$ reverses the sign, as defined in equation 11, we have to use a forward difference if $\gamma_1 > 0$ and a backward difference if $\gamma_1 < 0$. As such, we get the following two possible approximations to the advection equation

$$\frac{d}{dt}u_i = \frac{\gamma_1}{N\Delta t} \begin{cases} \frac{1}{\Delta x}(u_{i+1} - u_i) & \text{if } \gamma_1 > 0 \\ \frac{1}{\Delta x}(u_i - u_{i-1}) & \text{if } \gamma_1 < 0 \end{cases} \tag{12}$$

Here we used $u_i$ to index the nodal degrees of freedom at spatial position $x_i = i\Delta x$. Since $\Delta x = L/N$ and $L = 1$, we see that $1/\Delta x$ and $1/N$ cancel, leaving

$$\frac{d}{dt}u_i = \frac{\gamma_1}{\Delta t} \begin{cases} u_{i+1} - u_i & \text{if } \gamma_1 > 0 \\ u_i - u_{i-1} & \text{if } \gamma_1 < 0 \end{cases} \tag{13}$$

Further, we can use an Euler approximation to discretize in time. If we evaluate the right-hand side at the previous point in time, we obtain an explicit first-order upwind method.

$$\frac{u_i^{[t+1]} - u_i^{[t]}}{\Delta t} = \frac{\gamma_1}{\Delta t} \begin{cases} u_{i+1}^{[t]} - u_i^{[t]} & \text{if } \gamma_1 > 0 \\ u_i^{[t]} - u_{i-1}^{[t]} & \text{if } \gamma_1 < 0 \end{cases} \tag{14}$$

where we can easily re-arrange for the next state in time and see that $\Delta t$ cancels giving

$$u_i^{[t+1]} = u_i^{[t]} + \gamma_1 \begin{cases} u_{i+1}^{[t]} - u_i^{[t]} & \text{if } \gamma_1 > 0 \\ u_i^{[t]} - u_{i-1}^{[t]} & \text{if } \gamma_1 < 0 \end{cases} \tag{15}$$

We see that the discrete solution is solely parameterized by combined variable $\gamma_1$. Without loss of generality, let us assume that $\gamma_1 < 0$. Then, we can write the update on the entire state vector as

$$\underbrace{\begin{bmatrix} u_0^{[t+1]} \\ u_1^{[t+1]} \\ u_2^{[t+1]} \\ \vdots \\ u_{N-1}^{[t+1]} \end{bmatrix}}_{=\mathbf{u}_h^{[t+1]}} = \underbrace{\begin{bmatrix} 1+\gamma_1 & 0 & 0 & \cdots & 0 & -\gamma_1 \\ -\gamma_1 & 1+\gamma_1 & 0 & \cdots & 0 & 0 \\ 0 & -\gamma_1 & 1+\gamma_1 & \cdots & 0 & 0 \\ \vdots & \vdots & \vdots & \ddots & \vdots & \vdots \\ 0 & 0 & -\gamma_1 & \cdots & 1+\gamma_1 & 0 \\ 0 & 0 & 0 & \cdots & -\gamma_1 & 1+\gamma_1 \end{bmatrix}}_{=\mathbf{A}} \underbrace{\begin{bmatrix} u_0^{[t]} \\ u_1^{[t]} \\ u_2^{[t]} \\ \vdots \\ u_{N-1}^{[t]} \end{bmatrix}}_{=\mathbf{u}_h^{[t]}} \tag{16}$$

Clearly, the matrix $\mathbf{A}$ is a circulant matrix. Hence, we could equivalently express the update rule via a cross-correlation on periodic boundaries as

$$\mathbf{u}_h^{[t+1]} = \begin{bmatrix} -\gamma_1 & 1+\gamma_1 & 0 \end{bmatrix} \star_\infty \mathbf{u}_h^{[t]} \tag{17}$$

where $\star_\infty$ denotes the periodic cross-correlation. It is known that circulant matrices diagonalize using the discrete Fourier transform (DFT) matrix

$$\mathbf{F} = \begin{bmatrix} 1 & 1 & 1 & \cdots & 1 \\ 1 & \omega & \omega^2 & \cdots & \omega^{N-1} \\ 1 & \omega^2 & \omega^4 & \cdots & \omega^{2(N-1)} \\ \vdots & \vdots & \vdots & \ddots & \vdots \\ 1 & \omega^{N-1} & \omega^{2(N-1)} & \cdots & \omega^{(N-1)(N-1)} \end{bmatrix} \tag{18}$$

where $\omega = e^{-i2\pi/N}$ is the $N$-th primitive root of unity. For our explicit upwind scheme, this gives us the Fourier multiplier

$$\hat{\mathbf{e}}_h = 1 + \gamma_1 - \gamma_1 e^{\circ \mathbf{w}_h} \tag{19}$$

where $e^{\circ \mathbf{w}_h}$ denotes the element-wise exponential applied to the vector of scaled roots of unity $\mathbf{w}_h = -i\frac{2\pi}{N}[0, 1, \ldots, N/2+1]^T \in \mathbb{C}^{N/2+1}$. These allow us to advance the state in Fourier space via

$$\hat{\mathbf{u}}_h^{[t+1]} = \hat{\mathbf{e}}_h \odot \hat{\mathbf{u}}_h^{[t]}, \tag{20}$$

where $\odot$ denotes the element-wise product.

**Implicit First-Order Upwind Scheme**   If instead of evaluating the right-hand side at the previous point in time, we evaluate it at the next point in time, we obtain an implicit scheme. Assuming again $\gamma_1 < 0$, in state space it would read

$$\begin{bmatrix} \gamma_1 & 1-\gamma_1 & 0 \end{bmatrix} \star_\infty \mathbf{u}_h^{[t+1]} = \mathbf{u}_h^{[t]}. \tag{21}$$

Hence, in order to advance in time in state space, we would have to solve a linear system. Gladly, the diagonalization property of the DFT matrix allows us solve the system by mode-wise division in Fourier space. This gives the Fourier multiplier

$$\hat{\imath}_h = 1 \oslash \left(1 - \gamma_1 + \gamma_1 e^{\circ \mathbf{w}_h}\right) \tag{22}$$

where $\oslash$ denotes the element-wise division.

**Analytical Scheme in Fourier Space**   The explicit and implicit first-order upwind schemes incur an error due to the approximation of space and time. Both can be avoided by using an exact solution in Fourier space which is possible for all linear PDEs under periodic boundaries with bandlimited initial conditions. First, we use the exact first derivative in Fourier space given by

$$\mathcal{F}_h(\partial_x \mathbf{u}_h) = \underbrace{i\frac{2\pi}{L}\begin{bmatrix} 0 & 1 & 2 & \cdots & N/2+1 \end{bmatrix}}_{=\hat{\mathbf{d}}_h} \odot \underbrace{\mathbf{F}_h(\mathbf{u}_h)}_{=\hat{\mathbf{u}}_h} \tag{23}$$

where $\hat{\mathbf{d}}_h \in \mathbb{C}^{N/2+1}$ are the mode-wise complex-valued Fourier multipliers with which the state in Fourier space $\hat{\mathbf{u}}_h$ is element-wise multiplied to find the Fourier representation of the first derivative. Hence, in Fourier space, the semi-discrete advection equation reads

$$\frac{d}{dt}\hat{\mathbf{u}}_h = \frac{\gamma_1}{N\Delta t}\hat{\mathbf{d}}_h \odot \hat{\mathbf{u}}_h \tag{24}$$

Working in Fourier space lets us diagonalize the system. This makes the ODE much easier to solve. We can use the matrix exponential, which is simple to calculate when matrices are diagonal. This gives us an analytical time stepper in Fourier space.

$$\hat{\mathbf{u}}_h^{[t+1]} = \underbrace{e^{\circ \hat{\mathbf{d}}_h \frac{\gamma_1}{N \Delta t} \Delta t}}_{\hat{\alpha}_h} \odot \hat{\mathbf{u}}_h^{[t]} \tag{25}$$

where $e^\circ$ denotes the element-wise exponential. Let us further reformulate the Fourier multiplier $\hat{\alpha}_h$ as

$$\hat{\alpha}_h = \exp_\circ(\hat{\mathbf{d}}_h \frac{\gamma_1}{N \Delta t} \Delta t) \tag{26}$$

$$= \exp_\circ(\underbrace{i \frac{2\pi}{N} \begin{bmatrix} 0 & 1 & 2 & \cdots & N/2+1 \end{bmatrix} \gamma_1}_{-\hat{\mathbf{w}}_h}) \tag{27}$$

$$= \exp_\circ(-\gamma_1 \hat{\mathbf{w}}_h) \tag{28}$$

where we used $L = 1$.

### A.1.2 Error of the Schemes

The diagonalization in Fourier space allows to express the mode-wise error done by each scheme in terms of the error of the Fourier multiplier. For this, we use the following representations of the three schemes (again assuming $\gamma_1 < 0$) for a relative mode $\phi = m/N \in [0, 1/2]$

$$\hat{e}_\phi = 1 + \gamma_1 - \gamma_1 e^{-i2\pi\phi}, \tag{29}$$

$$\hat{\iota}_\phi = \frac{1}{1 - \gamma_1 + \gamma_1 e^{-i2\pi\phi}}, \tag{30}$$

$$\hat{\alpha}_\phi = e^{-i\gamma_1 2\pi\phi}. \tag{31}$$

Since all multipliers are complex-valued, we can compare the errors against the analytical scheme both in terms of magnitude and phase. The magnitude of the analytical reference is always $|\alpha_\phi| = 1$ which is to be expected since the pure hyperbolic transport should not alter the magnitude of the solution, i.e., there should be artificial diffusion.

Then, we get first the magnitude error of the explicit scheme

$$\frac{|\hat{e}_\phi| - |\hat{\alpha}_\phi|}{|\hat{\alpha}_\phi|} = \cos\left(\frac{1}{2} \cdot \mathrm{atan}_2\left(0, 4 \cdot \gamma_1^2 \cdot \sin^2(\pi \cdot \phi) + 4 \cdot \gamma_1 \cdot \sin^2(\pi \cdot \phi) + 1\right)\right) \tag{32}$$
$$\cdot \sqrt{\left|4 \cdot \gamma_1^2 \cdot \sin^2(\pi \cdot \phi) + 4 \cdot \gamma_1 \cdot \sin^2(\pi \cdot \phi) + 1\right|} - 1,$$

and the magnitude error of the implicit scheme

$$\frac{|\hat{\iota}_\phi| - |\hat{\alpha}_\phi|}{|\hat{\alpha}_\phi|} = -1 + \frac{1}{\sqrt{4 \cdot \gamma_1^2 \cdot \sin^2(\pi \cdot \phi) - 4 \cdot \gamma_1 \cdot \sin^2(\pi \cdot \phi) + 1}}. \tag{33}$$

Moreover, we have the phase error of the explicit scheme

$$\frac{|\arg(\hat{e}_\phi)| - |\arg(\hat{\alpha}_\phi)|}{|\arg(\hat{\alpha}_\phi)|} = \left|\frac{\arg\left(\gamma_1 - \gamma_1 \cdot e^{-2 \cdot i \cdot \pi \cdot \phi} + 1\right)}{\arg\left(e^{2 \cdot i \cdot \pi \cdot \gamma_1 \cdot \phi}\right)}\right| - 1, \tag{34}$$

and the phase error of the implicit scheme

$$\frac{|\arg(\hat{\iota}_\phi)| - |\arg(\hat{\alpha}_\phi)|}{|\arg(\hat{\alpha}_\phi)|} = \left|\frac{\arg\left(\frac{e^{2 \cdot i \cdot \pi \cdot \phi}}{-\gamma_1 \cdot e^{2 \cdot i \cdot \pi \cdot \phi} + \gamma_1 + e^{2 \cdot i \cdot \pi \cdot \phi}}\right)}{\arg\left(e^{2 \cdot i \cdot \pi \cdot \gamma_1 \cdot \phi}\right)}\right| - 1. \tag{35}$$

Those expressions have been plotted in figure 2 in the main text for representative choices of $\phi$ and $\gamma_1$.

### A.1.3 Closed-Form Emulator trained on the Implicit Scheme

As discussed in the main text, we will inform an emulator $f_\theta$ at one relative mode $\psi = M/N$. The functional form of the ansatz will be similar to the explicit scheme and is given by

$$\boldsymbol{u}_h^{[t+1]} = [\theta_1 \quad \theta_0 \quad 0] \star_\infty \boldsymbol{u}_h^{[t]} \tag{36}$$

in the state space, i.e., it has the same upwinding bias as the explicit scheme and hence should only be used for the assumed $\gamma_1 < 0$. We find its Fourier multiplier as

$$\hat{\mathbf{q}}_h = \theta_0 + \theta_1 e^{\circ \mathbf{w}_h}. \tag{37}$$

When fitting against the implicit scheme at relative mode $\hat{\iota}_\psi$, we find the following solutions for the complex-valued equation

$$\theta_0 = -\frac{\gamma_1}{4 \cdot \gamma_1^2 \cdot \sin^2(\pi \cdot \psi) - 4 \cdot \gamma_1 \cdot \sin^2(\pi \cdot \psi) + 1}, \tag{38}$$

$$\theta_1 = \frac{-4 \cdot \gamma_1 \cdot \sin^2(\pi \cdot \psi) + \gamma_1 + 1}{4 \cdot \gamma_1^2 \cdot \sin^2(\pi \cdot \psi) - 4 \cdot \gamma_1 \cdot \sin^2(\pi \cdot \psi) + 1}. \tag{39}$$

This can be plugged back into the ansatz to get the emulator

$$\hat{h}_{\hat{\iota}_\psi, \phi} = \frac{\left(-\gamma_1 + \left(-4 \cdot \gamma_1 \cdot \sin^2(\pi \cdot \psi) + \gamma_1 + 1\right) \cdot e^{2 \cdot i \cdot \pi \cdot \phi}\right) \cdot e^{-2 \cdot i \cdot \pi \cdot \phi}}{4 \cdot \gamma_1^2 \cdot \sin^2(\pi \cdot \psi) - 4 \cdot \gamma_1 \cdot \sin^2(\pi \cdot \psi) + 1}. \tag{40}$$

### A.1.4 Closed-Form Superiority when using the Implicit Scheme as Baseline

We can use the closed-form expression for the found emulator $\hat{h}_{\hat{\iota}_\psi, \phi}$ to formulate the closed-form superiority. For this, we choose the analytical scheme $\hat{\alpha}_\phi$ as the reference and use the training simulator (the implicit scheme, $\iota_\phi$) as the baseline.

This gives in terms of phase

$$\left| \frac{\arg(\hat{q}_{\hat{\iota}_\psi, \phi}) - \arg(\hat{\alpha}_\phi)}{\arg(\hat{\iota}_\phi) - \arg(\hat{\alpha}_\phi)} \right|$$
$$= \frac{\arg\left(\left(-\gamma_1 + \left(-4 \cdot \gamma_1 \cdot \sin^2(\pi \cdot \psi) + \gamma_1 + 1\right) \cdot e^{2 \cdot i \cdot \pi \cdot \phi}\right) \cdot e^{-2 \cdot i \cdot \pi \cdot \phi}\right) - \arg\left(e^{2 \cdot i \cdot \pi \cdot \gamma_1 \cdot \phi}\right)}{\arg\left(\frac{e^{2 \cdot i \cdot \pi \cdot \phi}}{-\gamma_1 \cdot e^{2 \cdot i \cdot \pi \cdot \phi} + \gamma_1 + e^{2 \cdot i \cdot \pi \cdot \phi}}\right) - \arg\left(e^{2 \cdot i \cdot \pi \cdot \gamma_1 \cdot \phi}\right)}. \tag{41}$$

This equation is given here to show that there are indeed closed-form expressions. Unfortunately, most of them did not amend themselves to easy simplification which is why we present further lengthy expressions in the supplemental material in the form of a HTML table: https://tum-pbs.github.io/emulator-superiority/symbolic-expressions.

### A.1.5 More Combinations of References

The main text only considered the case of the following combination of solvers:

- Training on the implicit scheme
- Baseline on the implicit scheme
- Testing on the analytical scheme

While testing on the analytical scheme is reasonable since it allows to exactly assess the full error conducted, changing up the training and baseline schemes allows study additional effects.

When we fit on the explicit scheme, we get the following solutions for the complex-valued equation

$$\theta_0 = -\gamma_1, \tag{42}$$

$$\theta_1 = \gamma_1 + 1. \tag{43}$$

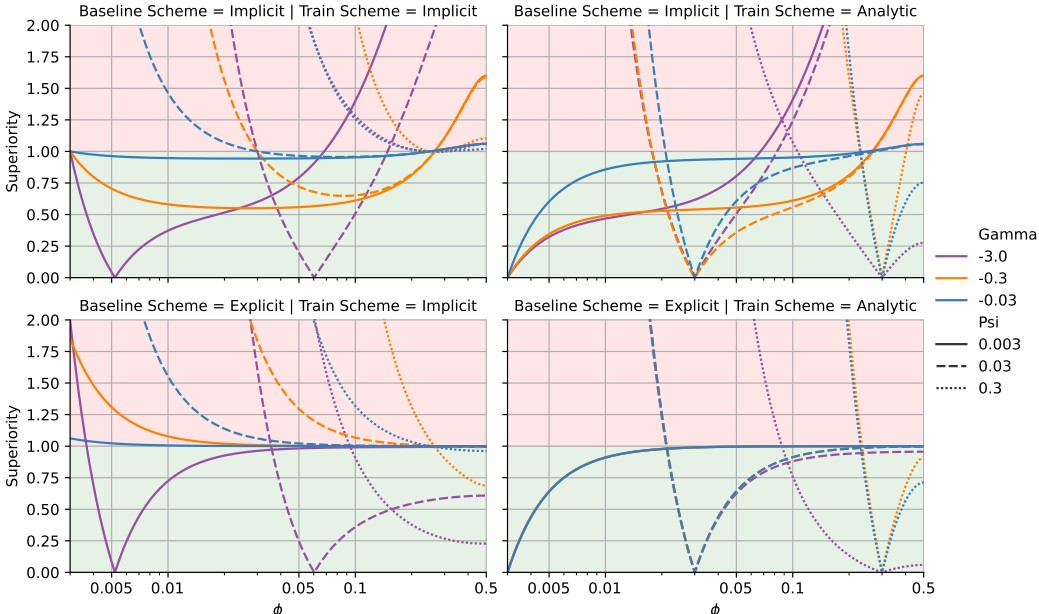

Figure 7: All possible combinations of training references and baseline references for the advection superiority when measured in terms of the magnitude error against the analytical scheme. The first column represents training on the implicit scheme, the second column displays training on the analytical scheme. The first row displays using the implicit scheme as the baseline reference, and the second row uses the explicit scheme as the baseline. The superiority figure in the top left is identical to figure 3 presented in the main text in which a favorable inductive bias leads to becoming better than the reference. In the bottom left we see that if we use the same training reference but baseline against the stronger explicit scheme, almost no superiority is achievable under the stability limit, except for $\phi \geq 0.25$ in which the implicit scheme was also better than the explicit scheme as shown in figure 2. However, surprisingly, our explicit ansatz becomes superior beyond the stability limit $|\gamma_1| > 1$ for regions of $\phi$ around the trained $\psi$. The second column highlights that superiority is even stronger if we use the analytical scheme as the training reference.

As expected, since the functional form of the ansatz is exactly the same as the training scheme, we fully "re-learn" the training scheme. As such, there can also be no superiority in terms of magnitude or phase, hence we have $\xi^{[t]} = 1$. As such, we also will not show any figures for this case.

This leaves us with the case of fitting against the analytical scheme which gives us the following solutions for the complex-valued equation

$$\theta_0 = -\frac{\sin\left(2 \cdot \pi \cdot \gamma_1 \cdot \psi\right)}{\sin\left(2 \cdot \pi \cdot \psi\right)}, \tag{44}$$

$$\theta_1 = \frac{\sin\left(2 \cdot \pi \cdot \psi \cdot (\gamma_1 + 1)\right)}{\sin\left(2 \cdot \pi \cdot \psi\right)}. \tag{45}$$

This yields the fitted emulator

$$\hat{h}_{\hat{\alpha}_\psi, \phi} = \frac{\left(e^{2 \cdot i \cdot \pi \cdot \phi} \cdot \sin\left(2 \cdot \pi \cdot \psi \cdot (\gamma_1 + 1)\right) - \sin\left(2 \cdot \pi \cdot \gamma_1 \cdot \psi\right)\right) \cdot e^{-2 \cdot i \cdot \pi \cdot \phi}}{\sin\left(2 \cdot \pi \cdot \psi\right)}. \tag{46}$$

Using either the implicit or the analytical scheme as the training reference, either the explicit or the implicit scheme as the baseline, and the analytical scheme as the testing reference, we discuss four potential combinations in figure 7.

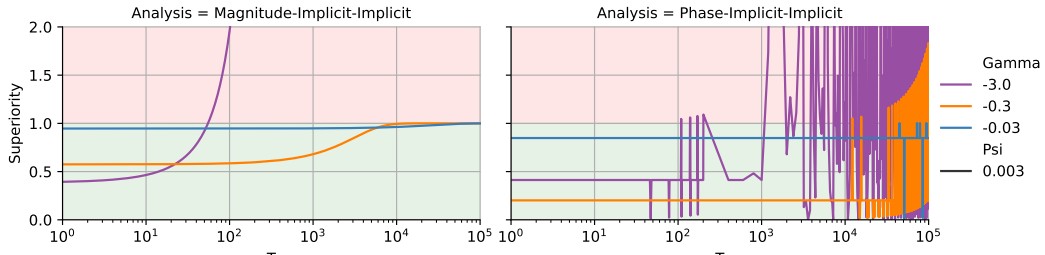

Figure 8: The superiority achieved in the theoretical advection example shows persistence across multiple test rollout steps. For regimes within the stability limit $|\gamma_1| < 1$, superiority converges to 1 after many time steps $T \gg 1$, demonstrating sustained performance parity with the baseline. Superiority in terms of phase or angle appears to persist until numerical issues in evaluating the $\arctan$ arise. For magnitude-based superiority (i.e., $\xi^{[t]} < 1$), the advantage is preserved for a significant number of practical time steps before eventually equalizing with the baseline.

### A.1.6  Influence of the Number of Test Rollout Steps

While the main text focused on a single-step evaluation scenario, here we extend our analysis to multi-step test rollouts. For this, we consider this form of the superiority

$$\xi^{[t]} = \left| \frac{|\hat{q}^t_{\hat{\iota}_\psi,\phi}| - |\tilde{\alpha}^t_\phi|}{|\hat{\iota}^t_\phi| - |\hat{\alpha}^t_\phi|} \right| \tag{47}$$

again with the implicit scheme $\hat{\iota}_\phi$ as the training reference, the implicit scheme $\hat{\iota}_\phi$ as the baseline, and the analytical scheme $\hat{\alpha}_\phi$ as the testing reference. We present the results in figure 8. While the results show that the superiority in terms of magnitude is lost for large $t$, it is maintained for a significant number of steps. For faster dynamics, i.e., higher $|\gamma_1|$, the superiority is lost faster.

### A.2  Diffusion Equation

We now turn to the one-dimensional linear diffusion equation, also known as the heat equation, given by

$$\partial_t u = \nu \partial_{xx} u, \tag{48}$$

where $\nu > 0$ is the diffusion coefficient. As with the advection equation, we consider a unit interval domain ($L = 1$) with periodic boundary conditions. To streamline the analysis, we introduce the reparametrized form of

$$\partial_t u = \frac{\gamma_2}{2N^2 \Delta t} \partial_{xx} u, \tag{49}$$

where the dimensionless parameter $\gamma_2$ is defined as

$$\gamma_2 = 2\nu N^2 \Delta t. \tag{50}$$

In the derivations to follow, the number of spatial points $N$ and the the time step $\Delta t$ will cancel.

### A.2.1  Deriving the Schemes for the Diffusion Equation

The second spatial derivative $\partial_{xx} u$ is commonly approximated using a second-order central finite difference scheme: $\partial_{xx} u_i \approx (u_{i+1} - 2u_i + u_{i-1})/\Delta x^2$. Substituting this into the reparametrized equation equation 49, we obtain the semi-discrete form:

$$\frac{d}{dt} u_i = \frac{\gamma_2}{2N^2 \Delta t} \frac{u_{i+1} - 2u_i + u_{i-1}}{\Delta x^2}. \tag{51}$$

Given that the spatial discretization uses $N$ intervals of length $\Delta x = 1/N$ (since $L = 1$), we have $\Delta x^2 = 1/N^2$. This $1/N^2$ term in the denominator cancels with the $N^2$ term in the definition of $\gamma_2$ within the fraction, simplifying the equation to:

$$\frac{d}{dt} u_i = \frac{\gamma_2}{2\Delta t} (u_{i+1} - 2u_i + u_{i-1}). \tag{52}$$

Applying a forward Euler method for the time derivative, $(u_i^{[t+1]} - u_i^{[t]})/\Delta t$, and evaluating the right-hand side at time $t$, yields the explicit Forward Time Centered Space (FTCS) scheme:

$$\frac{u_i^{[t+1]} - u_i^{[t]}}{\Delta t} = \frac{\gamma_2}{2\Delta t}(u_{i+1}^{[t]} - 2u_i^{[t]} + u_{i-1}^{[t]}). \tag{53}$$

The $\Delta t$ terms cancel, resulting in the fully discrete update rule:

$$u_i^{[t+1]} = u_i^{[t]} + \frac{\gamma_2}{2}(u_{i+1}^{[t]} - 2u_i^{[t]} + u_{i-1}^{[t]}). \tag{54}$$

This can be written in matrix form for the entire state vector $\mathbf{u}_h$:

$$\underbrace{\begin{bmatrix} u_0^{[t+1]} \\ u_1^{[t+1]} \\ \vdots \\ u_{N-1}^{[t+1]} \end{bmatrix}}_{=\mathbf{u}_h^{[t+1]}} = \underbrace{\begin{bmatrix} 1-\gamma_2 & \gamma_2/2 & \cdots & 0 & \gamma_2/2 \\ \gamma_2/2 & 1-\gamma_2 & \cdots & 0 & 0 \\ \vdots & \ddots & \ddots & \ddots & \vdots \\ 0 & \cdots & \gamma_2/2 & 1-\gamma_2 & \gamma_2/2 \\ \gamma_2/2 & \cdots & 0 & \gamma_2/2 & 1-\gamma_2 \end{bmatrix}}_{=\mathbf{A}_D} \underbrace{\begin{bmatrix} u_0^{[t]} \\ u_1^{[t]} \\ \vdots \\ u_{N-1}^{[t]} \end{bmatrix}}_{=\mathbf{u}_h^{[t]}}. \tag{55}$$

Note that the diagonal elements are $1 - \gamma_2$ because the term $+\frac{\gamma_2}{2}(-2u_i^{[t]})$ contributes $-\gamma_2 u_i^{[t]}$. The matrix $\mathbf{A}_D$ is circulant. Its operation is equivalent to a convolution with a kernel $[\gamma_2/2, 1 - \gamma_2, \gamma_2/2]$ (when centered at $1 - \gamma_2$). In Fourier space, the update rule becomes $\hat{\mathbf{u}}_h^{[t+1]} = \hat{\mathbf{e}}_h \odot \hat{\mathbf{u}}_h^{[t]}$, where the Fourier multiplier $\hat{\mathbf{e}}_h$ for mode $m$ (corresponding to relative mode $\phi = m/N \in [0, 1/2]$) is:

$$\hat{e}_\phi = 1 + \frac{\gamma_2}{2}(e^{i2\pi\phi} - 2 + e^{-i2\pi\phi}) \tag{56}$$

$$= 1 + \frac{\gamma_2}{2}(2\cos(2\pi\phi) - 2) \tag{57}$$

$$= 1 + \gamma_2(\cos(2\pi\phi) - 1). \tag{58}$$

The FTCS scheme for diffusion is stable if $\hat{e}_\phi \leq 1$, which implies $\gamma_2 \leq 1$ for the worst-case mode $(\cos(2\pi\phi) = -1)$.

**Implicit Scheme for Diffusion**  Using a backward Euler method for time discretization (evaluating the spatial derivative at time $t + 1$) yields the backward-in-time central-in-space (BTCS) scheme:

$$u_i^{[t+1]} - u_i^{[t]} = \frac{\gamma_2}{2}(u_{i+1}^{[t+1]} - 2u_i^{[t+1]} + u_{i-1}^{[t+1]}) \tag{59}$$

Rearranging gives:

$$u_i^{[t+1]} - \frac{\gamma_2}{2}(u_{i+1}^{[t+1]} - 2u_i^{[t+1]} + u_{i-1}^{[t+1]}) = u_i^{[t]} \tag{60}$$

The operator acting on $\mathbf{u}_h^{[t+1]}$ in Fourier space has the multiplier $1 - \gamma_2(\cos(2\pi\phi) - 1)$. Thus, the Fourier multiplier for the implicit BTCS scheme is:

$$\hat{\imath}_\phi = \frac{1}{1 - \gamma_2(\cos(2\pi\phi) - 1)}. \tag{61}$$

This scheme is unconditionally stable for $\gamma_2 > 0$.

**Analytical Scheme for Diffusion**  In Fourier space, the diffusion equation transforms to

$$\frac{d}{dt}\hat{u}_m = \nu(-k_m^2)\hat{u}_m = -\nu\left(\frac{2\pi m}{L}\right)^2 \hat{u}_m. \tag{62}$$

For $L = 1$, this simplifies to

$$\frac{d}{dt}\hat{u}_m = -\nu(2\pi m)^2 \hat{u}_m. \tag{63}$$

The solution after one time step $\Delta t$ becomes

$$\hat{u}_m^{[t+1]} = \hat{u}_m^{[t]} \exp(-\nu(2\pi m)^2 \Delta t). \tag{64}$$

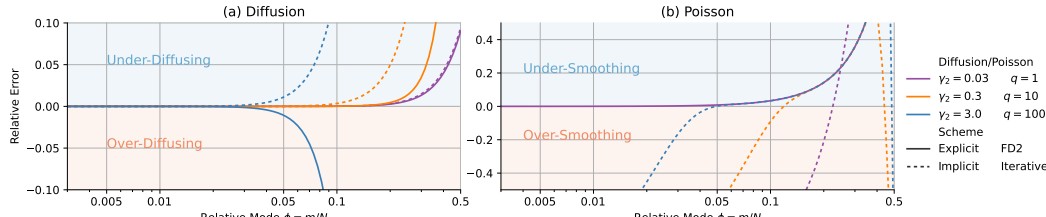

Figure 9: Error analysis for diffusion (a) and Poisson (b) schemes. Similar to the advection analysis in figure 2, the explicit scheme has a smaller error when within its stability limit $\gamma_2 < 1$ (a). For the Poisson equation, iterative solutions converge against the curve of the direct inversion with second-order finite difference scheme when the number of iterations $q$ increases (b).

Using the substitution $\gamma_2 = 2\nu N^2 \Delta t$, we have $\nu \Delta t = \gamma_2/(2N^2)$. The exponent can then be rewritten as

$$-\nu(2\pi m)^2 \Delta t = -\frac{\gamma_2}{2N^2}(2\pi m)^2 \tag{65}$$

$$= -\frac{\gamma_2}{2N^2} \cdot 4\pi^2 m^2 \tag{66}$$

$$= -\gamma_2 \frac{2\pi^2 m^2}{N^2}. \tag{67}$$

So, the analytical Fourier multiplier is

$$\hat{\alpha}_\phi = \exp\left(-\gamma_2 \frac{2\pi^2 m^2}{N^2}\right) = \exp\left(-\frac{\gamma_2}{2}(2\pi\phi)^2\right). \tag{68}$$

### A.2.2 Errors of the Diffusion Schemes

Using the relative mode $\phi = m/N$, the multipliers are

$$\hat{e}_\phi = 1 + \gamma_2(\cos(2\pi\phi) - 1), \tag{69}$$

$$\hat{\iota}_\phi = \frac{1}{1 - \gamma_2(\cos(2\pi\phi) - 1)}, \tag{70}$$

$$\hat{\alpha}_\phi = \exp\left(-\frac{\gamma_2}{2}(2\pi\phi)^2\right). \tag{71}$$

The relative magnitude errors are

$$\frac{|\hat{e}_\phi| - |\hat{\alpha}_\phi|}{|\hat{\alpha}_\phi|} = (\gamma_2 \cdot (\cos(2 \cdot \pi \cdot \phi) - 1) + 1) \cdot e^{2 \cdot \pi^2 \cdot \gamma_2 \cdot \phi^2} - 1. \tag{72}$$

and

$$\frac{|\hat{\iota}_\phi| - |\hat{\alpha}_\phi|}{|\hat{\alpha}_\phi|} = \frac{-2 \cdot \gamma_2 \cdot \sin^2(\pi \cdot \phi) + e^{2 \cdot \pi^2 \cdot \gamma_2 \cdot \phi^2} - 1}{2 \cdot \gamma_2 \cdot \sin^2(\pi \cdot \phi) + 1}. \tag{73}$$

Since these schemes for diffusion using central differences do not introduce phase errors (their Fourier multipliers are real), only magnitude errors are analyzed. These magnitude errors are depicted in figure 9a for various $\phi$ and $\gamma_2$.

### A.2.3 Closed-form Superiority Expressions for Diffusion

Similar to the advection case, an emulator (ansatz) can be trained on data from a low-fidelity diffusion solver (e.g., the explicit FTCS scheme $\hat{e}_\phi$, or the implicit BTCS scheme $\hat{\iota}_\phi$) and then evaluated against the analytical solution $\hat{\alpha}_\phi$. Here, we will choose the ansatz

$$\hat{q}_\phi = 1 + \theta(\cos(2\pi\phi) - 1), \tag{74}$$

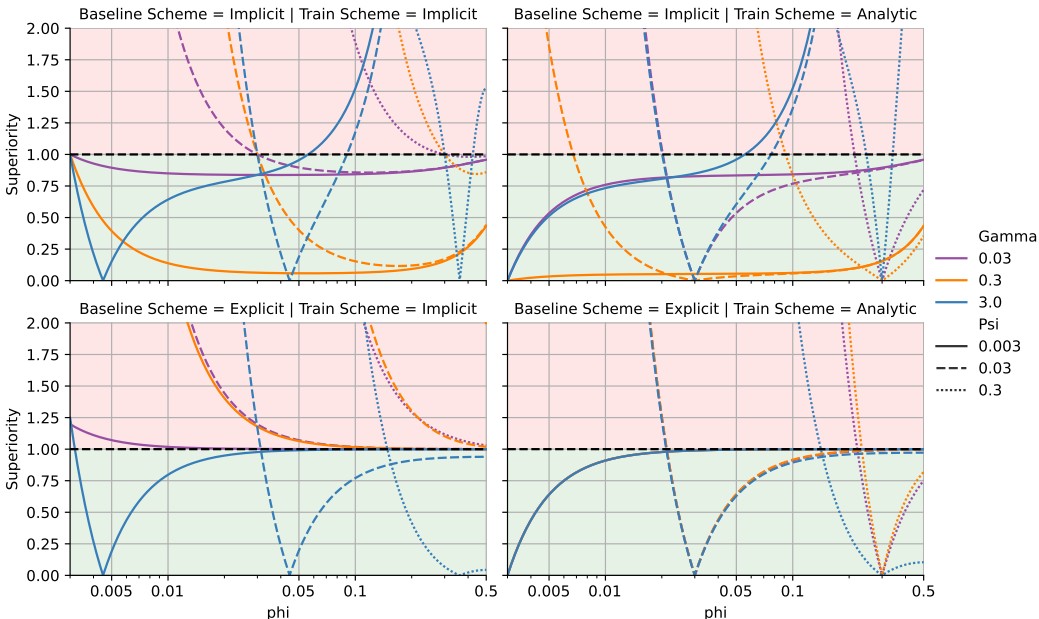

Figure 10: Superiority analysis for the diffusion equation across different combinations of training references (columns) and baseline references (rows). The analytical scheme serves as the testing reference for all evaluations, revealing regimes where trained emulators outperform their baseline solvers.

i.e., it has a functional form similar to the FTCS scheme but has the $\gamma_2$ value as a free parameter $\theta$. When training on the implicit scheme at relative mode $\psi = M/N \in [0, 1/2]$, we find

$$\theta = \frac{\gamma_2}{2 \cdot \gamma_2 \cdot \sin^2 (\pi \cdot \psi) + 1}. \tag{75}$$

In other words, in contrast to the FTCS scheme, our fitted ansatz finds a "corrected" $\gamma_2$. It converges to $\gamma_2$ if $\psi$ is close to 0. We can plug the found $\theta$ back into the ansatz. Since the ansatz, due to its inductive biases, better approximates $\hat{\alpha}_\phi$ than the training solver does (especially in regions where the training solver has large errors), superiority can be observed. Detailed expressions for $q_{\iota_\psi, \phi}$ (the fitted ansatz) and the resulting superiority ratios are provided in the supplemental HTML material: https://tum-pbs.github.io/emulator-superiority/symbolic-expressions. Various combinations of training and baseline schemes, all tested against the analytical solution, are presented in figure 10.

## A.3   Poisson Equation

The Poisson equation, $-\partial_{xx} u = f$, serves as a prototypical example of an elliptic partial differential equation (PDE). It is found at the core of many steady-state physical descriptions, such as modeling electrical potentials or structural deformations. Furthermore, it frequently appears as a sub-problem when integrating systems with constraints, for instance, in the pressure correction step (projection method) for solving the incompressible Navier-Stokes equations.

### A.3.1   Deriving the Schemes for the Poisson Equation

We consider a unit interval domain ($L = 1$) with periodic boundary conditions.

**Direct Finite Difference (FD) Scheme**   The second-order spatial derivative can be approximated using a second-order central finite difference scheme. For the Poisson equation, this yields:

$$-\frac{u_{i+1} - 2u_i + u_{i-1}}{\Delta x^2} = f_i, \tag{76}$$

where $\Delta x = 1/N$ is the spatial step size. This equation represents a linear system $\mathbf{A_P} \mathbf{u}_h = \mathbf{f}_h$, which can be solved directly.

In Fourier space, the operator corresponding to the central finite difference approximation of $-\partial_{xx}$ has a multiplier for mode $m$ (with relative mode $\phi = m/N \in [0, 1/2]$) given by

$$\hat{k}_{FD,\phi} = \frac{-(2\cos(2\pi\phi) - 2)}{\Delta x^2} = \frac{2(1 - \cos(2\pi\phi))}{\Delta x^2}. \tag{77}$$

Thus, the equation in Fourier space is $\hat{k}_{FD,\phi}\hat{u}_m = \hat{f}_m$. Solving for $\hat{u}_m$ gives the Fourier multiplier for the direct FD solution:

$$\hat{\beta}_\phi = \frac{1}{\hat{k}_{FD,\phi}} = \frac{\Delta x^2}{2(1 - \cos(2\pi\phi))}. \tag{78}$$

The solution for each mode is then $\hat{u}_m = \hat{\beta}_\phi \hat{f}_m$. This method, while using finite differences, provides a direct (non-iterative) solution in Fourier space.

**Iterative Scheme Based on FTCS (Richardson Iteration)**    Alternatively, the Poisson equation can be solved iteratively by considering it as the steady-state solution of a pseudo-time-dependent diffusion (heat) equation: $\partial_\tau u = \partial_{xx} u + f$. As pseudo-time $\tau \to \infty$, we expect $\partial_\tau u \to 0$, which recovers the solution to $-\partial_{xx} u = f$. Applying the Forward Time Central Space (FTCS) scheme to this pseudo-time problem leads to the Richardson iteration. In Fourier space, for mode $m$ (corresponding to relative mode $\phi = m/N \in [0, 1/2]$), the update is

$$\hat{u}_m^{[q+1]} = \hat{u}_m^{[q]} + \delta_\tau \left( \frac{2(\cos(2\pi\phi) - 1)}{\Delta x^2} \hat{u}_m^{[q]} + \hat{f}_m \right), \tag{79}$$

where $\delta_\tau$ is the pseudo-time step and $q$ is the iteration count. For optimal convergence (related to the FTCS stability for the heat equation), we choose the maximum stable pseudo-time step, $\delta_\tau = \Delta x^2/2$. Substituting this into equation 79 yields:

$$\hat{u}_m^{[q+1]} = \hat{u}_m^{[q]} + \frac{\Delta x^2}{2} \left( \frac{2(\cos(2\pi\phi) - 1)}{\Delta x^2} \hat{u}_m^{[q]} + \hat{f}_m \right) \tag{80}$$

$$= \cos(2\pi\phi)\hat{u}_m^{[q]} + \frac{\Delta x^2}{2}\hat{f}_m. \tag{81}$$

This is a linear affine recurrence relation. With an initial guess $\hat{u}_m^{[0]}$, the solution after $q$ iterations is:

$$\hat{u}_m^{[q]} = (\cos(2\pi\phi))^q \hat{u}_m^{[0]} + \frac{1 - (\cos(2\pi\phi))^q}{1 - \cos(2\pi\phi)} \frac{\Delta x^2}{2} \hat{f}_m. \tag{82}$$

A common initialization is $\hat{u}_m^{[0]} = 0$. Under this assumption, $\hat{u}_m^{[q]} = \hat{\iota}_{\phi,q}\hat{f}_m$, where the iteration-dependent Fourier multiplier for this iterative solver is:

$$\hat{\iota}_{\phi,q} = \frac{\Delta x^2}{2} \frac{1 - (\cos(2\pi\phi))^q}{1 - \cos(2\pi\phi)}. \tag{83}$$

In the limit as the number of iterations $q$ approaches infinity (with the important caveat that $2\pi\phi$ must not equal 0 or any integer multiple of $\pi$, which is equivalent to requiring that the relative mode $\phi$ must not equal 0 or any integer multiple of $1/2$), we observe that the iterative solver's Fourier multiplier $\hat{\iota}_{\phi,q}$ converges asymptotically to the direct finite difference solution multiplier $\hat{\beta}_\phi$ as defined in equation equation 78. This convergence behavior demonstrates that the iterative scheme ultimately produces a solution that matches the solution of the discretized Poisson equation obtained through direct finite difference methods.

**Analytical (Fourier-Spectral) Scheme**    While the direct FD solution $\hat{\beta}_\phi$ is exact for the discretized system equation 76, it still contains spatial discretization error compared to the true solution of the continuous PDE $-\partial_{xx} u = f$. A fully analytical solution method, often termed a Fourier-spectral method, utilizes the exact Fourier representation of the continuous differential operator. The continuous second derivative $\partial_{xx}$ acting on a function $u(x)$ transforms to multiplication by $(-k_m^2)$ in Fourier space, where $k_m = 2\pi m/L$ is the dimensional wavenumber for mode $m$. For $L = 1$, $k_m = 2\pi m = 2\pi N\phi$. Thus, for the continuous Poisson equation $-\partial_{xx} u = f$, its Fourier transform is $-(-k_m^2)\hat{u}_m = k_m^2 \hat{u}_m = \hat{f}_m$. The Fourier multiplier for the "true" analytical inverse Laplacian operator $(-\partial_{xx})^{-1}$ (using relative mode $\phi = m/N$) is therefore

$$\hat{\alpha}_\phi = \frac{1}{k_m^2} = \frac{1}{(2\pi N\phi)^2} = \frac{(\Delta x)^2}{(2\pi\phi)^2} \quad \text{(for } \phi \neq 0). \tag{84}$$

For $m = 0$ (the mean or DC component), $\hat{f}_0$ must be zero for a periodic solution to exist (compatibility condition), and $\hat{u}_0$ is typically set to zero or determined by other means (e.g., to ensure zero mean solution). This multiplier $\hat{\alpha}_\phi$ is free from spatial discretization error, assuming $f$ is sufficiently smooth and periodic such that its Fourier series converges appropriately.

### A.3.2 Errors of the Poisson Schemes

We will use the following versions of the Fourier multipliers

$$\hat{\alpha}_\phi = \frac{1}{4 \cdot \pi^2 \cdot \phi^2}, \tag{85}$$

$$\hat{\beta}_\phi = \frac{1}{2 - 2 \cdot \cos(2 \cdot \pi \cdot \phi)}, \tag{86}$$

$$\hat{\iota}_{\phi,q} = \frac{0.5 - 0.5 \cdot \cos^q(2 \cdot \pi \cdot \phi)}{1 - \cos(2 \cdot \pi \cdot \phi)}. \tag{87}$$

Note that we excluded the factor of $\Delta x^2$ in the definition of all schemes since it just acts as a global scaling factor and does not affect the relative errors (and the superiority expressions).

For evaluating the different approximate solution methods, we use the fully analytical (Fourier-spectral) solution $\hat{\alpha}_\phi$ (from equation 84, with $\phi = m/N$) as the ground truth reference. Let $\hat{e}_\phi$ denote the direct FD solution multiplier $\hat{\beta}_\phi$ (from equation 78), and $\hat{\iota}_{\phi,q}$ denote the iterative solver multiplier (from equation 83).

The relative magnitude error of the direct FD scheme is:

$$\frac{|\hat{e}_\phi| - |\hat{\alpha}_\phi|}{|\hat{\alpha}_\phi|} = \frac{\pi^2 \cdot \phi^2}{\sin^2(\pi \cdot \phi)} - 1 \tag{88}$$

And the relative magnitude error of the iterative scheme (after $q$ iterations) is:

$$\frac{|\hat{\iota}_{\phi,q}| - |\hat{\alpha}_\phi|}{|\hat{\alpha}_\phi|} = 2.0 \cdot \pi^2 \cdot \phi^2 \cdot \left| \frac{\cos^q(2 \cdot \pi \cdot \phi) - 1}{\cos(2 \cdot \pi \cdot \phi) - 1} \right| - 1.0 \tag{89}$$

These errors are plotted in figure 9b. One might also be interested in the error of the iterative scheme relative to the direct FD scheme it is attempting to converge to. This "iterative truncation error" is:

$$\frac{|\hat{\iota}_{\phi,q}| - |\hat{e}_\phi|}{|\hat{e}_\phi|} = 2.0 \cdot \sin^2(\pi \cdot \phi) \cdot \left| \frac{\cos^q(2 \cdot \pi \cdot \phi) - 1}{\cos(2 \cdot \pi \cdot \phi) - 1} \right| - 1 \tag{90}$$

### A.3.3 Closed-form Superiority Expressions for Poisson

As an ansatz, we choose the following functional form:

$$\hat{q}_\phi = \frac{\theta}{2(1 - \cos(2\pi\phi))}, \tag{91}$$

where $\theta$ is a free parameter. Its functional form is similar to the finite difference direct solver (FD) solution, but has the $\theta$ parameter as a free parameter. When this emulator is trained on data from an unconverged iterative solver $\hat{\iota}_{\psi,q}$ for a fixed $q$ and training mode $\psi$ and evaluated against the fully analytical (Fourier-spectral) solution ($\hat{\alpha}_\phi$), we find the parameter value

$$\theta = 1.0 - \cos^q(6.28318530717959 \cdot \psi). \tag{92}$$

Superiority can arise if the emulator better approximates $\hat{\alpha}_\phi$. The baseline for comparison is the training data source, $\hat{\iota}_{\phi,q}$. The superiority ratio is then:

$$\begin{aligned}
\xi_{\text{Poiss}} &= \left| \frac{|\hat{q}_{\hat{\iota}_\psi,\phi}| - |\hat{\alpha}_\phi|}{|\hat{\iota}_{\phi,q}| - |\hat{\alpha}_\phi|} \right| \\
&= \frac{\pi^2 \cdot \phi^2 \cdot \left| \frac{\cos^q(6.28318530717959 \cdot \psi) - 1.0}{\sin^2(\pi \cdot \phi)} \right| - 1}{2.0 \cdot \pi^2 \cdot \phi^2 \cdot \left| \frac{\cos^q(2 \cdot \pi \cdot \phi) - 1}{\cos(2 \cdot \pi \cdot \phi) - 1} \right| - 1.0}
\end{aligned} \tag{93}$$

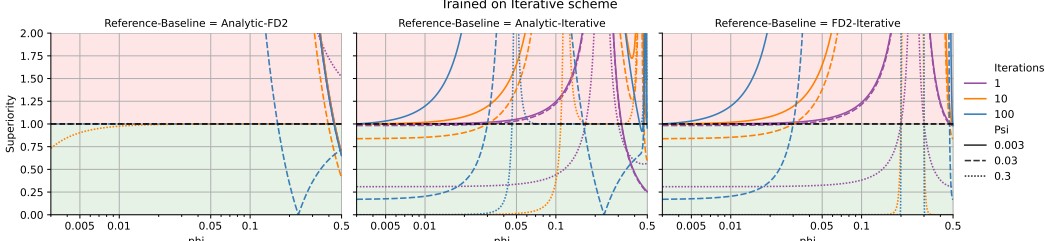

Figure 11: Analysis of emulator superiority for the Poisson equation when the emulator is trained on data from the iterative (Richardson) scheme (using $q$ iterations, denoted $\hat{\iota}_{\psi,q}$). The panels explore superiority under different evaluation configurations: The baseline solver (denominator of the superiority ratio, equation 5) is varied, for instance, using the iterative solver itself ($\hat{\iota}_{\phi,q}$) or the direct Finite Difference (FD) solver ($\hat{\beta}_\phi$). The high-fidelity reference solver (against which errors are measured) is also varied, comparing against either the direct FD solution ($\hat{\beta}_\phi$) or the fully analytical Fourier-spectral solution ($\hat{\alpha}_\phi$). This highlights how the emulator's performance advantage shifts based on whether it is compared to its direct training data or a more accurate solver, and whether the 'ground truth' for error calculation includes discretization errors (FD) or is error-free (analytical). The impact of the training data's convergence level ($q$) on achieving superiority is also demonstrated.

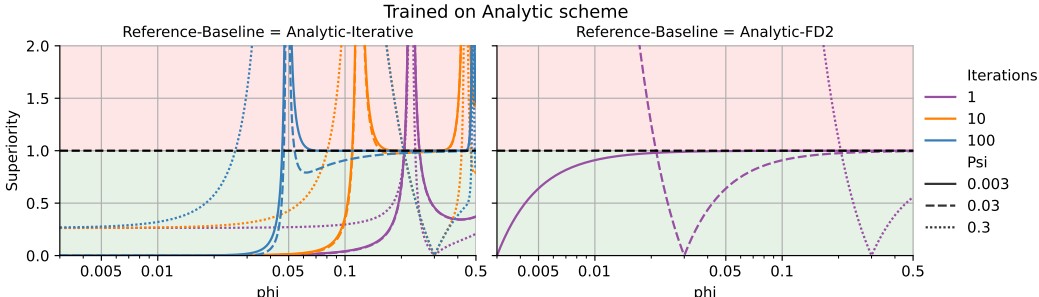

Figure 12: Analysis of emulator superiority for the Poisson equation when the emulator is trained exclusively on data from the fully analytical Fourier-spectral solution (denoted $\hat{\alpha}_\psi$). (Training on the direct Finite Difference (FD) scheme, $\hat{\beta}_\psi$, is not shown as it would lead to the emulator re-learning that scheme due to identical functional forms, precluding superiority against it). The panels explore how superiority, defined by the ratio in equation 5, is observed under different evaluation configurations. The baseline for comparison (denominator of the ratio) is varied, potentially including a less accurate iterative scheme (e.g., $\hat{\iota}_{\phi,q}$) or the FD scheme itself ($\hat{\beta}_\phi$). The high-fidelity reference (against which errors of both emulator and baseline are measured) is always the true analytical solution ($\hat{\alpha}_\phi$). This reveals how the choice of baseline and ultimate reference truth influences the observed superiority, even when the emulator is trained on high-quality data.

Detailed expressions are provided in the supplemental HTML material. Illustrative examples of such superiority for combinations of training, testing and baseline solvers are given in figure 11 and figure 12.

## A.4 Additional Derivations

### A.4.1 Derivation of the Closed-Form Optimizer for the Linear Case

We aim to show that for the linear simulator and the linear emulator, the optimization problem can be solved in closed-form, and that for single-mode training, the solution is independent of the specific distribution over the mode's amplitude and phase.

Recall the one-step prediction error objective using the mean-squared error (MSE) loss, $\zeta(\boldsymbol{a}, \boldsymbol{b}) = \frac{1}{N}\|\boldsymbol{a} - \boldsymbol{b}\|_2^2$:

$$L(\theta) = \mathbb{E}_{\mathbf{u}_h \sim \mathcal{I}_h} \left[ \frac{1}{N} \left\| f_\theta(\mathbf{u}_h) - \mathcal{P}_h(\mathbf{u}_h) \right\|_2^2 \right] \tag{94}$$

where $\mathbf{u}_h \sim \mathcal{I}_h$ denotes an initial state sampled from the distribution $\mathcal{I}_h$, $\mathcal{P}_h$ is the low-fidelity solver (our training reference), and $f_\theta$ is our linear emulator.

The key to simplifying this is to move to the Fourier domain. By Parseval's theorem, the L2-norm in state space is related to the L2-norm in Fourier space by a scaling factor: $\|\mathbf{x}\|_2^2 = \frac{1}{N} \|\hat{\mathbf{x}}\|_2^2$, where $\hat{\mathbf{x}}$ is the Discrete Fourier Transform (DFT) of $\mathbf{x}$. Applying this, the loss function becomes:

$$L(\theta) = \mathbb{E}_{\mathbf{u}_h \sim \mathcal{I}_h} \left[ \frac{1}{N^2} \left\| \widehat{f_\theta(\mathbf{u}_h)} - \widehat{\mathcal{P}_h(\mathbf{u}_h)} \right\|_2^2 \right] \tag{95}$$

By the Convolution Theorem, the cross-correlation defining our linear emulator $f_\theta$ becomes an element-wise product in the Fourier domain. Let $\hat{\mathbf{q}}_h(\theta)$ be the Fourier multiplier for our emulator and $\hat{\mathbf{r}}_h$ be the multiplier for the reference solver $\mathcal{P}_h$. The loss function is then:

$$L(\theta) = \mathbb{E}_{\mathbf{u}_h \sim \mathcal{I}_h} \left[ \frac{1}{N^2} \left\| \hat{\mathbf{q}}_h(\theta) \odot \hat{\mathbf{u}}_h - \hat{\mathbf{r}}_h \odot \hat{\mathbf{u}}_h \right\|_2^2 \right] = \mathbb{E}_{\mathbf{u}_h \sim \mathcal{I}_h} \left[ \frac{1}{N^2} \left\| (\hat{\mathbf{q}}_h(\theta) - \hat{\mathbf{r}}_h) \odot \hat{\mathbf{u}}_h \right\|_2^2 \right] \tag{96}$$

Now, we introduce the crucial assumption from our theoretical analysis: the training data distribution $\mathcal{I}_h$ only produces states with energy in a single Fourier mode, $M$. This means the Fourier representation $\hat{\mathbf{u}}_h$ is a sparse vector with only one non-zero complex-valued entry, $\hat{u}_M$, at index $M$. The L2-norm (which is the square root of the sum of squared magnitudes) collapses to a single term:

$$L(\theta) = \mathbb{E}_{\hat{u}_M} \left[ \frac{1}{N^2} \left| (\hat{q}_M(\theta) - \hat{r}_M) \cdot \hat{u}_M \right|^2 \right] \tag{97}$$

Using the property $|ab|^2 = |a|^2 |b|^2$ for complex numbers, we can separate the terms:

$$L(\theta) = \mathbb{E}_{\hat{u}_M} \left[ \frac{1}{N^2} \left| \hat{q}_M(\theta) - \hat{r}_M \right|^2 \left| \hat{u}_M \right|^2 \right] \tag{98}$$

The term $|\hat{q}_M(\theta) - \hat{r}_M|^2$ depends only on the parameters $\theta$ and the known properties of the solvers; it is constant with respect to the expectation over the data distribution. We can therefore factor it out of the expectation:

$$L(\theta) = \frac{1}{N^2} \left| \hat{q}_M(\theta) - \hat{r}_M \right|^2 \cdot \mathbb{E}_{\hat{u}_M} \left[ \left| \hat{u}_M \right|^2 \right] \tag{99}$$

This final form makes the key point clear. The term $\mathbb{E}[|\hat{u}_M|^2]$ is simply the expected energy of the training data at mode $M$. This is a positive constant determined by the distribution over the amplitude of the mode. Crucially, the phase of $\hat{u}_M$ is entirely absent, and the overall distribution of amplitude only contributes a constant positive scaling factor to the loss. Hence, we can write

$$L(\theta) \sim \left| \hat{q}_M(\theta) - \hat{r}_M \right|^2 \tag{100}$$

To minimize $L(\theta)$, we only need to minimize the term $|\hat{q}_M(\theta) - \hat{r}_M|^2$. This minimum is achieved if and only if:

$$\hat{q}_M(\theta) - \hat{r}_M = 0 \quad \implies \quad \hat{q}_M(\theta) = \hat{r}_M \tag{101}$$

Substituting the functional form for our linear ansatz at mode $M$ (with corresponding primitive root of unity $\omega_M = e^{-i2\pi M/N}$), we arrive at the complex-valued equation presented in the main text:

$$\theta_0 + \theta_1 \omega_M = \hat{r}_M \tag{102}$$

This equation can be solved for the two real-valued parameters $\theta_0$ and $\theta_1$. This derivation confirms that under the single-mode training assumption, the optimal parameters are independent of the specific distribution over the amplitude and phase of that mode. It is worth noting that if the training data contained multiple modes, the loss would become a weighted least-squares problem, where the optimal parameters would depend on the relative energy across the modes.

### A.4.2  Proof that One-Step Supervised Training is equivalent to Discrete Residua

We show here that for the linear systems considered in our paper, training with a PI loss formulated as the discrete PDE residual leads to the same optimal emulator parameters as training with the supervised loss if only a single mode $M$ is active. Therefore, our theoretical insights on superiority hold equally for this common PI training paradigm.

Let us consider a more general setting than in the main text of a general linear, time-invariant numerical solver that advances the state vector $\mathbf{u}_h^{[t]}$ to $\mathbf{u}_h^{[t+1]}$ according to

$$\mathbf{A}\mathbf{u}_h^{[t+1]} = \mathbf{B}\mathbf{u}_h^{[t]}, \tag{103}$$

where $\mathbf{A}$ and $\mathbf{B}$ are matrices representing the discretized spatial and temporal operators. This form covers explicit schemes (where $\mathbf{A} = \mathbf{I}$) and implicit schemes. The solver operator is thus $\mathcal{P}_h(\mathbf{u}_h^{[t]}) = \mathbf{A}^{-1}\mathbf{B}\mathbf{u}_h^{[t]}$. Our neural emulator, $f_\theta$, aims to approximate this operator.

We define two loss functions averaged over the training data distribution $\mathcal{I}_h$:

1. **One-Step Supervised Loss** ($\mathcal{L}_{\text{sup}}$): This measures the direct error between the emulator's output and the solver's output.

$$\mathcal{L}_{\text{sup}}(\theta) = \mathbb{E}_{\mathbf{u}_h \sim \mathcal{I}_h} \left[ \frac{1}{N} \left\| f_\theta(\mathbf{u}_h) - \mathcal{P}_h(\mathbf{u}_h) \right\|_2^2 \right] = \mathbb{E}_{\mathbf{u}_h \sim \mathcal{I}_h} \left[ \frac{1}{N} \left\| f_\theta(\mathbf{u}_h) - \boldsymbol{A}^{-1}\boldsymbol{B}\mathbf{u}_h \right\|_2^2 \right] \tag{104}$$

2. **Discrete Residual Loss** ($\mathcal{L}_{\text{res}}$): This measures how well the emulator's output satisfies the underlying discrete PDE.

$$\mathcal{L}_{\text{res}}(\theta) = \mathbb{E}_{\mathbf{u}_h \sim \mathcal{I}_h} \left[ \frac{1}{N} \left\| \mathbf{A} f_\theta(\mathbf{u}_h) - \mathbf{B}\mathbf{u}_h \right\|_2^2 \right] \tag{105}$$

For linear PDEs on periodic domains considered in our paper, the matrices $\mathbf{A}$ and $\mathbf{B}$ are circulant. Circulant matrices are diagonalized by the Discrete Fourier Transform (DFT), meaning the matrix operations become simple element-wise multiplications in the Fourier domain. Let $\hat{\mathbf{a}}$ and $\hat{\mathbf{b}}$ be the Fourier multipliers (eigenvalues) of $\mathbf{A}$ and $\mathbf{B}$, respectively. The Fourier multiplier of the solver $\mathcal{P}_h$ is then $\hat{\mathbf{r}}_h = \hat{\mathbf{a}}^{-1} \odot \hat{\mathbf{b}}$. Let $\hat{\mathbf{q}}_h(\theta)$ be the multiplier for our linear emulator $f_\theta$.

Using Parseval's theorem, we can express the losses in the Fourier domain. The **supervised loss** becomes

$$\mathcal{L}_{\text{sup}}(\theta) = \mathbb{E}_{\hat{\mathbf{u}}_h} \left[ \frac{1}{N^2} \left\| \hat{\mathbf{q}}_h(\theta) \odot \hat{\mathbf{u}}_h - \hat{\mathbf{r}}_h \odot \hat{\mathbf{u}}_h \right\|_2^2 \right] = \mathbb{E}_{\hat{\mathbf{u}}_h} \left[ \frac{1}{N^2} \left\| (\hat{\mathbf{q}}_h(\theta) - \hat{\mathbf{r}}_h) \odot \hat{\mathbf{u}}_h \right\|_2^2 \right]. \tag{106}$$

The **residual loss** can be rewritten by factoring out $\mathbf{A}$

$$\mathcal{L}_{\text{res}}(\theta) = \mathbb{E}_{\mathbf{u}_h} \left[ \frac{1}{N} \left\| \mathbf{A} \left( f_\theta(\mathbf{u}_h) - \underbrace{\mathbf{A}^{-1}\mathbf{B}}_{=: \mathcal{P}_h} \mathbf{u}_h \right) \right\|_2^2 \right] = \mathbb{E}_{\mathbf{u}_h} \left[ \frac{1}{N} \left\| \mathbf{A} \left( f_\theta(\mathbf{u}_h) - \mathcal{P}_h(\mathbf{u}_h) \right) \right\|_2^2 \right], \tag{107}$$

where we identified the solver operator. In the Fourier domain, this becomes

$$\mathcal{L}_{\text{res}}(\theta) = \mathbb{E}_{\hat{\mathbf{u}}_h} \left[ \frac{1}{N^2} \| \hat{\mathbf{a}} \odot (\hat{\mathbf{q}}_h(\theta) \odot \hat{\mathbf{u}}_h - \hat{\mathbf{r}}_h \odot \hat{\mathbf{u}}_h) \|_2^2 \right] = \mathbb{E}_{\hat{\mathbf{u}}_h} \left[ \frac{1}{N^2} \| \hat{\mathbf{a}} \odot (\hat{\mathbf{q}}_h(\theta) - \hat{\mathbf{r}}_h) \odot \hat{\mathbf{u}}_h \|_2^2 \right]. \tag{108}$$

Comparing the two loss functions, we see that $\mathcal{L}_{\text{res}}$ is a spectrally weighted version of $\mathcal{L}_{\text{sup}}$. There are three situations to consider:

1. The reference simulator is an explicit scheme, i.e., $A = I$. As such $\hat{a}$ will be vector full of ones and the objectives are identical, $\mathcal{L}_{\text{sup}}(\theta) = \mathcal{L}_{\text{res}}(\theta)$, leading to identical minimizers $\theta^*$.

2. The initial condition only has a single mode active. Similarly to how we argued in section A.4.1, the L2-norm collapse to a single term and the one relevant entry $\hat{a}_M$ is simply a constant scaling factor in the optimization with respect to $\theta$. Consequently, we have $\mathcal{L}_{\text{sup}}(\theta) = \frac{1}{|\hat{a}_M|^2} \mathcal{L}_{\text{res}}(\theta)$ and the optimizers $\theta^*$ are identical.

3. In the most general case of an implicit scheme $A \neq I$ and $\geq 2$ modes being present, supervised loss and residuum loss cannot easily be related to each other. In alignment with section A.4.1 the optimizers $\theta^*$ of both objectives depend on the distribution of energy in the modes. It must be noted that the residuum-based formulation additionally scales (and rotates) the energy in the modes of $\hat{u}_h$ due to the element-wise muliplication with $\hat{a}$.

In conclusion, under the assumption of only a single mode being present, our superiority analysis holds equally for both supervised and physics-informed training.

## B  Experimental Details

Our experiments are implemented in JAX (Bradbury et al., 2018) and build upon components of the APEBench suite (Koehler et al., 2024).

**Architectures**   In particular, we use the following neural architectures:

- ConvNet: A feedforward convolutional network with 10 hidden layers of 34 channels each. Each layer transition except for the last uses the ReLU activation. The effective receptive field is $10 + 1 = 11$ and the network has 31757 learnable parameters.

- UNet: A classical UNet with 2 hierarchical levels (i.e., three different spatial resolutions) and a base width of 12 hidden channels at the highest resolution. It uses double convolution blocks with ReLU activation (preceded by group normalization with one group) per level. The spatial resolution is halved at each of the 2 levels while the channel count doubles. Skip connections exist between the encoder and decoder parts. This configuration has 27193 learnable parameters and an effective receptive field of 29 per direction.

- Dilated ResNet: A Dilated Residual Network with 2 blocks, 32 hidden channels channels, and ReLU activation. Each block uses a series of three convolutions with dilation rates 1, 2, and 1. Each convolution is followed by group normalization (with one group) and then the ReLU activation. This configuration has 31777 parameters and an effective receptive field of 20 per direction.

- FNO: A Fourier Neural Operator with 4 blocks, 18 channels, and 12 active Fourier modes. Each block consists of a spectral convolution and a point-wise linear bypass, with the GELU activation applied to their sum. Lifting and projection layers are point-wise linear (1x1) convolutions. This configuration has 32527 parameters and an infinite receptive field due to the global nature of Fourier transforms.

- Vanilla Transformer: A transformer architectue considereing each spatial degree of freedom a token and performing dense self-attention over them. We consider a setting with 31669 parameters using 28 hidden channels across 4 transformer blocks (each consisting of a dense self-attention in space and a two-layered multi-layer perceptron whose inner dimension is twice as large as the number of hidden channels). Due to the global nature of the attention, the effective receptive field is infinite. However, in contrast to the convolutional

or Fourier-spectral based architectures, the Transformer does not have an inductive bias for the periodic boundary conditions.

All the architectures have trainable parameter numbers in the range of 27-33k to allow for a fair comparison.

**Metrics**    We train the emulators using the mean-squared error (MSE) as training metric and the normalized root-mean-squared error (nRMSE) as test metric.

## B.1    Nonlinear Emulators on Linear Advection

In this section, we detail the experimental setup for the nonlinear emulators applied to the linear advection equation, as presented in section 4.

**Experimental Variables and Training Configuration**    The experiments systematically varied three key factors: the combined variable $\gamma_1$ (which is proportional to the CFL number), the training (and corresponding baseline) simulator, and the initial condition distribution for testing. Specifically, we explored the following combinations:

- Autoregressive Generalization against implicit scheme:
    - $\gamma_1 = -3.0$
    - training simulator: implicit first-order upwind
    - training IC distribution: only first mode active
    - test IC distribution: only first mode active

- Autoregressive Generalization against explicit scheme:
    - $\gamma_1 = -0.9$
    - training simulator: explicit first-order upwind
    - training IC distribution: only first mode active
    - test IC distribution: only first mode active

- State-Space and Autoregressive Generalization against implicit scheme:
    - $\gamma_1 = -3.0$
    - training simulator: implicit first-order upwind
    - training IC distribution: only first mode active
    - test IC distribution: modes one to five active

These combinations were strategically chosen to investigate both autoregressive superiority (using the same initial condition distribution for training and testing) and state-space superiority (using different initial condition distributions). In all cases, the testing reference was the fully analytical solver. The domain is discretized into $N = 100$ points.

**Training and Evaluation Setup**    The training trajectories consist of 300 initial conditions rolled out for 1 time step. Hence, the training dataset contains 300 samples with an input and an output frame. The emulators were trained using a mean squared error (MSE) loss function to predict a single step ahead. For evaluation, we used 10 initial conditions rolled out for the 50 time steps shown in the main text.

The optimization process utilized the Adam optimizer (Kingma and Ba, 2015) with a batch size of 32. We used a warmup cosine decay learning rate schedule featuring a warmup phase followed by cosine decay, with a total of 10,000 training iterations. The learning rate ramped up from 0 to $1 \times 10^{-3}$ during the first 1,000 steps (warmup phase), then gradually decayed to zero following a cosine schedule for the remaining steps.

**Seed Study for Robustness**    We repeated each experiment 12 times with different random seeds. Results in the main text show the median performance across these runs along with the 50% percentile intervals.

**Computer Resources** We conducted this experiment on a single NVIDIA RTX3060 GPU. Running it for all the required seeds took less than 2 hours.

## B.2 Burgers Superiority

### B.2.1 Implicit Time Integrators for the Burgers Equation

The Burgers equation on the one-dimensional unit interval in non-conservative form with periodic boundary conditions reads

$$\frac{\partial u}{\partial t} + u\frac{\partial u}{\partial x} = \nu\frac{\partial^2 u}{\partial x^2} \qquad u(t,0) = u(t,1). \tag{109}$$

Similar to the linear PDEs in section A.1, we will discretize the domain into $N$ intervals of equal length $\Delta x = 1/N$. Then, we consider the left end of each interval a nodal degree of freedom and collect the values of $u$ at these points into the vector $\boldsymbol{u}_h$. In contrast to the linear PDEs, the solver described below will exclusively work in state space.

The matrix associated with the discretized second derivative in one dimension reads

$$\boldsymbol{L}_h := \frac{1}{(\Delta x)^2}\begin{bmatrix} -2 & 1 & 0 & \dots & 0 & 1 \\ 1 & -2 & 1 & \dots & 0 & 0 \\ 0 & 1 & -2 & \dots & 0 & 0 \\ \vdots & & \ddots & & \vdots \\ 1 & 0 & 0 & \dots & 1 & -2 \end{bmatrix}. \tag{110}$$

It it is not exclusively tri-diagonal but also has entries in the top right and bottom left corners.

The convection term requires special treatment because of its nonlinearity and the advection characteristic of the first derivative. Let $\boldsymbol{F}_h$ and $\boldsymbol{B}_h$ represent the forward or backward approximation of the first derivative in one dimension on periodic boundaries, respectively, via

$$\boldsymbol{F}_h := \frac{1}{\Delta x}\begin{bmatrix} -1 & 1 & 0 & \dots & 0 & 0 \\ 0 & -1 & 1 & \dots & 0 & 0 \\ 0 & 0 & -1 & \dots & 0 & 0 \\ \vdots & & \ddots & & \vdots \\ 1 & 0 & 0 & \dots & 0 & -1 \end{bmatrix}, \qquad \boldsymbol{B}_h := \frac{1}{\Delta x}\begin{bmatrix} 1 & 0 & 0 & \dots & 0 & 1 \\ -1 & 1 & 0 & \dots & 0 & 0 \\ 0 & -1 & 1 & \dots & 0 & 0 \\ \vdots & & \ddots & & \vdots \\ 0 & 0 & 0 & \dots & -1 & 1 \end{bmatrix}. \tag{111}$$

Again, note the element in the corner entries of the matrices. Then, we can build an upwind differentiation matrix based on the winds $\boldsymbol{w}_h$

$$\Gamma_h(\boldsymbol{w}_h) = \text{diag}\left(\underbrace{\max\left(\frac{\boldsymbol{s}_{-1}(\boldsymbol{w}_h) + \boldsymbol{w}_h}{2}, 0\right)}_{\text{positive winds}}\right)\boldsymbol{B}_h + \text{diag}\left(\underbrace{\max\left(\frac{\boldsymbol{s}_{1}(\boldsymbol{w}_h) + \boldsymbol{w}_h}{2}, 0\right)}_{\text{negative winds}}\right)\boldsymbol{F}_h. \tag{112}$$

Deducing the positive and negative winds from neighboring averages (using the periodic forward shift $s_{-1}$ and backward shift $s_1$ operators) is necessary to have correct movement if the winds change sign over the domain. If we use the discrete state vector $\boldsymbol{u}_h$ as winds $\boldsymbol{w}_h$, we can discretize the continuous equation via the method of lines as

$$\frac{\mathrm{d}\boldsymbol{u}_h}{\mathrm{d}t} + \Gamma_h(\boldsymbol{u}_h)\boldsymbol{u}_h = \nu\boldsymbol{L}_h\boldsymbol{u}_h. \tag{113}$$

Naturally, the spatial discretization of a *nonlinear* PDE leads to a system of *nonlinear* ODEs. If we treat it fully implicitly, here via a backward Euler in time for the time discretization, we get

$$\frac{\boldsymbol{u}_h^{[t+1]} - \boldsymbol{u}_h^{[t]}}{\Delta t} + \Gamma_h(\boldsymbol{u}_h^{[t+1]})\boldsymbol{u}_h^{[t+1]} = \nu\boldsymbol{L}_h\boldsymbol{u}_h^{[t+1]}. \tag{114}$$

In other words, in order to advance from $\boldsymbol{u}_h^{[t]}$ to $\boldsymbol{u}_h^{[t+1]}$ (which is what the numerical simulators $\mathcal{P}_h$, and eventually the neural emulator, will do), we need to solve a nonlinear algebraic system of

equations. The structure of the nonlinearity allows for a Picard iteration (Turek, 1999) (which can be viewed as a quasi-Newton method) that introduces an additional iteration index $[k]$ to the state vector at the next point in time

$$\frac{\boldsymbol{u}_h^{[t+1][k+1]} - \boldsymbol{u}_h^{[t]}}{\Delta t} + \Gamma_h(\boldsymbol{u}_h^{[t+1][k]})\boldsymbol{u}_h^{[t+1][k+1]} = \nu \boldsymbol{L}_h \boldsymbol{u}_h^{[t+1][k+1]}. \tag{115}$$

The key idea is that the upwiding matrix is linearized around the previous iterate $\boldsymbol{u}_h^{[t+1][k]}$. As such, one has to solve a linear system for $\boldsymbol{u}_h^{[t+1][k+1]}$ at each Picard iteration step, which reads

$$\underbrace{\left(\boldsymbol{I} + \Delta t\Gamma_h(\boldsymbol{u}_h^{[t+1][k]}) - \Delta t\nu \boldsymbol{L}_h\right)}_{=\boldsymbol{A}_h = \Lambda_h(\boldsymbol{u}_h^{[t+1][k]})} \boldsymbol{u}_h^{[t+1][k+1]} = \boldsymbol{u}_h^{[t]}. \tag{116}$$

Note that across the Picard iterations, the right-hand side is constant, i.e., independent of the iteration index. This is because it is always the previous state in time $\boldsymbol{u}_h^{[t]}$. The system matrix $\boldsymbol{A}_h$, on the other hand, is dependent on the previous iteration index $k$ because the upwinding matrix $\Gamma_h(\boldsymbol{u}_h^{[t+1][k]})$ has to be reassembled at each iteration step. We use $N = 60$ for the spatial discretization. As such, the matrices are small and we employ a direct solver via LU decomposition.

While convergence of the fixed-point iteration over $[k]$ is desirable to mimize the nonlinear residual between two consecutive time steps, a common approximation is to perform only a single Picard iteration per time step. This leads to a linear system of the form

$$\underbrace{\left(\boldsymbol{I} + \Delta t\Gamma_h(\boldsymbol{u}_h^{[t]}) - \Delta t\nu \boldsymbol{L}_h\right)}_{=\boldsymbol{A}_h = \Lambda_h(\boldsymbol{u}_h^{[t]})} \boldsymbol{u}_h^{[t+1]} = \boldsymbol{u}_h^{[t]}. \tag{117}$$

We call this the one-step (P1) implicit method.

In our experiments, we use the one-step (P1) implicit method as the training reference (and the baseline during superiority evaluation) for the neural emulator. This method linearizes the nonlinear term by evaluating it at the previous time step and performs only a single matrix assembly and linear system solve per time step. As a result, the nonlinear residual is not fully converged, which leads to errors in shock propagation, especially as the solution develops sharp gradients. This is qualitatively depicted in figure 6 in the main text.

For evaluation, we use a fully converged nonlinear implicit method as the testing reference. This method iterates between assembling the system matrix (which depends on the current iterate of the solution) and solving the resulting linear system, repeating until the nonlinear residual falls below a specified tolerance which we set to $10^{-5}$. This approach yields the correct shock propagation and serves as a high-fidelity reference for comparison.

In both the training and testing references, the linear systems are solved using LU decomposition, ensuring solutions are obtained to machine precision. While we use direct solvers here, employing an unconverged iterative linear solver (e.g., a fixed number of Krylov or Jacobi iterations) could introduce additional sources of error and is a potential avenue for further investigations into emulator superiority.

As initial conditions for both training and testing, we use a state with both the first Fourier mode and the mean/zero mode active. In figure 13, we show the effect of the unconverged Picard iteration on the solution.

### B.2.2 Experimental Details

**Experimental Setup**  We conducted our experiments using the UNet architecture described before. The model was trained on a discretized version of the Burgers equation with a viscosity coefficient of $\gamma_2 = 0.2$ and a convection difficulty parameter of $\delta_1 = -8.0$, representing a strongly nonlinear regime. The spatial domain was discretized using 60 grid points.

**Training and Testing Simulators**  For training, we use the P1 method which truncates the fixed-point iteration after one iteration each time step. This represents our low-fidelity simulator. For testing and evaluation, we used the same scheme but with full convergence, ensuring full convergence of the nonlinear system and thus serving as our high-fidelity reference.

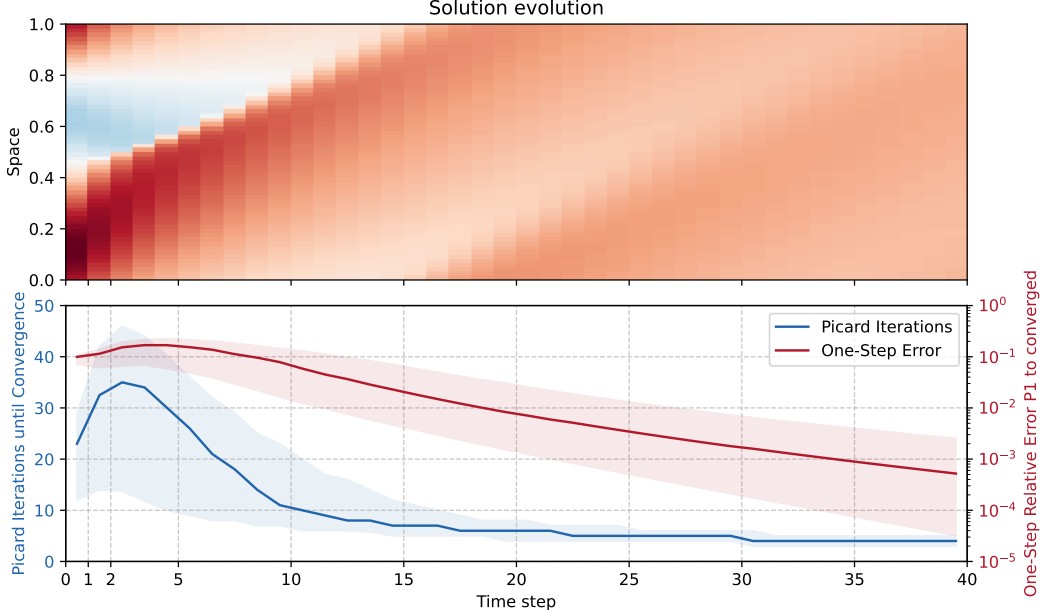

Figure 13: A sample trajectory from the given initial condition distribution evolved with the converged Picard method (top figure). In the lower figure, the blue curve despicts the number of Picard iterations (consisting of linear system assembly and direct LU solve) it requires to converge the fixed-point iteration to a tolerance of $10^{-5}$. The data points are places between two consecutive snapshots to indicate that they relate to the state transition. Requiring $\gg 1$ Picard iterations to converge while only performing one in the case of the P1 method incurs an error in terms of an unconverged nonlinear residual. Here, we measure a one-step error against the converged solution (depicted in red on a logarithmic scale). Shaded areas around the blue and red curve indicate the 50 percentile interval around the median out of 300 different initial conditions. Both the number of Picard iterations and the one-step error peak at the shock formation point and then go down due to the diffusive nature of the problem.

**Initial Conditions**   Both training and testing utilized Fourier-based initial conditions, focusing on the first Fourier mode with randomized phases and an additional offset (i.e., non-zero energy in the mean-mode/zero-mode). This approach generates smooth initial conditions that subsequently develop shocks through the nonlinear dynamics of the Burgers equation.

**Training Protocol**   The network was trained using a one-step prediction framework (temporal horizon train 1) with a mean squared error (MSE) loss function on 500 training samples. For evaluation, we performed autoregressive rollouts over 20 time steps (temporal horizon test 20) on 100 test samples, measuring performance with normalized root mean squared error (nRMSE).

**Optimization Strategy**   We employed the Adam optimizer (Kingma and Ba, 2015) with a batch size of 32 for 100,000 iterations. A warmup cosine learning rate schedule was used, starting from 0.0 and reaching a maximum of 3e-4 after 5,000 warmup steps, followed by cosine decay for the remaining iterations.

**Statistical Robustness**   To ensure statistical significance of our results, we conducted a thorough seed study with 30 different network initialization seeds, while maintaining consistent training and testing data through fixed seeds for those processes. This approach allowed us to quantify the variability in network performance while isolating the effects of initialization.

**Computer Resources**   We conducted this experiment on a single NVIDIA RTX3060 GPU. Running it for all the required seeds took less than 5 hours.

# C    Ablational Experiments

## C.1    Multi-Mode Case Study

If the training data contains multiple modes, the closed-form solution for the emulator parameters is no longer a simple equation for a single mode as shown in section A.4.1. Instead, it becomes a linear least-squares problem weighted by the concrete distribution of complex-values coefficients of $\hat{u}$. While an analytical solution still exists for this linear regression problem, the analysis becomes more complex because of this data dependency.

To provide an intuition, we redid the theoretical experiments numerically (training on a combination of modes and only testing on mode 5) with results displayed in table 1. We found that there is still superiority when more modes are present in the training initial condition as long as they are lower than what is being tested on (confirming again what we called "forward superiority" in section 3.1).

Table 1: The achieved state-space superiority in the linear scenario of section 3.1 (here measured on $m = 5$) depends on how the energy is distributed in the data it has seen during training. We still see "forward superiority" if the training modes are smaller than the testing mode.

| Modes in Train IC Dist | Superiority $\xi^{[1]}$ |
|---|---|
| 1 | 0.57 |
| 1,2 | 0.61 |
| 1,2,3,4 | 0.72 |
| 1,2,3,4,5 | 0.81 |
| 1,2,3,4,5,6 | 0.87 |
| 1,2,3,4,5,6,7 | 1.00 |
| 1-10 | 1.10 |
| 1-20 | 2.30 |
| 4,6 | 1.10 |
| 3,4 | 0.79 |

## C.2    Impact of Receptive Field in CNNs and Active Fourier Modes in FNO

Our results in figure 5c already hint that local architectures (ConvNet) can achieve stronger state-space superiority than global ones (FNO, Transformer). We hypothesize this is because a constrained, local inductive bias acts as a powerful regularizer, preventing the model from overfitting to the global error patterns of the low-fidelity solver.

To provide direct evidence and practical guidance, we performed a new ablation study on the linear advection case, systematically varying the effective receptive field of the ConvNet. We present the resuls in table 2. The configuration with a receptive field of 11 corresponds to the model used in the main paper. The key finding is that superiority is maximized when the receptive field is appropriately matched to the physical characteristics of the problem (in this case, dictated by $\gamma_1$ which is proportional to the CFL number). A receptive field of 4 yields the best performance, achieving a superiority ratio $\xi^{[t]}$ of 0.73 after 10 time steps. Receptive fields that are too small fail to capture the necessary physics, while those that are too large provide excess capacity that diminishes the superiority effect by allowing the model to learn more of the solver's non-physical behavior.

We also conducted an ablation study on the Fourier Neural Operator (FNO) by varying its number of active Fourier modes. The results of table 3 reveal a clear trade-off between model capacity and the implicit regularization that drives superiority. When severely under-parameterized (1 active mode), the FNO fails to learn the problem. The optimal performance is achieved at 2 active modes, which provides just enough capacity to model the target dynamics while the FNO's inherent spectral truncation acts as a powerful regularizer, filtering out numerical artifacts from the solver and leading to strong superiority (ratio of 0.59). As the number of active modes increases further (3+), the FNO gains the capacity to partially overfit to the solver's structured errors across a wider frequency band, which, while still allowing for superiority, diminishes its magnitude. This demonstrates that superiority is maximized when the model is expressive enough to capture the core physics but constrained enough to regularize away the training data's flaws.

Table 2: Influence of receptive field of the CNN on the achieved superiority ratio. This ablation is based on figure 5(c).

| Rec. Field/Time Step | Superiority $\xi^{[t]}$ at Time Step | | | | | | | | | |
|---|---|---|---|---|---|---|---|---|---|---|
| | 1 | 2 | 3 | 4 | 5 | 6 | 7 | 8 | 9 | 10 |
| 1 | 4.0 | 4.0 | 3.9 | 3.9 | 3.9 | 3.8 | 3.8 | 3.7 | 3.7 | 3.6 |
| 2 | 1.0 | 1.0 | 1.0 | 1.0 | 1.0 | 1.0 | 1.0 | 1.0 | 1.0 | 1.0 |
| 3 | 1.0 | 0.95 | 0.91 | 0.88 | 0.86 | 0.84 | 0.83 | 0.81 | 0.80 | 0.80 |
| 4 | 1.0 | 0.94 | 0.90 | 0.86 | 0.83 | 0.80 | 0.78 | 0.76 | 0.75 | 0.73 |
| 5 | 1.0 | 0.94 | 0.90 | 0.86 | 0.83 | 0.81 | 0.79 | 0.77 | 0.76 | 0.74 |
| 7 | 1.0 | 0.96 | 0.93 | 0.91 | 0.89 | 0.87 | 0.86 | 0.85 | 0.84 | 0.83 |
| 11 | 1.0 | 0.96 | 0.93 | 0.91 | 0.89 | 0.87 | 0.86 | 0.84 | 0.83 | 0.82 |

Table 3: Influence of active modes in FNO on the achieved superiority ratio. This ablation is based on figure 5(c).

| Modes/Time Step | Superiority $\xi^{[t]}$ at Time Step | | | | | | | | | |
|---|---|---|---|---|---|---|---|---|---|---|
| | 1 | 2 | 3 | 4 | 5 | 6 | 7 | 8 | 9 | 10 |
| 1 | 8.16 | 8.06 | 7.93 | 7.76 | 7.58 | 7.37 | 7.14 | 6.89 | 6.62 | 6.35 |
| 2 | 1.00 | 0.93 | 0.87 | 0.81 | 0.76 | 0.72 | 0.68 | 0.65 | 0.62 | 0.59 |
| 3 | 1.00 | 0.97 | 0.94 | 0.92 | 0.90 | 0.88 | 0.86 | 0.84 | 0.82 | 0.81 |
| 4 | 1.00 | 0.98 | 0.97 | 0.96 | 0.94 | 0.93 | 0.92 | 0.92 | 0.91 | 0.90 |
| 8 | 1.00 | 0.97 | 0.94 | 0.91 | 0.88 | 0.86 | 0.84 | 0.82 | 0.80 | 0.78 |
| 12 | 1.00 | 0.98 | 0.96 | 0.95 | 0.93 | 0.92 | 0.91 | 0.90 | 0.89 | 0.88 |

## C.3   Effect of Coarse Solver Fidelity by ablating the number of Picard Iterations

In section 5, we argue that the coarse solver needs to be sufficiently inaccurate to enable superiority. Naturally, the question arises by *how much*. While a universal, quantifiable threshold is likely problem-dependent, we can demonstrate the relationship between solver fidelity and the potential for superiority. We have conducted an ablation study for the number of Picard iterations in the Burgers example of section 4.2. We again used the UNet and display the superiority ratio rollout when trained and baselined against a Picard solver truncated after a certain number of iterations. A single iteration represents our original low-fidelity setup, while a higher number of iterations produces a more accurate, higher-fidelity solver.

Table 4: The effect of the number of Picard iterations on the autoregressive superiority achieved in the Burgers experiment.

| Picard Iter/Time Step | 1 | 2 | 3 | 4 | 5 | 6 | 7 | 8 |
|---|---|---|---|---|---|---|---|---|
| 1 (Lowest Fidelity) | 1 | 0.91 | 0.85 | 0.88 | 0.92 | 0.96 | 1 | 1.05 |
| 2 | 1 | 1 | 0.97 | 0.98 | 1.01 | 1.09 | 1.16 | 1.25 |
| 3 | 1 | 1.05 | 1.05 | 1.11 | 1.19 | 1.27 | 1.37 | 1.47 |
| 4 | 1 | 1.36 | 1.34 | 1.46 | 1.53 | 1.58 | 1.67 | 1.84 |

These results clearly demonstrate our central thesis: the potential for superiority is directly related to the magnitude of the structured error in the training data. When the training solver is very coarse, there is significant room for the emulator to learn a more regularized operator, resulting in a strong superiority effect ($\xi^{[t]}$ ratio dropping to 0.85). As the fidelity of the training solver increases, the structured error in the training data decreases, and the potential for superiority vanishes. When trained on a reasonably converged solver, the emulator simply learns to replicate its already accurate behavior, and no superiority is observed.

## C.4 Additional Architectures for Burgers Study

To provide a more complete picture, we extended our analysis of the nonlinear Burgers' equation experiment of section 4.2 to include all architectures from our study and present the results in table 5 (median superiority out of 30 seeds). The results robustly confirm that the superiority phenomenon is not limited to a single architecture but is instead closely tied to the inductive biases of the models. The key finding is that architectures with a spatial or spectral inductive bias (ConvNet, Dilated ResNet, FNO, UNet) all achieve significant superiority, outperforming the low-fidelity solver they were trained on. In stark contrast, the vanilla Transformer, which lacks any inherent spatial or spectral bias and treats the input as a sequence of tokens, fails to achieve superiority and becomes progressively worse than the baseline solver.

Table 5: Achieved superiority for Burgers emulation. This extends the experiment of figure 6.

| Arch/Time Step | Superiority $\xi^{[t]}$ at Time Step | | | | | | | | | |
|---|---|---|---|---|---|---|---|---|---|---|
| | 1 | 2 | 3 | 4 | 5 | 6 | 7 | 8 | 9 | 10 |
| ConvNet | 1.00 | 0.97 | 0.87 | 0.82 | 0.80 | 0.80 | 0.80 | 0.81 | 0.81 | 0.81 |
| Dilated ResNet | 1.00 | 0.91 | 0.79 | 0.73 | 0.69 | 0.66 | 0.65 | 0.67 | 0.66 | 0.65 |
| FNO | 1.00 | 1.01 | 0.91 | 0.87 | 0.86 | 0.89 | 0.93 | 1.00 | 1.08 | 1.12 |
| Transformer | 1.00 | 1.06 | 1.13 | 1.24 | 1.35 | 1.45 | 1.52 | 1.58 | 1.63 | 1.68 |
| UNet | 1.00 | 0.90 | 0.86 | 0.88 | 0.91 | 0.96 | 1.00 | 1.04 | 1.07 | 1.11 |

