# OpenReview forum: "Neural Emulator Superiority: When Machine Learning for PDEs Surpasses its Training Data"
_NeurIPS.cc/2025/Conference — NeurIPS 2025 poster_

### Official Review · Reviewer_GMRy · 2025-06-25

**Clarity:** 3
**Significance:** 2
**Originality:** 3
**Rating:** 4
**Confidence:** 3

**Summary:**

This paper introduces and investigates "emulator superiority". An interesting phenomenon where neural emulators trained solely on low-fidelity numerical solver data can outperform these solvers when evaluated against higher-fidelity references. Through theoretical analysis on linear PDEs, the authors demonstrate that certain neural architectures' inductive biases enable better error accumulation or generalization than their training solvers. Empirical experiments extend this concept to nonlinear PDEs, showing superiority across different architectures.

**Questions:**

- The paper notes that superiority may disappear when training data is already high-fidelity. Can we quantify a fidelity threshold below which emulators have room to improve?
- If the core motivation of the work is to advocate for low-fidelity simulation-based emulators, it would be beneficial to include a discussion on the trade-offs in computational cost. Currently, there is limited analysis of the cost-benefit balance when using neural emulators, particularly considering that solving high-fidelity PDEs with neural models can involve significant training and inference costs, which may offset the claimed advantages.
- How robust are the closed-form superiority proofs for linear PDEs to perturbations in discretization, boundary conditions, or higher-dimensional settings? Are these proofs extensible beyond 1D periodic domains and synthetic solvers?
- Were baseline solvers used in training (e.g., Picard-truncated Burgers solver) intentionally made poor to create room for superiority? Could the superiority vanish if the low-fidelity solver were just marginally improved?
- Can you try “non-toy” PDE systems (e.g., Navier-Stokes)?
- To me, some of the training conditions (e.g., ICs from a single Fourier mode) are overly restrictive. How would the results change if training data sampled from more physically diverse regimes?

**Ethical Concerns:**

["NO or VERY MINOR ethics concerns only"]

**Limitations:**

yes

**Quality:**

2

**Strengths And Weaknesses:**

Strengths:
- One of the strong aspects of this paper is its theoretical analysis. The authors provide clear and detailed proofs in the Fourier domain demonstrating conditions under which the "emulator superiority" can arise for linear PDEs.
- The core concept of emulator superiority is interesting.

Weaknesses:
- Despite robust theoretical treatment for linear PDEs, the paper’s empirical experiments are somewhat limited to idealized, canonical PDE examples. While appropriate for proof-of-concept, the generalizability to real-world multi-physics simulations (e.g., climate, combustion, plasma physics) remains unexplored.
- The paper sticks to one-step supervised training. It acknowledges but does not explore other paradigms like unrolled training or physics-informed losses, which might enhance or inhibit superiority. This makes me wonder to what extent is “superiority” an artifact of the chosen loss metric, test regime, or evaluation horizon? Would the observed superiority still exist if evaluated on alternative physics-informed metrics?
- The superiority proofs rely on mode-isolated training and evaluation. For practical PDE problems, this may not be the case. Is the assumption of single-mode initial conditions meaningful in real scientific scenarios?

---

> ### Author Rebuttal · Authors · 2025-07-30
>
> We are deeply grateful to the reviewer for their thorough and critical review.
>
> ---
>
> > Despite robust theoretical treatment for linear PDEs, the paper’s empirical
> > experiments are somewhat limited to idealized, canonical PDE examples.
>
> We concur with the reviewer that extending this work to complex multi-physics
> simulations is a critical direction for future research. Our focus on canonical
> PDEs was a strategic choice, allowing us to isolate and rigorously analyze the
> foundational principles of emulator superiority. We argue that this approach has
> broader implications because many complex systems, like those governed by the
> Navier-Stokes equations, are fundamentally composed of the very dynamics we
> study: advection, diffusion, and Poisson-like solves. By demonstrating that
> superiority emerges within these core building blocks, we provide strong
> evidence that the phenomenon is a general principle of learning from numerical
> data, not an artifact of simple systems. Understanding these dynamics in
> isolation is a necessary foundation for investigating more complex scenarios
> where multiple error sources are coupled and confounded. We will highlight this
> in the revised version.
>
> > [...] to what extent is “superiority” an artifact of the chosen
> > loss metric, test regime, or evaluation horizon? Would the observed
> > superiority still exist if evaluated on alternative physics-informed metrics?
>
> We agree that the training paradigm is a critical factor. Our linear theoretical
> investigations hold equally for unrolled training and physics-informed training
> (if in the form of discrete residuals over two consecutive time steps). If the
> reviewer is interested, we'd be happy to provide the full proof, and we will add
> it to the appendix of the revised version.
>
> For the nonlinear problems, we also investigated with unrolled training and
> found that it oftentimes reduces the effect of superiority because the emulator
> is expected to replicate the error accumulation of the solver. This is desirable
> if the reference solver is good, but not if it is bad. Since there are
> additional axes to unrolled training (like its length, weighting, additional
> reference frames etc.), we decided to focus on the one-step training paradigm
> for this paper.
>
> The reviewer also raises an interesting point about training and
> evaluating on physics-informed (PI) metrics. While the equivalence of one-step
> supervised training and discrete residua cannot directly be shown for more
> complicated setups, we still believe that the discrete residuum consistent with
> a coarse solver will give rise to the same superiority effect. Note that our
> formal definition of Equation (5) requires solvers and does not directly
> translate to PI metrics.
>
> > The superiority proofs rely on mode-isolated training and evaluation. For
> > practical PDE problems, this may not be the case. Is the assumption of
> > single-mode initial conditions meaningful in real scientific scenarios?
>
> First, we wish to clarify that while our theoretical proofs utilize single-mode
> analysis for analytical tractability, a significant portion of our empirical
> results already incorporate multi-mode dynamics. The experiment in Figure 4c
> explicitly tests on five active modes, while the nonlinear dynamics in the
> Burgers' experiment (Figure 5) generate a rich, interactive spectrum from smooth
> initial data. The theoretical focus on a single mode is a standard diagnostic
> tool from numerical analysis, which allowed us to isolate the core mechanism of
> superiority with a degree of rigor and clarity that is often unattainable in
> more complex numerical settings.
>
> To further bridge this gap and directly address the reviewer's concern, we have
> conducted a new numerical experiment based on our linear theoretical setup. As
> detailed further in our response to Reviewer jfet, we trained a linear emulator
> on initial conditions containing various combinations of Fourier modes and
> tested its state-space superiority on an unseen higher mode. This study confirms
> that "forward superiority" is a robust phenomenon: as long as the emulator is
> trained on modes with lower frequencies than the test mode, it consistently
> achieves superiority ( $\xi<1$). The results also reveal an intuitive pattern
> where the magnitude of this effect diminishes as the training data becomes more
> spectrally diverse. By seeing the solver's error profile across a wider range of
> modes, the emulator begins to replicate those systematic flaws more closely.
>
> > The paper notes that superiority may disappear when training data is already
> > high-fidelity. Can we quantify a fidelity threshold below which emulators have
> > room to improve?
>
> This is a key question. While a universal, quantifiable threshold is likely
> problem-dependent, we can demonstrate the relationship between solver fidelity
> and the potential for superiority. We have conducted an ablation study for the
> number of Picard iterations in the Burgers example of section 4.2. We again used
> the UNet and display the superiority ratio $\xi^{[t]}$ rollout when trained and
> baselined against a Picard solver truncated after a certain number of
> iterations. A single iteration represents our original low-fidelity setup, while
> a higher number of iterations produces a more accurate, higher-fidelity solver.
>
> |Picard Iter/Time Step|1|2|3|4|5|6|7|8|
> |---:|---:|---:|---:|---:|---:|---:|---:|---:|
> |1 (Lowest Fidelity)|1|0.91|**0.85**|0.88|0.92|0.96|1|1.05|
> |2|1|1|0.97|0.98|1.01|1.09|1.16|1.25|
> |3|1|1.05|1.05|1.11|1.19|1.27|1.37|1.47|
> |4|1|1.36|1.34|1.46|1.53|1.58|1.67|1.84|
>
> These results clearly demonstrate our central thesis: the potential for
> superiority is directly related to the magnitude of the structured error in the
> training data. When the training solver is very coarse, there is significant
> room for the emulator to learn a more regularized operator, resulting in a
> strong superiority effect ($\xi$ ratio dropping to 0.85). As the fidelity of the
> training solver increases, the structured error in the training data decreases,
> and the potential for superiority vanishes. When trained on a reasonably
> converged solver, the emulator simply learns to replicate its already accurate
> behavior, and no superiority is observed.
>
> > If the core motivation of the work is to advocate for low-fidelity simulation-based emulators, it would be beneficial to include a discussion on the trade-offs in computational cost.
>
> We thank the reviewer for this point, as it allows us to clarify our motivation.
> Our work addresses the common scientific workflow where emulators are trained on
> computationally feasible but imperfect solver data. Rather than a simple
> cost-benefit analysis, our core finding is twofold. First, we show that an
> emulator can learn a more regularized operator and surpass the fidelity of its
> training data, effectively bridging the gap towards a more accurate reference.
> Second, this has critical implications for benchmarking, as a superior emulator
> may be unfairly penalized by a flawed reference, a paradox that urges more
> careful evaluation practices across the field. We will revise the introduction
> to make this practical context and these fundamental implications more explicit.
>
> > How robust are the closed-form superiority proofs for linear PDEs?
> > ... Are they extensible?
>
> - Higher Dimensions: Our proofs extend directly to n-dimensional periodic
>   domains (on equidistant rectangular grids), as the Fourier basis remains
>   separable.
> - Boundary Conditions: For non-periodic boundaries, the Fourier basis is no
>   longer applicable. One would need to use other spectral bases (e.g.,
>   Chebyshev), which do not diagonalize the convolution operator in the same
>   simple way, making a closed-form proof significantly more complex.
> - Discretization: Our analysis is at a fixed discretization. Changing it changes
>   the parameters and thus the magnitude of superiority, but the underlying
>   mechanism remains.
>
> In summary, we believe our theoretical analysis is already presented in the most
> general setting that still permits a rigorous, closed-form proof. Beyond these
> conditions, numerical experiments are more appropriate.
>
> > Were baseline solvers used in training (e.g., Picard-truncated Burgers solver)
> > intentionally made poor to create room for superiority? Could the superiority
> > vanish if the low-fidelity solver were just marginally improved?
>
> The solvers were chosen because they have clear, well-understood error
> characteristics (e.g., the known numerical diffusion of an implicit scheme or
> the incorrect shock movement of truncated nonlinear solves). This is essential
> for transparent analysis and not unrealistic; for example, truncating nonlinear
> solves is common practice in algorithms like PISO for incompressible flow (see
> also (Turek, 1999) for a further discussion). As noted in our response to your
> previous question, superiority does indeed diminish as the solver is improved.
> This is an expected and central part of our thesis: the potential for
> improvement is directly related to the magnitude of the structured error in the
> training data.
>
> > Can you try “non-toy” PDE systems (e.g., Navier-Stokes)?
>
> Thank you for this suggestion. Extending our analysis to more complex PDE
> systems such as Navier-Stokes is an exciting direction for future work, but unfortunately beyond the scope of the rebuttal.
>
> > How would the results change if training data sampled
> > from more physically diverse regimes?
>
> This is an important point we addressed in our reply to a previous point with
> the experiment of mixed mode in the training dataset. We re-emphasize that our
> key nonlinear experiments already use multi-mode data for testing and/or involve
> spectrally rich dynamics. A difference between the training and testing
> distributions is a necessary precondition for state-space superiority, which is
> one of the phenomena we aim to characterize.

---

> > ### Comment · Reviewer_GMRy · 2025-08-04
> >
> > I’ve reviewed the responses and am good with most of the explanations, particularly those concerning the single-mode analysis. However, I remain unsure how transferable this phenomenon is to more complex systems. Nonetheless, it’s an interesting study, and I’ll keep my score as is. Good luck!

---

> > > ### Author Response · Authors · 2025-08-05
> > >
> > > We thank you for you your reply.

---

### Official Review · Reviewer_jfet · 2025-07-01

**Clarity:** 3
**Significance:** 2
**Originality:** 4
**Rating:** 4
**Confidence:** 4

**Summary:**

Authors investigate when a learned PDE solver can be more accurate than the classical PDE solver used to generate training data.

Theoretical analysis performed for three linear PDEs (convection, diffusion, stationary diffusion) and linear emulators revealed that learned solver can be more accurate if distributions of train and test data do not match.

For the nonlinear case the improved accuracy of learned solver was observed for nonlinear emulators (FNO, Dilated ResNet, Transformer, etc) on convection and Burgers equations for the case of identical distributions for train and test data.

**Questions:**

1. **Assumptions in the proof of theoretical results**

   Authors assumed that the training data come from a very specific distribution that covers only a single mode. In this case it is clear (not explained explicitly) that the learned model will depend on the train set only from the mode number and both amplitude and phase are irrelevant. I have several questions on that:

   a. Can the authors please supply missing proof that both amplitude and phase are irrelevant? I would suggest showing that equation (8) can be achieved when the usual loss function is considered, e.g., mean (with respect to distribution on phase and amplitude) $L_2$ error.

   b. The whole formulation seems rather artificial with a weak dependence on training data. Can the authors please defend their choice of learning problem? Why is this simple scheme sufficient and interesting for operator learning?

   c. Since authors work with linear models, it seems to be possible to obtain closed-form solutions for arbitrary data. Can the authors comment on the possible extension of their results on more realistic data distributions (e.g., more than one mode)?

2. **Theoretical results are trivial from the standpoint of backward error analysis**

   Solution for PDE can be considered as an evaluation problem $y = \phi(x)$, where $x$ is input data, $y$ is solution to PDE and $\phi$ is an unknown map that brings input data to the solution. The approximate method that solves PDE is some other map $\hat{\phi}(x)$ that approximates $\phi(x)$ sufficiently well. Standard forward error used by authors is $\left\|\phi(x)-\hat{\phi}(x)\right\|$, where $\left\|\cdot\right\|$ is a suitable norm.

   Backward error is based on an entirely different idea. Consider perturbation of input data $\Delta x$ such that $\hat{\phi}(x) = \phi(x + \Delta x)$. In case this perturbation is possible to find $\hat{\phi}(x)$ can be considered as an *exact* solver for *wrong* input data. For solver $\hat{\phi}$ backward error is thus $\left\|\Delta x\right\|$ (here $\left\|\cdot\right\|$ is potentially different norm) and is related to forward error by condition number of the problem.

   If both $\phi(x)$ and $\hat{\phi}(x)$ are continuous one may expect that one can find input $\widetilde{x}$ near $x + \Delta x$ such that $\left\|\phi(\widetilde{x}) - \hat{\phi}(\widetilde{x})\right\| \leq \epsilon$ for small $\epsilon$. Such inputs can be found by solving $\arg\min_{x}\left\|\phi(x) - \hat{\phi}(x)\right\|$ (at least locally near $x$).

   Authors considered two different solution methods for the same problem $\phi_1(x)$, $\phi_2(x)$. These methods in general have different backward errors $\Delta x_1(x)$ and $\Delta x_2(x)$. Because of that these methods also have different solutions for the optimisation problem above $x_1 = \arg\min_{x}\left\|\phi(x) - \phi_1(x)\right\|\neq x_2 = \arg\min_{x}\left\|\phi(x) - \phi_2(x)\right\|$.

   These observations do not look too interesting or surprising. And the main problem is we deliberately try to find inputs such that one method is superior to the other one. Several questions are in order:

   a. Can the authors please comment on the considerations on backward error analysis?

   b. On the higher level authors simply note that different methods are good on different inputs. This observation is well known in the literature on numerical methods. For example, trapezoidal rules have exponential convergence for periodic functions and are expected to give very small errors in this case, but for non-periodic functions the error will increase. Given that it is easy to find functions for which trapezoidal rule is superior to, say, Simpson's rule and vice versa. Can the authors explain what made their result conceptually different?

   c. There is a clear parallel between adversarial examples for DL classification and inputs that lead to superiority. Did authors try to consider this parallel? What authors think in general on this relation?

3. **Empirical results for Burgers equation are not explained by theory**

   In my view the results presented for Burgers equation is the most surprising one. It is surprising because superiority shows up in the same solution curve, still out of distribution but not completely unrelated to the training data.

   a. It does not seem to me that theoretical results explain this behaviour. Can authors comment on whether a linear case is related to nonlinear one?

   b. Can this result be replicated for other equations (e.g., KS, KdV) and architectures (e.g., FNO, ResNet)?

   c. Do these results survive when one considers more diverse training sets?

In conclusion, I find questions raised by the author interesting and potentially fruitful for further study. Unfortunately, I do not find answers that authors provide to be useful or particularly interesting. Being said that I remain open to change my opinion after the rebuttal. I encourage authors to defend their position, supply more results and continue research on the topic.

**Ethical Concerns:**

["NO or VERY MINOR ethics concerns only"]

**Final Justification:**

To summarise, my opinion does not change after reading other reviews and rebuttal by authors:

1. The question authors ask is genuinely interesting
2. Available theoretical results are not well-developed
3. Numerical results are not convincing

**Limitations:**

yes

**Quality:**

2

**Strengths And Weaknesses:**

**Strengths:**
1. Authors ask genuinely interesting question
2. Both theoretical and empirical considerations are available

**Weaknesses:**
1. Both theoretical and empirical results are not convincing
2. The setup authors consider is far from establish practices

I provide more details in the section below.

---

> ### Author Rebuttal · Authors · 2025-07-30
>
> We sincerely thank the reviewer for their deep and critical engagement with our
> work.
>
> ---
>
> > - Both theoretical and empirical results are not convincing
> > - The setup authors consider is far from established practices
>
> We understand the reviewer's concerns and will address the specific points that
> led to this conclusion in the detailed sections below. We believe our setup,
> while simplified for analytical tractability, is designed to isolate a
> fundamental phenomenon relevant to the established practice of using imperfect
> simulation data to train emulators, a common scenario in resource-constrained
> scientific and engineering domains.
>
> > Can the authors please supply missing proof that both amplitude and phase are
> > irrelevant? [...]
>
> This is an excellent point. The proof was omitted for brevity, but we agree it
> is essential for the rigor of our argument. The key insight is that for a linear
> convolutional ansatz and a Mean-Squared Error (MSE) loss, the optimization
> problem diagonalizes in Fourier space. *If only a single mode is present*, the
> expectation over the data becomes a constant factor (i.e., the distribution over
> phase and amplitude is irrelevant as long as it is nonzero) and the minimizer is
> found by the first-order condition, which leads directly to Equation (8). We
> will add the derivation to the appendix of the revised pdf and are happy to
> provide it as a discussion reply if the reviewer is interested.
>
> > The whole formulation seems rather artificial with a weak dependence on
> > training data. Can the authors please defend their choice of learning problem?
> > Why is this simple scheme sufficient and interesting for operator learning?
>
> We acknowledge that the single-mode training setup is a simplification. This
> choice was deliberate and serves two purposes. This controlled setting is what
> allows us to derive a data-independent closed-form analytical solution for the
> trained emulator. This provides a clear, interpretable, and undeniable
> demonstration of the superiority principle without confounding numerical
> factors. Moreover, probing a system's behavior with single Fourier modes is a
> foundational technique in linear systems theory and numerical analysis. It
> allows for a precise characterization of a system's response across different
> frequencies.
>
> Our "simple scheme" is therefore sufficient and interesting because it allows us
> to *isolate and prove* the existence of a counter-intuitive phenomenon. This
> provides the foundational insight needed to understand why superiority might
> occur in more complex, realistic operator learning scenarios where analytical
> treatment is impossible.
>
> > Since authors work with linear models, it seems to be possible to obtain
> > closed-form solutions for arbitrary data. Can the authors comment on the
> > possible extension of their results on more realistic data distributions
> > (e.g., more than one mode)?
>
> If the training data contains multiple modes, the closed-form solution for the
> emulator parameters is no longer a simple equation for a single mode. Instead,
> it becomes a linear least-squares weighted by the concrete distribution of
> complex-values coefficients. While an analytical solution still exists for this
> linear regression problem, the analysis becomes more complex because of this
> data dependency.
>
> To illustrate this, we redid the theoretical experiments numerically (training
> on a combination of modes and only testing on mode 5). We found that there is
> still superiority when more modes are present in the training initial condition
> as long as they are lower than what is being tested on (confirming again what we
> called "forward superiority" in line 202 of the original manuscript).
>
> |Modes in Train IC Dist|Superiority $\xi^{[1]}$|
> |:---|:---|
> |1|0.57|
> |1,2|0.61|
> |1,2,3,4|0.72|
> |1,2,3,4,5|0.81|
> |1,2,3,4,5,6|0.87|
> |1,2,3,4,5,6,7|1.0|
> |1-10|1.1|
> |1-20|2.3|
> |4,6|1.1|
> |3,4|0.79|
>
> > Can the authors please comment on the considerations on backward error
> > analysis?
>
> We thank the reviewer for bringing up this insightful perspective. We agree that
> from a backward error analysis standpoint, our low-fidelity solver and our
> learned emulator can be viewed as exact solvers for two slightly different,
> perturbed problems. However, this perspective, while valid, misses the central
> and most novel aspect of our work: the learning dynamic itself.
>
> Our contribution is not merely to show that two different numerical methods
> exist and have different error properties. Our key finding is that we can learn
> a new method (the emulator) by only observing an existing, inferior method (the
> low-fidelity solver), and this new method can provably be closer to a
> high-fidelity reference. Standard backward error analysis compares pre-existing
> methods; it does not account for the process of creating a new, superior method
> from an imperfect one, which is the core of our insights.
>
> >  ...authors simply note that different methods are good on different inputs.
> This observation is well known... Given that it is easy to find functions for
> which trapezoidal rule is superior to, say, Simpson's rule and vice versa. Can
> the authors explain what made their result conceptually different?
>
> The conceptual difference lies in the origin of the superior method. The analogy
> of choosing between the Trapezoidal and Simpson's rules involves selecting from
> a set of pre-existing, analytically derived methods. A user knowingly picks the
> right tool for the job.
>
> In our work, the superior method (the emulator) is not analytically pre-defined.
> It is discovered through a data-driven learning process that has access only to
> data from an inferior method. The novelty is in demonstrating that this learning
> process can implicitly correct the known, structured flaws of its training data
> source, a mechanism different from selecting an existing quadrature rule.
>
> > There is a clear parallel between adversarial examples for DL classification
> > and inputs that lead to superiority. Did authors try to consider this
> > parallel? What authors think in general on this relation?
>
> Both phenomena reveal the "fragility" and specificity of what a neural network
> learns. An adversarial example exploits a model's learned decision boundary to
> cause a misclassification, often via a perceptually minor perturbation. This is
> typically viewed as a failure mode. A superiority-inducing input, as we've
> framed it, exploits the difference between the emulator's learned dynamics and
> the solver's flawed dynamics. However, we see this as a constructive outcome.
> The emulator is "fragile" to the specific errors of the solver, and instead of
> replicating them, it learns a more physically regularized behavior. One could
> view our work as identifying a regime where a model's sensitivity to
> out-of-distribution data does **not** lead to a nonsensical output, but to a
> physically more plausible one.
>
> > It does not seem to me that theoretical results explain this behaviour. Can
> > authors comment on whether a linear case is related to nonlinear one?
>
> The theoretical results provide the conceptual blueprint. The key insight from
> our linear analysis is that an emulator with a favorable inductive bias can
> achieve superior spectral generalization; it can learn to handle high-frequency
> modes more accurately even when only trained on low-frequency data.
>
> This is also what we observe in the Burgers' experiment. The low-fidelity solver
> (truncated Picard iteration) makes significant errors in propagating the
> high-frequency components that constitute the sharp shock front. The UNet is
> trained only on the first two frames, which are dominated by low-frequency
> content. To succeed at the multi-step rollout, the UNet must learn an operator
> that correctly handles the emergent high-frequency modes of the shock. We
> believe that its ability to do so, outperforming its training data, is a direct
> nonlinear analogue of the spectral generalization proven in our linear theory.
>
> > Can this result be replicated for other equations (e.g., KS, KdV) and
> > architectures (e.g., FNO, ResNet)?
>
> **Architectures:** Yes, the superiority phenomenon on the Burgers' equation is
> not unique to the UNet. As detailed in our response to Reviewer K6sC, we
> repeated this experiment for all architectures from Section 4.1. The key finding
> from this new experiment is that architectures with favorable inductive
> biases—namely ConvNet, Dilated ResNet, FNO, and the UNet—all achieve significant
> periods of superiority (superiority ratio < 1). This demonstrates the generality
> of the effect. Conversely, the vanilla Transformer, which lacks a strong spatial
> or spectral prior, fails to achieve superiority. This strongly suggests that a
> suitable inductive bias is a more critical ingredient for this phenomenon than
> raw model capacity. We hypothesize that modern Transformer variants
> incorporating convolutional-like structures (e.g., Swin Transformers) would
> likely exhibit superiority as well.
>
> **Equations:** Our study was intentionally designed to analyze the fundamental
> "building blocks" of more complex physical systems. Many sophisticated problems,
> such as the Navier-Stokes equations in fluid dynamics, are governed by a
> combination of simpler physical processes. Our paper provides the foundational
> understanding, and such cases represent very interesting directions for future
> work.
>
> **Regarding the specific examples:** We hypothesize that the Korteweg-de Vries
> (KdV) equation, which develops rich, non-chaotic spectral content (solitons), is
> a strong candidate for exhibiting similar superiority as in the Burgers
> (primarily adding a third-order derivative). Chaotic systems like the KS
> equation present more complex challenges where regions with superiority may be
> harder to identify.
>
> > Do these results survive when one considers more diverse training sets?
>
> This is a key question for future work; the new results above give first
> indicators for generalization to diverse training sets.

---

> > ### Comment · Reviewer_jfet · 2025-08-04
> >
> > ```
> > To illustrate this, we redid the theoretical experiments numerically (training on a combination of modes and only testing on mode 5). We found that there is still superiority when more modes are present in the training initial condition as long as they are lower than what is being tested on (confirming again what we called "forward superiority" in line 202 of the original manuscript).
> > ```
> > I would like to thank the authors for this additional experiment. It is interesting that superiority is observed for more complex training data. The results indicate theoretical claims of authors can be strengthened. A more general theorem on superiority of learned emulator for a wide class of numerical integrator could be a valuable contribution.
> >
> > ```
> > Our contribution is not merely to show that two different numerical methods exist and have different error properties. Our key finding is that we can learn a new method (the emulator) by only observing an existing, inferior method (the low-fidelity solver), and this new method can provably be closer to a high-fidelity reference.
> >
> > ...
> >
> > In our work, the superior method (the emulator) is not analytically pre-defined. It is discovered through a data-driven learning process that has access only to data from an inferior method. The novelty is in demonstrating that this learning process can implicitly correct the known, structured flaws of its training data source, a mechanism different from selecting an existing quadrature rule.
> > ```
> > But what is a learning process here? What authors consider theoretically is a solution to optimisation problem with quadratic loss also known as ordinary least squares.
> >
> > I believe I can do the same with trapezoidal and Simpson's rule. Both of those methods are examples of (closed) Newton-Cotes formulas. Newton-Cotes formulas of order $n$ can integrate exactly polynomial up to $n$. Authors considered "trainable" emulators from the class of circulant matrices. I am going to consider Toeplitz matrix defined by local rules $\alpha f(x_i) + \beta f(x_{i+1}) + \gamma f(x_{i+2}) + \dots$. Clearly, both Simpson's and trapezoidal rule fit this description. Now I know that Simpson's rule provide exact result for polynomials of order up to $2,$ so I use polynomials of order $1$ as training data and integrate them with Simpson's rule. Since these are polynomials of order $1$ both rules will integrate them exactly. My ground truth data is exact solution and I consider a OLS problem:
> >
> > $$\mathbb{E}\_{f=a + b x; a,b\sim N(0, 1)}\left[\left(a + \frac{b}{2} - \alpha f(0) - \beta f(1)\right)^2\right]=\mathbb{E}\_{f=a + b x; a,b\sim N(0, 1)}\left[\left((1-\alpha-\beta)a + \left(\frac{1}{2}-\beta\right)b\right)^2\right] = (1-\alpha-\beta)^2 + \left(\frac{1}{2}-\beta\right)^2$$
> >
> > When we minimise this with respect to $a$ and $b$ we recover $a = b = \frac{1}{2}$ which is a trapezoidal rule. Now I apply this "learned" rule to the periodic functions and obtain superiority the same way as authors did.
> >
> > In summary, I do not think that "learning" add a lot of novelty to the picture.
> >
> > ```
> > One could view our work as identifying a regime where a model's sensitivity to out-of-distribution data does not lead to a nonsensical output, but to a physically more plausible one.
> > ```
> > I agree that adversarial examples in DL are interesting because they show how brittle is the performance of deep learning system. My point is adversarial examples are found by optimising the input under constraints of small perturbations to obtain a large perturbation of the output. Authors of this work are doing precisely the same, but without constraints on the magnitude of perturbation, and with superiority as an optimisation target.
> >
> > To summarise, my opinion does not change after reading other reviews and rebuttal by authors:
> > 1. The question authors ask is genuinely interesting
> > 2. Available theoretical results are not well-developed
> > 3. Numerical results are not convincing

---

> > > ### Author Response · Authors · 2025-08-05
> > >
> > > > A more general theorem on superiority of learned emulator for a wide class of
> > > > numerical integrator could be a valuable contribution.
> > >
> > > We thank the reviewer for noting the value of our new numerical results on
> > > multi-mode training data and for the suggestion of a more general theorem. We
> > > agree this is a valuable direction for future research. The primary goal of our
> > > current theoretical analysis was to isolate the core mechanism of superiority in
> > > the most clear and undeniable setting possible (i.e., the single-mode case),
> > > free from the confounding factor of specific energy distributions across modes.
> > >
> > > To strengthen our paper and demonstrate the robustness of the principle, we will
> > > incorporate the new multi-mode numerical experiment (from our initial rebuttal)
> > > into the appendix of the revised manuscript. This will provide concrete evidence
> > > that the phenomenon of "forward superiority" is not an artifact of the
> > > simplified single-mode setup.
> > >
> > > > [on the analogy to Newton-Cotes rules]
> > >
> > > We appreciate the reviewer's detailed construction of the analogy. However, it
> > > differs from our work in three critical ways. First, as we noted previously, is
> > > the **inverted hierarchy of quality**: our work demonstrates learning a superior
> > > method from an inferior one, a counter-intuitive result not captured by learning
> > > a lower-order method from a higher-order one.
> > >
> > > The second, and perhaps most crucial, distinction lies in the **evaluation
> > > context**. The reviewer's analogy demonstrates superiority by training on one
> > > class of functions (polynomials, for which Simpson's rule is well-suited) and
> > > then evaluating on a different class (smooth periodic functions,
> > > where the Trapezoidal rule is known to be exceptionally accurate). Our work, in
> > > contrast, demonstrates superiority *within the same, consistent problem domain*.
> > > We train our emulator on data from the advection equation with periodic initial
> > > conditions, and we evaluate it on the very same task.
> > >
> > > This leads to the third distinction: the **role of imperfect data during
> > > training**. In the reviewer's construction, both the "teacher" (Simpson's rule)
> > > and the "student" (Trapezoidal rule) are perfect on the training data (1st-order
> > > polynomials). In our work, the situation is different. Both the low-fidelity
> > > solver and the emulator's ansatz are *imperfect* at representing the true
> > > dynamics of the single Fourier mode used for training (which we discuss in
> > > Figure 1). Therefore, it is not a simple recovery of an exact method; it is an
> > > optimization to find the best possible compromise—a new operator that best fits
> > > the flawed training data according to the loss function.
> > >
> > > Nonetheless, the example with an analogy to quadrature methods is an interesting one that we'd be happy to add to our manuscript to clarify the subtleties of our approach.
> > >
> > > > [On the relation to adversarial examples] Authors of this work are doing
> > > > precisely the same, but without constraints on the magnitude of perturbation,
> > > > and with superiority as an optimisation target.
> > >
> > > We'd like to clarify an important misunderstanding regarding our methodology and its
> > > relation to adversarial examples. The reviewer's claim that we use "superiority
> > > as an optimisation target" is factually incorrect. Our training objective, as
> > > defined in Equation (3), is exclusively to minimize the one-step prediction
> > > error between the emulator and the low-fidelity solver. Superiority is an
> > > emergent property measured during a separate evaluation phase (Equation (5)); it
> > > is never part of the training loss function. While we do test our linear models
> > > on out-of-distribution inputs, this is not an "optimization" to find fragile
> > > inputs, but rather a systematic analysis across Fourier modes—a standard
> > > diagnostic technique to characterize a system's frequency response. More
> > > importantly, this interpretation does not apply to the second half of our
> > > findings. In our experiments on autoregressive superiority (Figures 4a, 4b) and
> > > the nonlinear Burgers' equation (Figure 5), the training and testing initial
> > > conditions are drawn from the exact same distribution. Here, superiority arises
> > > not from a contrived input but because the emulator has learned dynamics that
> > > accumulate errors more favorably over long-term, multi-step rollouts. This
> > > demonstrates a fundamentally different and more robust mechanism than simple
> > > input sensitivity. That said, the reviewer raises an interesting prospect for
> > > future work: actively optimizing for inputs that maximize superiority for a
> > > trained emulator could be a powerful tool for analyzing model robustness, and we
> > > will add this to our discussion of future directions.

---

> ### Comment · Reviewer_jfet · 2025-08-05
>
> ```
> First, as we noted previously, is the inverted hierarchy of quality: our work demonstrates learning a superior method from an inferior one, a counter-intuitive result not captured by learning a lower-order method from a higher-order one.
> ```
> "Low order" and "hight order" are just the way to classify methods that is not always helpful. The practical efficiency of the method can have nothing to do with order as we have already seen with Simpson's rule and trapesoidal rule.
>
> ```
> The second, and perhaps most crucial, distinction lies in the evaluation context. The reviewer's analogy demonstrates superiority by training on one class of functions (polynomials, for which Simpson's rule is well-suited) and then evaluating on a different class (smooth periodic functions, where the Trapezoidal rule is known to be exceptionally accurate).
> ```
> The same can be said about the setup suggested by authors: they train on band-limited functions with largest frequency $a$ and evaluate on band-limited functions with larges frequency $b$. Besides that I do not need periodic functions in my example. It is enough to have functions approximated by smooth piecewise polynomials (e.g., splines) that are concentrated within the interval.
>
> ```
> This leads to the third distinction: the role of imperfect data during training. In the reviewer's construction, both the "teacher" (Simpson's rule) and the "student" (Trapezoidal rule) are perfect on the training data (1st-order polynomials).
> ```
> That also can be addressed with my construction. I chose exact data deliberately to recover unmodified trapezoidal rule.
>
> ```
> We'd like to clarify an important misunderstanding regarding our methodology and its relation to adversarial examples. The reviewer's claim that we use "superiority as an optimisation target" is factually incorrect. Our training objective, as defined in Equation (3), is exclusively to minimize the one-step prediction error between the emulator and the low-fidelity solver.
> ```
> I believe I understand in sufficient details the construction proposed by authors. My claim is not that authors directly optimise for superiority. In place of that they indirectly optimise for superiority by intentionally focusing on the examples where learned method is superior to the base one, used to generate training data. In this sense there is a selection for "adversarial inputs." Of course this is not a problem in itself since authors do not claim that the learned method is universally better. However, the general filling of the approach is that authors suggest to find some inputs that gives better results for learned method.
>
> ---
>
> In my view our disagreement is largely on subjective points. I claim that the results are not interesting enough, but do not challenge the technical validity of the contribution.
>
> Given that other reviewers expressed their interest in the result, I will rise my score and will hope that AC and meta reviewers provide extra input that will ultimately help to decide whether the paper is suitable for the publication or not.

---

> > ### Author Response · Authors · 2025-08-06
> >
> > We sincerely thank the reviewer for their continued, thoughtful engagement, for confirming the technical validity of our work, and for raising their score.
> >
> > The critical dialogue has been invaluable for helping us sharpen the manuscript's narrative. In particular, we found the framing of our results through the critical lens of adversarial examples to be an insightful perspective that we did not initially consider. We will ensure this viewpoint is discussed in the revised manuscript to provide a more nuanced context for our results.

---

### Official Review · Reviewer_8yEP · 2025-07-02

**Clarity:** 3
**Significance:** 3
**Originality:** 3
**Rating:** 5
**Confidence:** 4

**Summary:**

This paper outlines the idea of emulator superiority, where a neural surrogate trained on low fidelity solver data is able to outperform said low-fidelity solver when evaluated on a high fidelity data. The paper also provides some theoretical analysis showing that a linear emulator in limited settings can have lower error than implicit and explicit numerical schemes in comparison to an analytical solution.

**Questions:**

1. In the nonlinear PDE experiments, isn't it possible that the reason for neural network superiority on certain timescales is due to the ability of neural networks to smoothly interpolate, while the static low fidelity solver cannot do this?

2. In the case of the advection equation, the narrative says "Our theoretical analysis reveals several crucial conditions for achieving superiority. First, we must train a low-capacity linear ansatz on a high-capacity (non-analytical) reference method." How does this assumption fit with neural network architectures when applied to real data? It is often the case that modern neural networks are high capacity, so doesn't this mean that in many practical scenarios, neural network superiority will not be achieved?

3. In Figure 1, there are labels called "Amplifying" and "Dampening". Does "Amplifying" mean that the scale of the solution is increasing or diverging? And does "Dampening" mean the solution scale is decreasing?

4. For the Burger's equation experiment, it is stated that "for this experiment, we train a vanilla UNet (Ronneberger et al., 2015) on only the initial "smooth" frames from this low-fidelity solver." Does that mean that the UNet is trained with a limited time-series rollout from the low fidelity solver and then during the test phase, it is made to do a longer rollout than it was trained with?

**Ethical Concerns:**

["NO or VERY MINOR ethics concerns only"]

**Final Justification:**

In light of some of the clarifications offered by the authors during the rebuttal phase, I believe the question answered by the paper regarding "emulator superiority" and corresponding theoretical insights will serve the scientific machine learning community well and hence I will raise my score to `Accept'.

**Limitations:**

yes

**Paper Formatting Concerns:**

1. Certain figure captions and text refer to subfigures with letters (a), (b), etc. However the corresponding figures/plots do not have any alphabet labels (e.g., Figure 3, 5). I suggest the authors improve the plots to include said labels.

2. Some equations in the appendix are too long and overflow the margin, for example Equation 41 on page 16.

**Quality:**

3

**Strengths And Weaknesses:**

# Strengths:
1. The authors do a good job of formalizing their problem as well as describing the different types of generalization that are important for neural emulators applied to differential equations.
2. The paper has useful theoretical results with closed form solutions for linear PDEs using a linear parametric ansatz.

# Weaknesses:
1. While the theoretical results for linear PDEs are good, the results for the nonlinear neural architectures are mixed, especially in the case of state-space superiority.

2. The practical applicability of this paper is limited because of the problem setup. If a high fidelity simulation data is available, it is unlikely that a researcher would train a neural network with low fidelity data and then evaluate it on high fidelity data. More clarity on when exactly the proposed problem setting is valid is necessary.

---

> ### Author Rebuttal · Authors · 2025-07-30
>
> We sincerely thank the reviewer for the valuable feedback. We were glad to hear
> the problem formalization and theoretical results to be considered a key strength
> of the paper. It was our goal to establish a clear, formal foundation for
> "emulator superiority" and to ground it in rigorous theoretical analysis, and
> happy that this was recognized. We appreciate the opportunity to address their
> questions and concerns.
>
> ---
>
> > While the theoretical results for linear PDEs are good, the results for the
> > nonlinear neural architectures are mixed, especially in the case of
> > state-space superiority.
>
> This is a key observation that we believe reinforces, rather than contradicts,
> our central thesis. The "mixed" results are a direct consequence of the
> conditions for superiority we outline, and our aim was to provide a realistic
> assessment: Superiority arises from an interplay
> between the emulator's inductive biases and the solver's error characteristics,
> and as such cannot be taken for granted.
>
> Our results in Figure 4c show that high-capacity architectures with global
> receptive fields (FNO, Transformer) struggle to achieve state-space superiority.
> We hypothesize this is because their powerful representational capacity allows
> them to "overfit" to the low-fidelity solver, perfectly replicating its
> dynamics—including its systematic errors.
>
> In contrast, the ConvNet, whose local inductive bias is a poorer match for the
> global dynamics of the solver, is forced to learn a more regularized and
> physically plausible local operator. This "disadvantage" in representational
> power becomes an advantage for generalization, leading to superiority. The mixed
> results are therefore not a weakness of the experiment, but rather an empirical
> validation of the conditions required for superiority to emerge.
>
> Please also note the additional experiments on the Burgers equation we present
> below as answer to your second question.
>
> > The practical applicability of this paper is limited because of the problem
> > setup. If a high fidelity simulation data is available, it is unlikely that a
> > researcher would train a neural network with low fidelity data and then
> > evaluate it on high fidelity data. More clarity on when exactly the proposed
> > problem setting is valid is necessary.
>
> We agree that clarifying the practical context is crucial. Indeed, if enough
> high-fidelity (HF) data is available, it is preferable to train on it or to
> cleverly mix it with excessively available low-fidelity (LF) data.
>
> However, there are many fields of science and engineering in which high-fidelity
> data is a scarce "validation" resource, e.g., climate science, and turbulence
> modeling. A researcher may only be able to afford a handful of HF runs to serve
> as a "gold standard" for validation. In contrast, they can generate massive
> datasets using computationally cheap, low-fidelity (LF) solvers. One can imagine
> emulators to be trained on this large LF dataset. Our work demonstrates the
> surprising outcome that in certain cases such an emulator can learn dynamics that
> are actually more accurate than the training data suggests, potentially
> approaching the quality of an expensive HF reference.
>
> More broadly, our work highlights that (almost) all numerical solvers are
> imperfect data generators (that being "high-fidelity" is not a binary attribute). Our
> framework provides a lens through which to understand how neural networks
> interact with the inherent, structured errors present in any simulation data,
> which is a fundamental question for scientific machine learning.
>
> Ultimately, the fairness of benchmarking an emulator depends critically on the
> quality of the reference data: If the reference trajectories used for evaluation
> contain structured errors, and the neural emulator happens to learn an operator
> that systematically corrects or improves upon these errors, the benchmarking
> process may underestimate the emulator’s true performance. In such cases, the
> emulator could be penalized for being more accurate than the reference, leading
> to an unfair assessment. This is something that is universally applicable and
> might even appear in settings where the reference data is high-fidelity.
>
> We will add a paragraph to the introduction to make these practical scenarios
> more explicit.
>
> > In the nonlinear PDE experiments, isn't it possible that the reason for neural
> > network superiority on certain timescales is due to the ability of neural
> > networks to smoothly interpolate, while the static low fidelity solver cannot
> > do this?
>
> Our interpretation as to why we see autoregressive superiority together with
> state-space superiority is because the emulators (by virtue of their inductive
> biases) learn a better generalization to higher frequency modes than their
> low-fidelity training data generator. Actually, correctly predicting shocks for
> the Burgers would not require smooth interpolation.
>
> > In the case of the advection equation, the narrative says "Our theoretical
> > analysis reveals several crucial conditions for achieving superiority. First,
> > we must train a low-capacity linear ansatz on a high-capacity (non-analytical)
> > reference method." How does this assumption fit with neural network
> > architectures when applied to real data? It is often the case that modern
> > neural networks are high capacity, so doesn't this mean that in many practical
> > scenarios, neural network superiority will not be achieved?
>
> The reviewer's intuition aligns with results of the experiment in Figure 4c (for
> which nonlinear emulators are tested to generalize from training on only mode 1
> to testing on mode 5). Only the ConvNet with low capacity but strong inductive
> bias shows (autoregressive and) state-space superiority. However, this is only
> for the advection equation which does not move any energy between the modes. The
> Burgers equation does this and we see that the fairly high-capacity UNet is able
> to achieve superiority on it.
>
> To provide a more complete picture, we extended our analysis of the nonlinear
> Burgers' equation experiment (Section 4.2) to include all architectures from our
> study. The results robustly confirm that the superiority phenomenon is not
> limited to a single architecture but is instead closely tied to the inductive
> biases of the models. The key finding is that architectures with a spatial or
> spectral inductive bias (ConvNet, Dilated ResNet, FNO, UNet) all achieve
> significant superiority, outperforming the low-fidelity solver they were trained
> on. The Dilated ResNet is a standout performer, achieving a remarkable
> superiority ratio of 0.65. In stark contrast, the vanilla Transformer, which
> lacks any inherent spatial or spectral bias and treats the input as an
> unstructured set of tokens, fails to achieve superiority and becomes
> progressively worse than the baseline solver.
>
> Below are the median superiority ratios ($\xi^{[t]}$) over 30 random seeds for
> each architecture.
>
> |Network/Time Step|1|2|3|4|5|6|7|8|9|10|
> |:------------------|--:|--:|--:|--:|--:|--:|--:|--:|--:|--:|
> |ConvNet|1|0.97|0.87|0.82|0.8|0.8|0.8|0.81|0.81|0.81|
> |Dilated ResNet|1|0.91|0.79|0.73|0.69|0.66|0.65|0.67|0.66|0.65|
> |FNO|1|1.01|0.91|0.87|0.86|0.89|0.93|1|1.08|1.12|
> |Transformer|1|1.06|1.13|1.24|1.35|1.45|1.52|1.58|1.63|1.68|
> |UNet|1|0.9|0.86|0.88|0.91|0.96|1|1.04|1.07|1.11|
>
> It should be noted that modern Transformer variants like SWin resemble a
> convolution-like inductive bias and could thus again show superiority.
>
> >  In Figure 1, there are labels called "Amplifying" and "Dampening". Does
> >  "Amplifying" mean that the scale of the solution is increasing or diverging?
> >  And does "Dampening" mean the solution scale is decreasing?
>
> Yes, the reviewer's interpretation is correct. These labels describe the effect
> of a numerical scheme on the magnitude of each Fourier mode's complex-valued
> coefficient. "Dampening" means the magnitude of a mode is reduced
> ($|\hat{u}_k^{[t+1]}| < |\hat{u}_k^{[t]}|$ for mode $k$), which corresponds to
> diffusion. "Amplifying" means the magnitude is increased ($|\hat{u}_k^{[t+1]}| >
> |\hat{u}_k^{[t]}|$). For a purely hyperbolic problem like advection, any
> modification of the magnitude is a numerical artifact. While unphysical,
> numerical diffusion is often unavoidable and preferred over the opposite. If the
> magnitude increases, it typically leads to instability (after multiple steps,
> the solution diverges). This occurs for the explicit scheme beyond its stability
> limit ($|\gamma_1| > 1$). We will clarify this directly in the figure caption in
> the revised manuscript.
>
> > For the Burger's equation experiment, it is stated that "for this experiment,
> > we train a vanilla UNet ... on only the initial "smooth" frames from this
> > low-fidelity solver." Does that mean that the UNet is trained with a limited
> > time-series rollout from the low fidelity solver and then during the test
> > phase, it is made to do a longer rollout than it was trained with?
>
> The reviewer's understanding is correct. The training dataset consists
> only of one-step transitions $(u^{[0]}, u^{[1]})$ where $u^{[0]}$ is a smooth
> initial state (low-frequency content). During testing, the emulator is
> initialized with similar smooth states but must perform a longer autoregressive
> rollout (10 steps). During this longer rollout, the solution develops sharp
> shocks (high-frequency content). The superiority arises because the UNet,
> despite only being trained on smooth-to-smooth transitions, learns a physical
> operator that correctly propagates these emergent shocks more accurately than
> the truncated Picard solver it was trained on.
>
> > - Certain figure captions and text refer [...]
> > - Some equations in the appendix are too long [...]
>
> We thank the reviewer for catching these formatting issues and will improve our
> manuscript accordingly.

---

> > ### Comment · Reviewer_8yEP · 2025-08-07
> > **Response to Author Rebuttal**
> >
> > Thank you to the authors for their detailed responses to the posed questions. This has clarified many points about the proposed method.

---

> > > ### Author Response · Authors · 2025-08-07
> > >
> > > Thank you for your review and for acknowledging that our responses have clarified your questions. We appreciate your valuable feedback, which has helped us identify key areas for improvement in the manuscript.
> > >
> > > We are grateful for your time and thoughtful engagement with our work.

---

### Official Review · Reviewer_k6sC · 2025-07-02

**Clarity:** 3
**Significance:** 3
**Originality:** 3
**Rating:** 4
**Confidence:** 3

**Summary:**

This paper studies a surprising phenomenon in neural PDE modelling where a neural network solver trained only on low-fidelity numerical solver can outperform that solver in accuracy when evaluated against high-fidelity simulation results. The author conducts analysis on linear and non-linear PDEs with both linear and non-linear ansatz like UNet.

**Questions:**

Can the authors provide a general intuition for what kinds of non-linear PDEs are more amenable to neural networks with limited receptive fields (e.g., feedforward ConvNets), and which ones may require larger context (e.g., long-range dependencies in turbulent flows or stiff PDEs)?

**Ethical Concerns:**

["NO or VERY MINOR ethics concerns only"]

**Final Justification:**

The paper studies an interesting topic and provides clarification in the rebuttal stage. Despite certain limitations related to problem scope and the novelty of the theoretical analysis, I lean towards acceptance and keep my original rating.

**Limitations:**

The authors acknowledge one major limitation in the paper, the solver needs to be sufficiently “coarse” for neural emulator to exhibit superiority.

**Quality:**

3

**Strengths And Weaknesses:**

Strengths:

1. The study of whether a neural solver trained on low-fidelity data can outperform the source is novel and important to scientific ML.

2. The analysis for linear PDEs is clear and provides intuition for more complex non-linear cases. They show how superiority arises by analyzing linearized spectral errors and optimal filters.

Weakness:

1. The experiments and analysis are limited to 1D PDEs, which are far from the complexity of most real-world applications. The analysis for linear PDEs relies heavily on mode-wise error decomposition, which works for 1D linear PDEs due to the orthogonality and separability of modes. It remains unclear whether the analysis will extend to more realistic scenarios involving nonlinear, multi-dimensional PDEs, such as those found in fluid dynamics, where mode interactions, geometry are significantly more complex and high-fidelity simulations are expensive or difficult to obtain.

2. The analysis of nonlinear neural architectures (e.g., UNets, Transformers, FNOs) is relatively scarce. It could be interesting to see more in-depth discussion regarding the relationship between effective receptive field of model and its rollout error / superiority.

---

> ### Author Rebuttal · Authors · 2025-07-30
>
> We thank the reviewer for highlighting the novelty of our core question and for
> finding our analysis of linear PDEs clear and intuitive. It was precisely our
> goal to build a solid, understandable foundation that could inform research on
> more complex systems. The feedback is highly constructive, and we appreciate the
> opportunity to clarify our contributions and strengthen the paper by addressing
> the points raised.
>
> ---
>
> > The experiments and analysis are limited to 1D PDEs, which are far from the
> > complexity of most real-world applications. The analysis for linear PDEs
> > relies heavily on mode-wise error decomposition, which works for 1D linear
> > PDEs due to the orthogonality and separability of modes. It remains unclear
> > whether the analysis will extend to more realistic scenarios involving
> > nonlinear, multi-dimensional PDEs, such as those found in fluid dynamics,
> > where mode interactions, geometry are significantly more complex and
> > high-fidelity simulations are expensive or difficult to obtain.
>
> The reviewer raises a very important point regarding the generalization of our
> findings. We agree that extending this analysis to multi-dimensional, nonlinear
> systems is the critical next step. Our choice to focus on 1D PDEs was a
> deliberate methodological decision to ensure rigor and clarity in establishing
> this new phenomenon.
>
> The 1D setting, especially for linear PDEs, allows for exact analytical
> treatment via Fourier analysis. This enabled us to provide a closed-form proof
> for the existence of superiority (Section 3), isolating the core mechanism
> without confounding factors from complex geometries or multi-dimensional mode
> interactions. Establishing this principle on a solid theoretical footing was our
> primary goal.
>
> We believe our findings are more generalizable than they might appear at first
> glance. Many complex problems in fluid dynamics are governed by equations (e.g.,
> Navier-Stokes) that can be seen as a combination of fundamental physical
> processes: nonlinear convection (like the Burgers' equation), diffusion, and an
> incompressibility constraint (enforced via a Poisson equation). Our work
> demonstrates the principle of superiority for each of these core building
> blocks. It is therefore reasonable to hypothesize that the phenomenon will also
> manifest in more complex systems that combine these elements. Our paper provides
> the foundational understanding necessary to begin investigating these more
> challenging scenarios.
>
> > The analysis of nonlinear neural architectures (e.g., UNets, Transformers,
> > FNOs) is relatively scarce. It could be interesting to see more in-depth
> > discussion regarding the relationship between effective receptive field of
> > model and its rollout error / superiority.
>
>
> This is an excellent suggestion. A deeper dive into how architectural properties
> influence superiority would indeed strengthen the paper. We will expand on this
> in the revised manuscript. Below, we present additional interpretation and new
> results that provide more intuition.
>
> ### Performance on Linear vs. Nonlinear Problems
>
> In the linear advection case (Figure 4), we saw that only architectures with
> strong, constraining inductive biases (like the local ConvNet) could achieve
> state-space superiority. Higher-capacity models like UNet and FNO failed to do
> so, as the problem lacked the dynamic complexity to prevent them from simply
> overfitting to the solver's errors.
>
> However, the situation changes for the nonlinear Burgers' equation, where the
> dynamics naturally generate a rich spectrum of interacting modes. To provide a
> more complete picture, we extended our Burgers' experiment to all architectures.
> The results robustly confirm that superiority is not limited to a single model
> but is instead closely tied to having a favorable inductive bias. We found that
> all architectures with a spatial or spectral bias (ConvNet, Dilated ResNet, FNO,
> UNet) achieve significant superiority by learning a better representation of the
> emergent shock dynamics. The Dilated ResNet was a standout performer, reaching a
> superiority ratio of 0.65. In contrast, the vanilla Transformer, which lacks an
> inherent structural bias, failed to achieve superiority.
>
> Below are the median superiority ratios ($\xi^{[t]}$) for the Burgers' equation
> experiment.
>
> |Network/Time Step|1|2|3|4|5|6|7|8|9|10|
> |:------------------|--:|--:|--:|--:|--:|--:|--:|--:|--:|--:|
> |ConvNet|1|0.97|0.87|0.82|0.8|0.8|0.8|0.81|0.81|0.81|
> |Dilated ResNet|1|0.91|0.79|0.73|0.69|0.66|0.65|0.67|0.66|0.65|
> |FNO|1|1.01|0.91|0.87|0.86|0.89|0.93|1|1.08|1.12|
> |Transformer|1|1.06|1.13|1.24|1.35|1.45|1.52|1.58|1.63|1.68|
> |UNet|1|0.9|0.86|0.88|0.91|0.96|1|1.04|1.07|1.11|
>
> This suggests that even high-capacity models can achieve superiority in
> physically rich settings, provided their inductive bias aligns with the problem
> structure (e.g., local convolutions or spectral filters). It should be noted
> that modern Transformer variants like SWin resemble a convolution-like inductive
> bias and could thus again show superiority.
>
> ### Ablation Study on Model Capacity and Inductive Bias
>
> To further isolate the effect of architectural properties, we conducted new
> ablation studies on both ConvNet and FNO architectures, this time on the linear
> advection problem. The detailed results can be found in our response to Reviewer
> s59z, but the key findings strongly reinforce the importance of matching the
> model's inductive bias to the problem.
>
> For the ConvNet, we systematically varied its receptive field. We observed that
> superiority was maximized when the receptive field size was closely aligned with
> the physical scale of information propagation (dictated by the CFL number).
> Receptive fields that were too small failed to learn the dynamics, while those
> that were too large provided excess capacity that allowed the model to begin
> replicating the solver's non-physical errors, thus weakening the superiority
> effect.
>
> We found a similar trend for the FNO when varying its number of active Fourier
> modes. A configuration with just enough modes to represent the core physics
> achieved the best performance, as the FNO's spectral truncation acted as a
> powerful regularizer against the solver's high-frequency artifacts. As we added
> more modes, the FNO's increased capacity led to it partially overfitting to the
> solver's error profile, diminishing the magnitude of superiority.
>
> Together, these studies provide strong evidence that achieving superiority is a
> direct consequence of the interplay between model capacity and implicit
> regularization. The best results occur when an architecture is expressive enough
> to capture the necessary physics but constrained enough by its inductive bias to
> regularize away the structured flaws of its training data.
>
> > Can the authors provide a general intuition for what kinds of non-linear PDEs
> > are more amenable to neural networks with limited receptive fields (e.g.,
> > feedforward ConvNets), and which ones may require larger context (e.g.,
> > long-range dependencies in turbulent flows or stiff PDEs)?
>
> This is a very insightful question that gets to the heart of designing effective
> neural emulators (independent of whether one wants to achieve superiority). Our
> general intuition is that the ideal receptive field size is dictated by the
> physical characteristics of information propagation within the PDE system over a
> single time step. In the case of the prototypical advection equation, this is
> dictated by the CFL number which we used to parameterize our experiments.
>
> For more complicated problems, this depends on the dominant physical effect. If
> the problem behaves primarily hyperbolic, e.g., compressible Navier-Stokes
> simulations, then there might be a local region of influence and as long as the
> time steps are not too large (i.e., information does not travel too far) local
> convolutional architectures can be used. However, if the dominant effect
> includes a Poisson-like component (like incompressibility constraints for
> incompressible Navier-Stokes simulations or volume conservation or other global
> constraints) that could require global information to be propagated. There is
> reason to believe that local emulators can to some extent approximate those
> behaviors but very likely will not achieve the same performance as global
> emulators. We refer the reviewer to benchmark works that attempt to provide
> corresponding data points more holistically, e.g., PDEBench, PDEArena or
> APEBench.
>
> > The authors acknowledge one major limitation in the paper, the solver needs to
> > be sufficiently “coarse” for neural emulator to exhibit superiority.
>
> This is an important and valid limitation. The requirement that the low-fidelity
> solver be "sufficiently coarse" for emulator superiority to emerge may seem
> restrictive at first glance. However, we argue that this is not an unreasonable
> assumption in practice. In many real-world scientific and engineering
> applications, practitioners routinely make deliberate simplifications or
> approximations to reduce computational cost or to enable tractable simulations.
> For example, truncating nonlinear solves is common practice in algorithms like
> PISO for incompressible flow, and reduced-order models are widely used in
> climate and fluid dynamics. These practical choices often introduce systematic
> errors or coarsen the solution, creating precisely the kind of setting where a
> neural emulator could, in principle, learn to correct or surpass the baseline
> solver.
>
> Thus, our focus on scenarios with intentionally limited or coarse solvers is
> well-motivated by practical considerations. While our results suggest that
> neural emulators can sometimes recover or even exceed the accuracy of the
> original solver in these settings, we acknowledge that this is not something we
> can guarantee in general at this stage. The phenomenon is promising, but its
> robustness and generality across broader classes of problems remain important
> directions for future research.

---

> > ### Comment · Reviewer_k6sC · 2025-08-05
> >
> > I would like to thank the authors for the response and additional experiments. I will keep my orginal score.

---

> > > ### Author Response · Authors · 2025-08-05
> > >
> > > We thank the reviewer for their reply.

---

### Official Review · Reviewer_s59z · 2025-07-06

**Clarity:** 4
**Significance:** 3
**Originality:** 3
**Rating:** 5
**Confidence:** 3

**Summary:**

This paper introduces emulator superiority which defines the phenomenon that machine learning models trained on low-fidelity solver may outperform the data-generating solver in terms of several metrics that measure autoregressive generalization, state-space generalization as defined in the paper. This emulator superiority is brought about by two primary sources: smaller accumulation error from multi-step rollouts and the inductive bias of the specific neural network used as an operator solver.

**Questions:**

(1) Do any variations in the model contribute to higher superiority? Such as the kernel size or the number of modes in FNO. These experiments may provide guidance when solving a new problem not shown in this paper, such as Navier-Stokes.

(2) Real-world observations often contain structured noise. Assuming we use such data rather than idealized solvers, what would this mean to this superiority?

**Ethical Concerns:**

["NO or VERY MINOR ethics concerns only"]

**Final Justification:**

My concerns and questions are well explained and also I reviewed other reviewers' points. So, I increased my score to accept.

**Limitations:**

Yes

**Paper Formatting Concerns:**

None.

**Quality:**

3

**Strengths And Weaknesses:**

Strength:

- This paper clearly illustrates experimental motivations, preliminaries and experimental settings, helping readers to follow. The granularity of discussions seems well spread over the paper --- where main descriptions are in the manuscript and in-depth mathematical descriptions are appropriately deferred to appendix.

- Motivations are well described and strong – that I agree the conventional wisdoms goes that the model is as good as the data (but not better). However, recent empirical findings suggest strong extrapolating capacities of neural networks especially in predicting high frequency modes given low-fidelity data, which calls for the analyses over the underlying reasons of such.

- This paper seems novel in that it differs from prior approaches which explicitly trains a model to perform superresolution or extrapolation.

Weaknesses:

- Limited nonlinear problems: would bear larger contribution if the superiority is met across a more complex and harder prediction tasks.

- While several potential error source are enumerated, such as spatial discretization errors, linearization errors, etc, it seems impossible to dissect exactly why and where this emulator superiority arises seems a daunting task. E.g., what inductive bias of ConvNet / ResNET / FNO / Transformer constitute for what portion of the superiority is hard to disentangle. In such case, the practical value of this work seems questionable.

---

> ### Author Rebuttal · Authors · 2025-07-30
>
> We sincerely thank the reviewer for their thoughtful and constructive feedback.
> Their recognition of our paper's strong motivation, clear presentation, and
> novelty is very encouraging. It is especially gratifying to find agreement on
> the importance of challenging the "model is as good as the data" paradigm. We
> appreciate the opportunity to address the reviewer's comments and believe the
> resulting discussion and additions will significantly strengthen our manuscript.
>
> ---
>
> > Limited nonlinear problems: would bear larger contribution if the superiority
> > is met across a more complex and harder prediction tasks.
>
> We agree with the reviewer that demonstrating superiority in more complex
> settings is an important future direction. Our decision to focus on foundational
> linear PDEs and the Burgers' equation was a deliberate methodological choice
> aimed at achieving maximum clarity and rigor.
>
> Complex systems, such as the Navier-Stokes equations, involve multiple, coupled
> sources of numerical error (e.g., from discretization, linearization, and
> pressure-correction steps). In such settings, isolating and analyzing the
> specific interactions that give rise to emulator superiority would be
> difficult. By starting with simpler, well-understood systems, we
> were able to provide a rigorous theoretical proof (Section 3) for the existence
> of superiority. This controlled environment allowed us to precisely attribute
> the phenomenon to the interplay between an emulator's inductive bias and the
> structured, mode-dependent errors of the low-fidelity solver (as shown in Figures
> 1 and 2), providing a clear and interpretable foundation for the concept.
>
> We believe that this stepwise approach—beginning with canonical PDEs and
> gradually moving toward more complex, multi-physics systems—is essential for
> building a robust understanding of emulator superiority. Importantly, many
> complex PDEs, such as the Navier-Stokes equations, are fundamentally composed of
> the same building blocks we have investigated: advection, diffusion, and
> nonlinear interactions. By rigorously demonstrating the superiority phenomenon
> across these foundational components, we provide strong evidence that the
> principle is not an isolated curiosity, but rather a fundamental aspect of
> learning from numerical data that is likely to persist in more complicated,
> real-world systems. While additional sources of error and coupling may introduce
> new challenges, it is not unreasonable to hypothesize that our insights will
> generalize to these richer domains. We are committed to extending our analysis
> to such settings in future work, and see our current results as a necessary and
> rigorous foundation for that endeavor.
>
>
> > While several potential error source are enumerated, such as spatial
> > discretization errors, linearization errors, etc, it seems impossible to
> > dissect exactly why and where this emulator superiority arises seems a
> > daunting task. E.g., what inductive bias of ConvNet / ResNET / FNO /
> > Transformer constitute for what portion of the superiority is hard to
> > disentangle. In such case, the practical value of this work seems
> > questionable.
>
> This is a crucial point that connects to our choice of problems. The primary
> practical value of our work lies in formally identifying and proving the
> existence of the emulator superiority phenomenon itself. Our linear examples
> were purposefully chosen precisely because they do allow us to disentangle the
> sources of superiority in a controlled manner. The practical implication is
> immediate: our findings challenge the fundamental assumption that an emulator is
> strictly bound by its training data's fidelity. This has direct consequences for
> how the community benchmarks and evaluates neural emulators. As we argue (lines
> 342-347), using flawed numerical references can unfairly penalize emulators that
> have correctly learned more physically plausible dynamics. Our work provides the
> vocabulary and framework to understand this paradoxical effect. While
> disentangling every factor in a highly complex system remains a future
> challenge, establishing the core principle on a solid theoretical and empirical
> foundation is, we believe, a significant and practical contribution.
>
> > Do any variations in the model contribute to higher superiority? Such as the
> > kernel size or the number of modes in FNO. These experiments may provide
> > guidance when solving a new problem not shown in this paper, such as
> > Navier-Stokes.
>
> This is an excellent question that gets to the heart of how architectural
> choices influence superiority. Our results in Figure 4c already hint that local
> architectures (ConvNet) can achieve stronger state-space superiority than global
> ones (FNO, Transformer). We hypothesize this is because a constrained, local
> inductive bias acts as a powerful regularizer, preventing the model from
> overfitting to the global error patterns of the low-fidelity solver.
>
> To provide direct evidence and practical guidance, we performed a new ablation
> study on the linear advection case, systematically varying the effective
> receptive field of the ConvNet. The key finding is that superiority is maximized
> when the receptive field is appropriately matched to the physical
> characteristics of the problem (in this case, dictated by the CFL number). A
> receptive field of 4 yields the best performance, achieving a superiority ratio
> $\xi^{[t]}$ of 0.73 after 10 time steps. Receptive fields that are too small
> fail to capture the necessary physics, while those that are too large provide
> excess capacity that diminishes the superiority effect by allowing the model to
> learn more of the solver's non-physical behavior.
>
> The table below shows the superiority ratio rollout for ConvNets with different
> receptive fields. The configuration with a receptive field of 11 corresponds to
> the model used in the main paper.
>
> |Rec.Field/TimeStep|1|2|3|4|5|6|7|8|9|10|
> |-----------------:|--:|--:|--:|--:|--:|--:|--:|--:|--:|--:|
> |1|4.0|4.0|3.9|3.9|3.9|3.8|3.8|3.7|3.7|3.6|
> |2|1.0|1.0|1.0|1.0|1.0|1.0|1.0|1.0|1.0|1.0|
> |3|1.0|0.95|0.91|0.88|0.86|0.84|0.83|0.81|0.80|0.80|
> |4|1.0|0.94|0.90|0.86|0.83|0.80|0.78|0.76|0.75|0.73|
> |5|1.0|0.94|0.90|0.86|0.83|0.81|0.79|0.77|0.76|0.74|
> |7|1.0|0.96|0.93|0.91|0.89|0.87|0.86|0.85|0.84|0.83|
> |11|1.0|0.96|0.93|0.91|0.89|0.87|0.86|0.84|0.83|0.82|
>
> We also conducted an ablation study on the Fourier Neural Operator (FNO) by
> varying its number of active Fourier modes. The results reveal a clear trade-off
> between model capacity and the implicit regularization that drives superiority.
> When severely under-parameterized (1 active mode), the FNO fails to learn the
> problem. The optimal performance is achieved at 2 active modes, which provides
> just enough capacity to model the target dynamics while the FNO’s inherent
> spectral truncation acts as a powerful regularizer, filtering out numerical
> artifacts from the solver and leading to strong superiority (ratio of 0.59). As
> the number of active modes increases further (3+), the FNO gains the capacity to
> partially overfit to the solver's structured errors across a wider frequency
> band, which, while still allowing for superiority, diminishes its magnitude.
> This demonstrates that superiority is maximized when the model is expressive
> enough to capture the core physics but constrained enough to regularize away the
> training data's flaws.
>
> |Act. Modes/Time Step|1|2|3|4|5|6|7|8|9|10|
> |---------------:|-----:|-----:|-----:|-----:|-----:|-----:|-----:|-----:|-----:|-----:|
> |1|8.16|8.06|7.93|7.76|7.58|7.37|7.14|6.89|6.62|6.35|
> |2|1|0.93|0.87|0.81|0.76|0.72|0.68|0.65|0.62|0.59|
> |3|1|0.97|0.94|0.92|0.9|0.88|0.86|0.84|0.82|0.81|
> |4|1|0.98|0.97|0.96|0.94|0.93|0.92|0.92|0.91|0.9|
> |8|1|0.97|0.94|0.91|0.88|0.86|0.84|0.82|0.8|0.78|
> |12|1|0.98|0.96|0.95|0.93|0.92|0.91|0.9|0.89|0.88|
>
> > Real-world observations often contain structured noise. Assuming we use such
> > data rather than idealized solvers, what would this mean to this superiority?
>
> The reviewer raises an important point about the applicability of our framework
> to real-world data. Our formal definition of "emulator superiority" (Equation 5)
> fundamentally relies on the existence of three distinct components: a
> low-fidelity solver (the training source), a high-fidelity solver (the reference
> "truth"), and the emulator.
>
> In the case of real-world observations, there is no explicit "low-fidelity
> solver," only the data-generating process of nature, and the "high-fidelity"
> ground truth is unknown. Therefore, our superiority analysis, as framed, does
> not directly apply.
>
> However, we can address the spirit of the question by considering a modified
> scenario where perfect synthetic data is corrupted by structured noise. In this
> case, the emulator's ability to "outperform" its noisy training data would
> depend on its capacity for denoising. This relates to the standard machine
> learning insight that models can learn robust representations from noisy
> examples, but it is a phenomenon that is distinct from correcting the structured
> numerical errors that are the focus of our paper.
>
> Another scenario would be that only initial conditions are corrupted by
> (observational) noise, while the dynamics themselves are noiseless. In such
> cases, the impact on superiority depends critically on the structure and
> intensity of the noise. If the noise significantly alters the spectral content
> of the initial state, it may obscure the clear separation of modes or patterns
> required for the emulator to achieve state-space or autoregressive superiority.
> In particular, strong or broadband noise can mask the underlying signal, making
> it difficult for the emulator to distinguish between true dynamics and
> noise-induced artifacts, and thus potentially undermining the conditions
> necessary for superiority to emerge. However, in such a setting we also believe
> that learning a good emulator will be generally challenging.
>
> We believe this is a fascinating avenue for future work. We will add a
> discussion to the Limitations & Outlook section.

---

> > ### Comment · Reviewer_s59z · 2025-08-07
> >
> > My concerns and questions are well explained. This is an interesting research topic. This challenging topic is well modeled and exposited. Thank you for your effort, and I lean toward acceptance.

---

> > > ### Author Response · Authors · 2025-08-07
> > >
> > > Thank you for your thoughtful and constructive review. We are very grateful for your positive assessment of our work and are especially encouraged by your statement that you lean towards acceptance.
> > >
> > > Your questions prompted us to conduct new ablation studies that directly led to a stronger, more practical set of results. We believe your feedback has significantly enhanced the paper's contribution.
> > >
> > > Thank you again for your engagement and support.

---

### Decision · Program_Chairs · 2025-09-17

**Decision:**

Accept (poster)

**Comment:**

This paper formalizes and studies an interesting phenomenon where neural operators trained on data produced from low-fidelity solver (e.g., only apply 1 iteration of spatial Newton solver in each time step) can sometimes surpass the solver itself when compared to high-fidelity references (analytical for example). All reviewers found the experiments interesting, so did I. I would like to mention that there is some new development in interpreting operator learning as a statistical learning method (i.e., discretization invariance is really an implication of in-distribution inference), see e.g., Reinhardt et al. arXiv:2412.17582, so the coarse data still may correctly capture the correct distribution.

Here are my revision suggestions in addition to the reviewers': CFL condition ($\gamma$, or Courant number) is usually given as positive, please refer to any FDM textbook and fix this. Note for heat eq-like spatiotemporal problems (3.2), bounds on $\gamma_2$ are refered to as CFL condition as well (for explicit marching schemes). Section 3 title reads "Proof", however, after reading it I felt it was more like "A Case for Supriority". The letter $m$ and $M$ need to be more clearly defined in Section 3, I have to refer to the Appendix to find their meanings in Figure 1.